# Non-Hermitian Floquet Topological Matter—A Review

**DOI:** 10.3390/e25101401

**Published:** 2023-09-29

**Authors:** Longwen Zhou, Da-Jian Zhang

**Affiliations:** 1College of Physics and Optoelectronic Engineering, Ocean University of China, Qingdao 266100, China; 2Key Laboratory of Optics and Optoelectronics, Qingdao 266100, China; 3Engineering Research Center of Advanced Marine Physical Instruments and Equipment of MOE, Qingdao 266100, China; 4Department of Physics, Shandong University, Jinan 250100, China

**Keywords:** Floquet system, non-Hermitian system, topological phases of matter

## Abstract

The past few years have witnessed a surge of interest in non-Hermitian Floquet topological matter due to its exotic properties resulting from the interplay between driving fields and non-Hermiticity. The present review sums up our studies on non-Hermitian Floquet topological matter in one and two spatial dimensions. We first give a bird’s-eye view of the literature for clarifying the physical significance of non-Hermitian Floquet systems. We then introduce, in a pedagogical manner, a number of useful tools tailored for the study of non-Hermitian Floquet systems and their topological properties. With the aid of these tools, we present typical examples of non-Hermitian Floquet topological insulators, superconductors, and quasicrystals, with a focus on their topological invariants, bulk-edge correspondences, non-Hermitian skin effects, dynamical properties, and localization transitions. We conclude this review by summarizing our main findings and presenting our vision of future directions.

## 1. Introduction

When a physical system is driven periodically in time, its properties can be drastically modified, leading to new phases and phenomena beyond the static limit [1,2,3,4,5,6]. One such example, which can be traced back to the early history of human civilization, is the Archimedes screw pump. Under periodic operations, the Archimedes screw could transfer water from low-lying rivers into high-lying irrigation ditches, rather than letting the water follow its natural flowing tendency. Another example is the tide caused by the combined effects of the gravitational forces exerted by the Moon and its periodic orbiting around the Earth. In the quantum domain, rich nonequilibrium features have been identified in periodically driven systems over the past decades, such as the Rabi oscillation [7,8,9], stimulated Raman adiabatic passage [10,11,12], dynamical localization [13,14,15], Thouless pump [16,17,18], time crystal [19,20,21], Floquet topological phase [22,23,24] and integer quantum Hall effect from chaos [25,26,27] (for more information, see the reviews [28,29,30,31,32,33,34,35,36,37,38,39,40,41,42,43,44,45,46,47,48,49,50,51,52,53,54,55,56,57,58,59,60] and references therein). Among all these discoveries, Floquet topological matter stands out as one pivotal impetus in the study of periodically driven quantum systems. Over the past twenty years, it has attracted great attention in the context of quantum dynamics [33,36,37], quantum simulation [34,35,43], condensed matter physics [38,39,42], and so on.

The topological properties of a periodically driven system are mainly coded in its Floquet states, which are eigenstates of the evolution operator over a driving period. This one-period propagator is called the Floquet operator. For a system described by the Hamiltonian $\hat{H}\left(t\right)=\hat{H}\left(t+T\right)$, the Floquet operator can be expressed as $\hat{U}=\hat{T}e^{-i\int_{t_{0}}^{t_{0}+T}\hat{H}\left(t\right)dt}$. Here *t* denotes time, *T* is the driving period, $t_{0}$ denotes the initial time of the period, $\hat{T}$ performs the time ordering, and we have set the Planck constant ℏ=1. Solving the eigenvalue Equation $\hat{U}|\psi \rangle =e^{-iE}|\psi \rangle$ gives us the Floquet eigenstate |ψ〉 and the quasienergy *E*. The latter is a phase factor and defined modulus 2π. Its range E∈[−π,π) is usually referred to as the Floquet quasienergy Brillouin zone. Moreover, the quasienergy is stroboscopically conserved and plays the role of “energy” in driven systems with only discrete-time translational symmetries. If the system also obeys the spatial translational symmetry, the quasienergy dispersion E(k) with respect to the conserved quasimomentum k could also be grouped into bands, which are thus called Floquet bands. Under appropriate conditions, these quasienergy bands could show nontrivial topological properties, which can be captured by topological invariants of the corresponding Floquet–Bloch states or the Floquet operator itself. Periodically driven quantum systems could, thus, support a new class of nonequilibrium phases, which is nowadays known as Floquet topological matter.

Similar to static topological phases, Floquet topological phases can also possess symmetry-protected edge states under open boundary conditions and exhibit quantized dynamical or transport signals. However, three key features distinguish Floquet topological phases from their static counterparts. First, suitable driving fields can break the symmetry dynamically and open gaps around the touching points of static energy bands, yielding band inversions and topologically nontrivial band structures [22]. This aspect of driving can usually be taken into account by the Floquet effective Hamiltonians obtained from high-frequency expansions of the driving potential [61,62,63,64]. Second, as the quasienergy *E* is bounded from above (E=π) and below (E=−π), Floquet bands can meet with each other at E=±π and develop nontrivial windings around the whole quasienergy Brillouin zone E∈[−π,π). These spectral windings result in unique states of matter in periodically driven systems, such as Floquet semi-metals with Floquet band holonomy [65,66,67], degenerated edge modes at E=±π (anomalous Floquet π modes) [68,69,70,71] and anomalous chiral edge states in Floquet topological insulators (FTIs) [72,73,74], which have no counterparts in static systems. Third, the driving field could assist in the formation of long-range, and even spatially non-decaying coupling among different degrees of freedom in a lattice [24], leading to Floquet phases with large topological invariants, rich topological transitions, and a substantial number of topological edge states [75,76,77,78,79,80,81,82]. These phases and boundary states go beyond the description of any static tight-binding models in typical situations (i.e., with finite-range or spatially decaying hopping amplitudes). The investigation of the above-mentioned features has not only led to new classification schemes of Floquet matter in theories [83,84,85,86] but also promoted experimental realizations of numerous Floquet topological phases in solid-state and quantum simulator setups [87,88,89,90,91,92,93,94,95,96,97,98,99,100,101,102], bringing new hope for applications in ultrafast electronics [38] and topological quantum computing [103,104,105].

Non-Hermitian physics deals with classical and open quantum systems subject to measurements, dissipation, gain, and loss, or nonreciprocal effects [106,107,108,109]. A physical system described by a Hamiltonian $\hat{H}$ is regarded as non-Hermitian if $\hat{H}$ is not self-adjoint, i.e., $\hat{H}\neq {\hat{H}}^{†}$. The spectrum of such a non-Hermitian Hamiltonian is complex in general and the resulting dynamics is non-unitary. In recent years, the study of topological phases in non-Hermitian systems has attracted much attention from both theoretical and experimental sides (see Refs. [110,111,112,113,114,115,116,117,118,119,120,121,122,123,124,125,126,127,128,129,130,131,132,133,134] for reviews). The investigation of these nonequilibrium phases may help to deepen our understanding of topological matter in open systems and optimize the design of noise-resilient quantum devices.

The recent boom in non-Hermitian topological matter is driven by a couple of key concepts, including the parity and time-reversal (PT) symmetry, the exceptional point (EP), and the non-Hermitian skin effect (NHSE). PT symmetry can appear in open systems with balanced gain and loss. A PT-symmetric, non-Hermitian Hamiltonian has a real spectrum in the PT unbroken regime [135,136]. PT symmetry was thus viewed as a possible way to lift the restriction of Hermiticity for Hamiltonians in early studies [109]. Recently, it was found that the topological and transport properties of a system may also change when it undergoes a PT-symmetry-breaking transition, yielding phases unique to non-Hermitian Hamiltonians [137,138,139]. Models with PT symmetry, thus, became a focus on the study of non-Hermitian topological matter [112,113,114,116,117,120,126,134]. The combination of PT and other symmetries further results in various classification schemes for non-Hermitian topological phases that go beyond their Hermitian counterparts [140,141,142,143,144,145,146,147,148]. The EP is a class of level degeneracy point unique to non-Hermitian operators. At an EP, the geometric and algebraic multiplicities of a non-Hermitian matrix are different, leading to the breakdown of its diagonalizability and the coalescence of its eigenvectors [106,110,111]. In recent years, various gapless band structures (nodal points, lines, loops, surfaces, knots, etc.) formed by EPs were discovered, giving rise to rich non-Hermitian topological (semi-)metallic phases with intriguing transport properties [118,120,124,125,127,149]. In addition, EPs were also found to play key roles in topological energy transfer [150,151,152], high-precision sensing [153,154,155,156], topological lasers [157,158,159], and strongly correlated phases [124,125]. The NHSE refers to the accumulation of bulk states around the edges of an open-boundary non-Hermitian lattice. It highlights the extreme sensitivity of non-Hermitian physics to the boundary conditions of a system [122,128,133]. This phenomenon not only blurs the distinction between bulk and edge states in non-Hermitian models but also breaks the well-established bulk–boundary correspondence in Hermitian topological matter [160,161,162,163,164]. Over the past few years, a couple of theoretical frameworks have been introduced to characterize the NHSE and its related topological phenomena [165,166,167,168,169,170,171,172,173,174,175], which are accompanied by the experimental observations of the NHSE in AMO systems, electrical circuits, and metamaterials [176,177,178,179,180,181,182,183]. Entanglement transitions associated with the NHSE were also identified in a recent study [184]. Further progress has been made in the study of non-Hermitian topological matter in randomly disordered [185,186,187,188,189], quasiperiodic [190,191,192,193,194], and many-body systems [195,196,197,198,199,200,201,202,203,204,205,206,207,208,209].

With all these developments, a natural follow-up is to consider the system in a more general situation, in which it is subject to both time-periodic drivings and non-Hermitian effects. Such non-Hermitian Floquet systems may possess exotic dynamical phenomena and topological phases with no static or Hermitian analogies. On the theoretical side, the investigation of driven non-Hermitian systems may lead to the discovery of new topological states and bring about the improvement of classification schemes for nonequilibrium phases of matter in general. On the practical side, the exploration of non-Hermitian Floquet matter is helpful to the design of new approaches for preparing or stabilizing topologically nontrivial states and controlling material properties. It also stimulates new ideas for realizing quantum devices and quantum computing protocols that are robust to perturbations caused by the environment. Though still at the early stage, much progress has been made in the realization and characterization of non-Hermitian Floquet phases [210,211,212,213,214,215,216,217,218,219,220,221,222,223,224,225,226,227,228,229,230,231,232,233,234,235,236,237,238,239,240,241,242,243,244,245,246,247,248,249,250,251,252,253,254,255]. In this review, we limit our scope to the discussion of a number of typical topological phases we discovered in non-Hermitian Floquet systems [256,257,258,259,260,261,262,263,264,265,266]. A schematic illustration is given in Figure 1. In Section 2, we give a pedagogical introduction to some key aspects of Floquet systems, including their dynamical and topological characterizations. In Section 3, we present typical examples of non-Hermitian Floquet topological insulators, superconductors, and quasicrystals and summarize their main physical properties, with a focus on the features that are unique to driven non-Hermitian systems. In Section 4, we conclude this review, briefly mention some relevant studies, and discuss potential future work.

## 2. Background

We start with a recap of the basic description of a non-Hermitian Floquet system. The Hamiltonian of such a system satisfies $\hat{H}\left(t\right)=\hat{H}\left(t+T\right)$, and there exists t∈[0,T] such that $\hat{H}\left(t\right)\neq {\left[\hat{H}\left(t\right)\right]}^{†}$. Here *t* denotes time and *T* is the driving period. The state of the system evolves according to the Schrödinger equation
(1)$$i\frac{\partial }{\partial t}|\Psi \left(t\right)\rangle =\hat{H}\left(t\right)|\Psi \left(t\right)\rangle ,$$
where we have set ℏ=1. We first show that this equation can be solved using Floquet states even though $\hat{H}\left(t\right)$ is non-Hermitian. This is followed by different ways of obtaining the Floquet states in general situations, in the high-frequency regime, and in the adiabatic regime. We next discuss the symmetry, topological invariants, and dynamical characterizations of non-Hermitian Floquet states, with a focus on the types of physical systems explored in our previous studies. We conclude this section by presenting some tools for characterizing the spectrum properties and localization transitions in non-Hermitian Floquet disordered systems.

### 2.1. Floquet Theorem

We sketch a proof of the Floquet theorem in this subsection [267]. It follows the proof of the Bloch theorem for waves in one-dimensional periodic lattices [268]. Applying the Fourier expansion to our time-periodic Hamiltonian $\hat{H}\left(t\right)$ in the time–frequency domain, we find
(2)$$\hat{H}\left(t\right)=\sum_{n}\hat{H}\left(\omega_{n}\right)e^{i\omega_{n}t},\;\omega_{n}=n\omega =n\frac{2\pi }{T},\;n\in Z.$$
Here, ω is the driving frequency. For an infinite system, the set of plane waves $\{|\psi_{\varepsilon }\left(t\right)\rangle =|\varphi \left(r\right)\rangle e^{-i\varepsilon t}\}$ can be chosen as a suitable basis. We can write down a matrix expression for $\hat{H}\left(t\right)$ in the orthonormal and complete basis $\left\{|\psi_{\varepsilon }\left(t\right)\rangle \right\}$. Acting $\hat{H}\left(t\right)$ on $|\psi_{\varepsilon }\left(t\right)\rangle$, we obtain
(3)$$\hat{H}\left(t\right)|\psi_{\varepsilon }\left(t\right)\rangle =\sum_{n}\hat{H}\left(\omega_{n}\right)|\varphi \left(r\right)\rangle e^{-i(\varepsilon -\omega_{n})t}=\sum_{n}\hat{H}\left(\omega_{n}\right)|\psi_{\varepsilon -\omega_{n}}\left(t\right)\rangle .$$
For any given n∈Z, the state $|\psi_{\varepsilon -\omega_{n}}\left(t\right)\rangle$ belongs to the subspace
(4)$$S_{\varepsilon }=\{|\psi_{\varepsilon }\left(t\right)\rangle ,|\psi_{\varepsilon \pm \omega }\left(t\right)\rangle ,|\psi_{\varepsilon \pm 2\omega }\left(t\right)\rangle ,\ldots ,|\psi_{\varepsilon \pm n\omega }\left(t\right)\rangle ,\ldots \}.$$
It is clear that any two subspaces $S_{\varepsilon }$ and $S_{\varepsilon^{\prime }}$ $\left(\varepsilon^{\prime }\neq \varepsilon \right)$ are decoupled under the action of $\hat{H}\left(t\right)$ if Re$\left(\varepsilon -\varepsilon^{\prime }\right)\in \left(-\omega ,\omega \right)$. Moreover, $S_{\varepsilon }$ and $S_{\varepsilon^{\prime }}$ are equivalent if there exists an n∈Z such that $\varepsilon -\varepsilon^{\prime }=n\omega$. Therefore, we can study the dynamics of the system separately in each subspace $S_{\varepsilon }$ for Re(ε)∈[−π/T,π/T), which is usually called the first quasienergy Brillouin zone (BZ). The quasienergy ε is, thus, a conserved quantity due to the discrete-time translational symmetry of $\hat{H}\left(t\right)$, similar to the conserved quasimomentum due to the discrete-space translational symmetry of a static Hamiltonian. In the subspace $S_{\varepsilon }$, a solution of the Schrödinger equation can be written as
(5)$$|\Psi_{\varepsilon }\left(t\right)\rangle =\sum_{n}c\left(\varepsilon +\omega_{n}\right)|\psi_{\varepsilon +\omega_{n}}\left(t\right)\rangle =e^{-i\varepsilon t}\sum_{n}c\left(\varepsilon +\omega_{n}\right)|\varphi \left(r\right)\rangle e^{-i\omega_{n}t},$$
where $\left\{c\left(\varepsilon +\omega_{n}\right)\right\}$ are complex coefficients. It is clear that Equation (Equation 5) possesses a time-periodic component
(6)$$|u_{\varepsilon }\left(t\right)\rangle \equiv \sum_{n}c\left(\varepsilon +\omega_{n}\right)|\varphi \left(r\right)\rangle e^{-i\omega_{n}t}=|u_{\varepsilon }\left(t+T\right)\rangle .$$
Therefore, we can express the general solution $|\Psi_{\varepsilon }\left(t\right)\rangle$ as the product of an oscillating phase factor $e^{-i\varepsilon t}$ and a time-periodic Floquet mode $|u_{\varepsilon }\left(t\right)\rangle$, i.e.,
(7)$$|\Psi_{\varepsilon }\left(t\right)\rangle =e^{-i\varepsilon t}|u_{\varepsilon }\left(t\right)\rangle .$$
It also implies that
(8)$$|\Psi_{\varepsilon }\left(t+T\right)\rangle =e^{-i\varepsilon T}|\Psi_{\varepsilon }\left(t\right)\rangle .$$
The latter equation indicates that the only change in the state $|\Psi_{\varepsilon }\left(t\right)\rangle$ after undergoing a one-period evolution is to pick up an exponential factor $e^{-i\varepsilon T}$. We refer to the set {|Ψε(t)〉|Re(ε)∈[−π/T,π/T)} as Floquet eigenstates of the system. They form an orthonormal and complete basis at each instant of time *t*.

Since the evolution from $|\Psi_{\varepsilon }\left(t\right)\rangle$ to $|\Psi_{\varepsilon }\left(t+T\right)\rangle$ is governed by the Schrödinger equation, we can also express Equation (Equation 8) as
(9)$$\hat{U}\left(t+T,t\right)|\Psi_{\varepsilon }\left(t\right)\rangle =e^{-i\varepsilon T}|\Psi_{\varepsilon }\left(t\right)\rangle ,$$
where $\hat{U}\left(t+T,t\right)=\hat{T}e^{-i\int_{t}^{t+T}\hat{H}\left(t^{\prime }\right)dt^{\prime }}$ is nothing but the Floquet operator (evolution operator over one driving period) of the system. When we are only concerned with the stroboscopic dynamics, the initial time dependence of Equation (Equation 9) is not important. In this case, we can set t=0 in Equation (Equation 9) and express the Floquet eigenvalue equation as
(10)$$\hat{U}|\Psi_{E}\rangle =e^{-iE}|\Psi_{E}\rangle ,$$
where $\hat{U}=\hat{T}e^{-i\int_{0}^{T}\hat{H}\left(t\right)dt}$ and we have introduced E=εT as the dimensionless quasienergy, whose first BZ is given by [−π,π). To sum up, we find that the solution of Equation (Equation 1) with a time-periodic $\hat{H}\left(t\right)$ has the form of Equations (Equation 7) or (Equation 8), where $|\Psi_{\varepsilon }\left(t\right)\rangle$ is an eigenstate of the Floquet operator of the system. For stroboscopic observations, all the Floquet eigenstates can thus be obtained by solving the eigenvalue Equation (Equation 10) of the Floquet operator $\hat{U}$. Due to the completeness of Floquet eigenstate basis $\left\{|\Psi_{E}\rangle \right\}$, we can expand an arbitrary initial state of the system as $|\Psi \left(0\right)\rangle =\sum_{E}c_{E}|\Psi_{E}\rangle .$ The resulting state after an evolution over *n* driving periods is then given by
(11)$$|\Psi \left(t\right)\rangle =\sum_{E}c_{E}e^{-inE}|\Psi_{E}\rangle .$$

Note that for a non-Hermitian $\hat{H}\left(t\right)$, $\hat{U}$ is generally non-unitary and the quasienergy *E* may have a non-vanishing imaginary part. In this case, the real part of *E* still belongs to the range of [−π,π) and our arguments leading to the general solution (Equation 11) can be satisfied.

### 2.2. Floquet Eigenvalue Equation

In the most general situations, we can solve Equation (Equation 10) numerically by the split-operator method [3]. Dividing the evolution periodic *T* into *N* segments with a large enough *N*, we can express the Floquet operator $\hat{U}$ approximately as
(12)$$\hat{U}≃e^{-i\hat{H}\left(\left(N-1\right)\Delta t\right)\Delta t}e^{-i\hat{H}\left(\left(N-2\right)\Delta t\right)\Delta t}\cdots e^{-i\hat{H}\left(\Delta t\right)\Delta t}e^{-i\hat{H}\left(0\right)\Delta t},$$
where Δt=T/N. Over each small time interval Δt, $\hat{H}\left(t\right)$ is approximately time-independent and we can diagonalize it numerically at t=ℓΔt as
(13)$$\hat{H}\left(\ell \Delta t\right)=V_{\ell }D_{\ell }V_{\ell }^{-1}.$$
Each column of $V_{\ell }$ represents a right eigenvector of $\hat{H}\left(\ell \Delta t\right)$. The evolution operator over the one-time interval Δt then takes the form
(14)$$e^{-i\hat{H}\left(\ell \Delta t\right)\Delta t}=V_{\ell }e^{-iD_{\ell }\Delta t}V_{\ell }^{-1},\;\ell =0,1,\ldots ,N-1.$$
The multiplication of all the $V_{\ell }e^{-iD_{\ell }\Delta t}V_{\ell }^{-1}$ terms from right to left for ℓ=0 to N−1 yields Equation (Equation 12), which further converges to the exact Floquet operator in the limit N→∞. We can, thus, numerically solve the Floquet eigenvalue equation by diagonalizing the approximated $\hat{U}$ in Equation (Equation 12). This approach works in principle for systems with any individual or multiple driving frequencies. But it may become time-consuming in practice for certain continuously or slowly driven systems.

When the driving field takes the form of periodic kicking or quenching, the series in Equation (Equation 12) can be greatly simplified and even obtained exactly. Here, we give several examples. Consider a time-periodic Hamiltonian of the form
(15)$$\hat{H}\left(t\right)={\hat{H}}_{0}+\sum_{\ell \in Z}\delta \left(t/T-\ell \right){\hat{H}}_{1},$$
where δ(t/T−ℓ) is the delta function peaked at t=ℓT, i.e., each integer multiple of the driving period. The dynamics over each driving period, thus, constitutes a free evolution part controlled by ${\hat{H}}_{0}$ and a delta kicking force controlled by ${\hat{H}}_{1}$. The quantum kicked rotor is one representative example of such a system [28]. Focusing on the one-period evolution from $t=\ell T-0^{+}$ to $t=\left(\ell +1\right)T-0^{+}$, we find the Floquet operator of $\hat{H}\left(t\right)$ to be
(16)$$\begin{array}{cc} \hat{U}= & e^{-i\int_{\ell T+0^{+}}^{\left(\ell +1\right)T-0^{+}}\hat{H}\left(t\right)dt}e^{-i\int_{\ell T-0^{+}}^{\ell T+0^{+}}\hat{H}\left(t\right)dt}\\ = & e^{-i\int_{\ell T+0^{+}}^{\left(\ell +1\right)T-0^{+}}{\hat{H}}_{0}dt}e^{-i\int_{\ell T-0^{+}}^{\ell T+0^{+}}\delta \left(t/T-\ell \right){\hat{H}}_{1}dt}\\ = & e^{-i{\hat{H}}_{0}T}e^{-i{\hat{H}}_{1}T}. \end{array}$$
Similarly, if there are two kicks separated by a time interval τ within each driving period, the Hamiltonian could take the form of
(17)$$\hat{H}\left(t\right)={\hat{H}}_{0}+\sum_{\ell \in Z}\delta \left(t/T-\tau /T-\ell \right){\hat{H}}_{2}+\sum_{\ell \in Z}\delta \left(t/T-\ell \right){\hat{H}}_{1}.$$
For the evolution from time $t=\ell T-0^{+}$ to $t=\left(\ell +1\right)T-0^{+}$, the Floquet operator now takes the form of
(18)$$\hat{U}=e^{-i{\hat{H}}_{0}\left(T-\tau \right)}e^{-i{\hat{H}}_{2}T}e^{-i{\hat{H}}_{0}\tau }e^{-i{\hat{H}}_{1}T}.$$
One typical example of such a system is the double-kicked quantum rotor [24]. For a periodically quenched Hamiltonian in the form of
(19)$$\hat{H}\left(t\right)=\left\{\begin{array}{cc} {\hat{H}}_{1} & t\in [\ell T,\ell T+T_{1})\\ {\hat{H}}_{2} & t\in [\ell T+T_{1},\ell T+T_{1}+T_{2}) \end{array}\right.,$$
where $T=T_{1}+T_{2}$, we can also directly obtain the corresponding Floquet operator from $t=\ell T-0^{+}$ to $t=\left(\ell +1\right)T-0^{+}$ as
(20)$$\begin{array}{cc} \hat{U}= & e^{-i\int_{\ell T+T_{1}}^{\ell T+T-0^{+}}\hat{H}\left(t\right)dt}e^{-i\int_{\ell T-0^{+}}^{\ell T+T_{1}-0^{+}}\hat{H}\left(t\right)dt}\\ = & e^{-i\int_{\ell T+T_{1}}^{\ell T+T-0^{+}}{\hat{H}}_{2}dt}e^{-i\int_{\ell T-0^{+}}^{\ell T+T_{1}-0^{+}}{\hat{H}}_{1}dt}\\ = & e^{-i{\hat{H}}_{2}T_{2}}e^{-i{\hat{H}}_{1}T_{1}}. \end{array}$$
The discrete-time quantum walk can be viewed as one example of such a periodically quenched system [31]. Time-periodic quenches are also frequently implemented in the study of discrete time crystals [19,20,21]. When $[{\hat{H}}_{1},{\hat{H}}_{2}]\neq 0$, the quenches (or kicks) may effectively generate long-range coupling in the system according to the Baker–Campbell–Hausdorff formula, leading to Floquet phases with large topological invariants and many boundary states. This point will be explicitly demonstrated by the examples discussed in Section 3.

For a continuously driven system, the solution of the Floquet eigenvalue equation may also be obtained approximately in terms of the frequency (Sambe) space formalism [269,270,271]. Inserting the Floquet state in Equation (Equation 7) into the Schrödinger Equation (Equation 1) and reorganizing the terms, we find
(21)$$\left[\hat{H}\left(t\right)-i\frac{\partial }{\partial t}\right]|u_{\varepsilon }\left(t\right)\rangle =\varepsilon |u_{\varepsilon }\left(t\right)\rangle .$$
Using the Fourier expansion of $\hat{H}\left(t\right)$ and $|u_{\varepsilon }\left(t\right)\rangle =\sum_{n}e^{-i\omega_{n}t}|u_{\varepsilon }\left(\omega_{n}\right)\rangle$, we further obtain
(22)$$\sum_{m,n}\hat{H}\left(\omega_{m}\right)e^{i(\omega_{m}-\omega_{n})t}|u_{\varepsilon }\left(\omega_{n}\right)\rangle -\sum_{n}\omega_{n}e^{-i\omega_{n}t}|u_{\varepsilon }\left(\omega_{n}\right)\rangle =\varepsilon \sum_{n}e^{-i\omega_{n}t}|u_{\varepsilon }\left(\omega_{n}\right)\rangle .$$
Multiplying $\frac{1}{T}e^{i\omega_{\ell }t}$ from the left on both sides of Equation (Equation 22) and performing the integral over a driving period *T*, we arrive at
(23)$$\sum_{m}\left({\hat{H}}_{m-n}-\omega_{m}\delta_{m,n}\right)|u_{\varepsilon }\left(\omega_{m}\right)\rangle =\varepsilon |u_{\varepsilon }\left(\omega_{n}\right)\rangle ,$$
where
(24)$${\hat{H}}_{m-n}\equiv \hat{H}\left(\omega_{m-n}\right)=\frac{1}{T}\int_{0}^{T}e^{-i\omega_{m-n}t}\hat{H}\left(t\right)dt,$$
and $\omega_{m-n}=\left(m-n\right)\omega =\left(m-n\right)\frac{2\pi }{T}$. Equation (Equation 23) is an infinite-dimensional matrix equation of the form
(25)$$\left[\begin{array}{ccccccc} \ddots & \ddots & \ddots & \ddots & \ddots & \ddots & \ddots \\ \ddots & {\hat{H}}_{0}+2\omega & {\hat{H}}_{1} & {\hat{H}}_{2} & {\hat{H}}_{3} & {\hat{H}}_{4} & \ddots \\ \ddots & {\hat{H}}_{-1} & {\hat{H}}_{0}+\omega & {\hat{H}}_{1} & {\hat{H}}_{2} & {\hat{H}}_{3} & \ddots \\ \ddots & {\hat{H}}_{-2} & {\hat{H}}_{-1} & {\hat{H}}_{0} & {\hat{H}}_{1} & {\hat{H}}_{2} & \ddots \\ \ddots & {\hat{H}}_{-3} & {\hat{H}}_{-2} & {\hat{H}}_{-1} & {\hat{H}}_{0}-\omega & {\hat{H}}_{1} & \ddots \\ \ddots & {\hat{H}}_{-4} & {\hat{H}}_{-3} & {\hat{H}}_{-2} & {\hat{H}}_{-1} & {\hat{H}}_{0}-2\omega & \ddots \\ \ddots & \ddots & \ddots & \ddots & \ddots & \ddots & \ddots \end{array}\right]\left[\begin{array}{c} \vdots \\ |u_{\varepsilon }\left(\omega_{-2}\right)\rangle \\ |u_{\varepsilon }\left(\omega_{-1}\right)\rangle \\ |u_{\varepsilon }\left(\omega_{0}\right)\rangle \\ |u_{\varepsilon }\left(\omega_{1}\right)\rangle \\ |u_{\varepsilon }\left(\omega_{2}\right)\rangle \\ \vdots \end{array}\right]=\varepsilon \left[\begin{array}{c} \vdots \\ |u_{\varepsilon }\left(\omega_{-2}\right)\rangle \\ |u_{\varepsilon }\left(\omega_{-1}\right)\rangle \\ |u_{\varepsilon }\left(\omega_{0}\right)\rangle \\ |u_{\varepsilon }\left(\omega_{1}\right)\rangle \\ |u_{\varepsilon }\left(\omega_{2}\right)\rangle \\ \vdots \end{array}\right].$$
Note that each ${\hat{H}}_{m-n}$ has the same Hilbert space dimension *d* as the original Hamiltonian $\hat{H}\left(t\right)$. As an example, for a harmonically driven system described by the Hamiltonian
(26)$$\hat{H}\left(t\right)={\hat{H}}_{0}+\hat{V}e^{i\omega t}+\hat{W}e^{-i\omega t},$$
the above equation reduces to the following block tridiagonal form
(27)$$\left[\begin{array}{ccccccc} \ddots & \ddots & \ddots & \ddots & \ddots & \ddots & \ddots \\ \ddots & {\hat{H}}_{0}+2\omega & \hat{V} & 0 & 0 & 0 & \ddots \\ \ddots & \hat{W} & {\hat{H}}_{0}+\omega & \hat{V} & 0 & 0 & \ddots \\ \ddots & 0 & \hat{W} & {\hat{H}}_{0} & \hat{V} & 0 & \ddots \\ \ddots & 0 & 0 & \hat{W} & {\hat{H}}_{0}-\omega & \hat{V} & \ddots \\ \ddots & 0 & 0 & 0 & \hat{W} & {\hat{H}}_{0}-2\omega & \ddots \\ \ddots & \ddots & \ddots & \ddots & \ddots & \ddots & \ddots \end{array}\right]\left[\begin{array}{c} \vdots \\ |u_{\varepsilon }\left(\omega_{-2}\right)\rangle \\ |u_{\varepsilon }\left(\omega_{-1}\right)\rangle \\ |u_{\varepsilon }\left(\omega_{0}\right)\rangle \\ |u_{\varepsilon }\left(\omega_{1}\right)\rangle \\ |u_{\varepsilon }\left(\omega_{2}\right)\rangle \\ \vdots \end{array}\right]=\varepsilon \left[\begin{array}{c} \vdots \\ |u_{\varepsilon }\left(\omega_{-2}\right)\rangle \\ |u_{\varepsilon }\left(\omega_{-1}\right)\rangle \\ |u_{\varepsilon }\left(\omega_{0}\right)\rangle \\ |u_{\varepsilon }\left(\omega_{1}\right)\rangle \\ |u_{\varepsilon }\left(\omega_{2}\right)\rangle \\ \vdots \end{array}\right].$$

In practical calculations, one should truncate the infinite-dimensional matrix in Equation (Equation 25) at a sufficiently high harmonics Nω, leading to an Nd×Nd dimensional Floquet effective Hamiltonian, whose eigenvalue problem can be numerically solved. Assuming the characteristic energy scale of ${\hat{H}}_{m-n}$ for all m−n to be Ω, we can take a smaller *N* to achieve the truncation for a larger ratio of ω/Ω, i.e., for a high-frequency driving field. Instead, for a resonantly or slowly driven system, more harmonics should be kept during the truncation. All the discussions presented in this subsection hold for both Hermitian and non-Hermitian Hamiltonians $\hat{H}\left(t\right)$.

### 2.3. Floquet Effective Hamiltonian and High-Frequency Expansion

From the Floquet operator $\hat{U}$ of a periodically driven system, one can formally define its Floquet effective Hamiltonian as
(28)$${\hat{H}}_{\mathrm{eff}}=\frac{i}{T}\ln\hat{U}\Leftrightarrow \hat{U}=e^{-i{\hat{H}}_{\mathrm{eff}}T}.$$

Taking into account the fact that the quasienergies are defined modulus 2π/T, the ${\hat{H}}_{\mathrm{eff}}$ contains the same physical information as $\hat{U}$. Yet, it provides us with more room to treat the properties of Floquet systems in analogy with static Hamiltonian models. For a continuously driven system, the explicit form of ${\hat{H}}_{\mathrm{eff}}$ is usually involved. This can be inspected from Equation (Equation 12), as the $\hat{H}\left(\ell \Delta t\right)$ and $\hat{H}\left(m\Delta t\right)$ do not commute for ℓ≠m in general. When the frequency of the driving field is high enough, an approximate series expression for ${\hat{H}}_{\mathrm{eff}}$ can be obtained via high-frequency expansion methods [61,62,63,64]. Here, we recap one such method in its full generality, which is applicable to both Hermitian and non-Hermitian systems.

We first assume that the time-periodic Hamiltonian $\hat{H}\left(t\right)$ can be decomposed into a static part ${\hat{H}}_{0}$ and a periodically modulated part $\hat{V}\left(t\right)$, i.e.,
(29)$$\hat{H}\left(t\right)={\hat{H}}_{0}+\hat{V}\left(t\right),\;\hat{V}\left(t\right)=\hat{V}\left(t+T\right).$$
Here, T=2π/ω is the driving period, with ω being the driving frequency. Next, we apply a similarity transformation to the evolved state |Ψ(t)〉 in the Schrödinger Equation (Equation 1), yielding a rotated state
(30)$$|\Phi \left(t\right)\rangle =\hat{U}\left(t\right)|\Psi \left(t\right)\rangle =e^{i\hat{K}\left(t\right)}|\Psi \left(t\right)\rangle .$$
Here $\hat{K}\left(t\right)$ is sometimes called the kick operator. It encodes the information regarding the micromotion dynamics of the system. Plugging $|\Psi \left(t\right)\rangle =e^{-i\hat{K}\left(t\right)}|\Phi \left(t\right)\rangle$ into Equation (Equation 1) leads to the transformed Schrödinger equation
(31)$$i\frac{\partial }{\partial t}|\Phi \left(t\right)\rangle ={\hat{H}}_{\mathrm{eff}}|\Phi \left(t\right)\rangle ,$$
where
(32)$${\hat{H}}_{\mathrm{eff}}=e^{i\hat{K}\left(t\right)}\hat{H}\left(t\right)e^{-i\hat{K}\left(t\right)}+i\frac{de^{i\hat{K}\left(t\right)}}{dt}e^{-i\hat{K}\left(t\right)}.$$
The aim of the high-frequency expansion method is to find a time-independent ${\hat{H}}_{\mathrm{eff}}$ by transferring all time-dependent terms into the kick operator $\hat{K}\left(t\right)$. When such a purpose is formally achieved, we can express the Floquet evolution of a system as
(33)$$\hat{U}\left(t_{1},t_{0}\right)|\Psi \left(t_{0}\right)\rangle =e^{-i\hat{K}\left(t_{1}\right)}e^{-i{\hat{H}}_{\mathrm{eff}}\left(t_{1}-t_{0}\right)}e^{i\hat{K}\left(t_{0}\right)}|\Psi \left(t_{0}\right)\rangle .$$
That is, the system is subject to an initial kick associated with the operator $\hat{K}\left(t_{0}\right)$, then evolved under the time-independent ${\hat{H}}_{\mathrm{eff}}$, and finally subject to a second kick carried out by the operator $\hat{K}\left(t_{1}\right)$. There is no need to perform any time-ordered integral in the calculation of $\hat{U}\left(t_{1},t_{0}\right)$.

Assuming the driving frequency ω to be large, we may expand ${\hat{H}}_{\mathrm{eff}}$ and $\hat{K}\left(t\right)$ into power series of 1/ω as
(34)$${\hat{H}}_{\mathrm{eff}}=\sum_{n=0}^{\infty }\frac{1}{\omega^{n}}{\hat{H}}_{\mathrm{eff}}^{\left(n\right)},\;\hat{K}=\sum_{n=1}^{\infty }\frac{1}{\omega^{n}}{\hat{K}}^{\left(n\right)}.$$
In the meantime, we can apply Taylor expansions to the two terms in Equation (Equation 32) and express ${\hat{H}}_{\mathrm{eff}}$ as
(35)$$\begin{array}{cc} {\hat{H}}_{\mathrm{eff}}= & \sum_{n=0}^{\infty }\frac{i^{n}}{n!}\left[\overset{n\;\mathrm{of}\;\hat{K}}{\overset{︷}{\hat{K},[\hat{K},\ldots ,\hat{K},[\hat{K}},\;}\hat{H}]]\right]\\ + & \sum_{n=1}^{\infty }i\frac{i^{n}}{n!}\left[\overset{\left(n-1\right)\;\mathrm{of}\;\hat{K}}{\overset{︷}{\hat{K},[\hat{K},\ldots ,\hat{K},[\hat{K}},\;}\partial_{t}\hat{K}]]\right]. \end{array}$$
Note that we have concealed the explicit time-dependence of $\hat{K}\left(t\right)$ and ${\hat{K}}^{\left(n\right)}\left(t\right)$ in the above two equations for brevity. To proceed, we impose two further requirements for $\hat{K}\left(t\right)$,
(36)$$\hat{K}\left(t\right)=\hat{K}\left(t+T\right),\;\int_{0}^{T}\hat{K}\left(t\right)dt=0.$$
Inserting Equation (Equation 34) and the Fourier expansion
(37)$$\hat{H}\left(t\right)={\hat{H}}_{0}+\sum_{m\neq 0}{\hat{V}}_{m}e^{im\omega t}$$
into Equation (Equation 35), we arrive at
(38)$$\begin{array}{cc} & \sum_{n=0}^{\infty }\frac{1}{\omega^{n}}{\hat{H}}_{\mathrm{eff}}^{\left(n\right)}\\ = & \sum_{n=0}^{\infty }\frac{i^{n}}{n!}\left[\overset{n\;\mathrm{of}\;\hat{K}}{\overset{︷}{\sum_{p}\frac{1}{\omega^{p}}{\hat{K}}^{\left(p\right)},[\sum_{q}\frac{1}{\omega^{q}}{\hat{K}}^{\left(q\right)},\ldots ,\sum_{r}\frac{1}{\omega^{r}}{\hat{K}}^{\left(r\right)},[\sum_{s}\frac{1}{\omega^{s}}{\hat{K}}^{\left(s\right)}},}\left({\hat{H}}_{0}+\sum_{m\neq 0}{\hat{V}}_{m}e^{im\omega t}\right)]]\right]\\ + & \sum_{n=1}^{\infty }\frac{i^{n+1}}{n!}\left[\overset{\left(n-1\right)\;\mathrm{of}\;\hat{K}}{\overset{︷}{\sum_{p}\frac{1}{\omega^{p}}{\hat{K}}^{\left(p\right)},[\sum_{q}\frac{1}{\omega^{q}}{\hat{K}}^{\left(q\right)},\ldots ,\sum_{r}\frac{1}{\omega^{r}}{\hat{K}}^{\left(r\right)},[\sum_{s}\frac{1}{\omega^{s}}{\hat{K}}^{\left(s\right)}},}\;\partial_{t}\sum_{m}\frac{1}{\omega^{m}}{\hat{K}}^{\left(m\right)}]]\right]. \end{array}$$
The high-frequency expansions of ${\hat{H}}_{\mathrm{eff}}$ and $\hat{K}$ are obtained by equating both sides of Equation (Equation 38) at each order of 1/ω.

We can now carry out the calculations order-by-order to find the first few terms in the series of ${\hat{H}}_{\mathrm{eff}}$ and $\hat{K}$. At the zeroth order, we could obtain
(39)$${\hat{H}}_{\mathrm{eff}}^{\left(0\right)}={\hat{H}}_{0}+\hat{V}\left(t\right)-\frac{1}{\omega }\partial_{t}{\hat{K}}^{\left(1\right)}$$
from Equation (Equation 38). To ensure that ${\hat{H}}_{\mathrm{eff}}^{\left(0\right)}$ is time-independent, we must choose
(40)$$\frac{1}{\omega }\partial_{t}{\hat{K}}^{\left(1\right)}=\hat{V}\left(t\right)=\sum_{m\neq 0}{\hat{V}}_{m}e^{im\omega t},$$
such that
(41)$${\hat{K}}^{\left(1\right)}=\sum_{m\neq 0}\frac{1}{im}{\hat{V}}_{m}e^{im\omega t}.$$

Therefore, up to the zeroth order of 1/ω, we have the following approximation for the effective Hamiltonian
(42)$${\hat{H}}_{\mathrm{eff}}\approx {\hat{H}}_{\mathrm{eff}}^{\left(0\right)}={\hat{H}}_{0}.$$
Up to the first order of 1/ω, we have the following approximation for the kick operator
(43)$$\hat{K}\left(t\right)\approx \frac{1}{\omega }{\hat{K}}^{\left(1\right)}=\frac{1}{\omega }\sum_{m\neq 0}\frac{{\hat{V}}_{m}}{im}e^{im\omega t}.$$
At the first order, we could find from Equation (Equation 38) that
(44)$$\begin{array}{cc} {\hat{H}}_{\mathrm{eff}}^{\left(1\right)} & =i\left[{\hat{K}}^{\left(1\right)},{\hat{H}}_{0}+\hat{V}\left(t\right)\right]-\frac{1}{\omega }\partial_{t}{\hat{K}}^{\left(2\right)}-\frac{i}{2}\left[{\hat{K}}^{\left(1\right)},\frac{1}{\omega }\partial_{t}{\hat{K}}^{\left(1\right)}\right]\\ & =i\left[{\hat{K}}^{\left(1\right)},{\hat{H}}_{0}+\hat{V}\left(t\right)\right]-\frac{1}{\omega }\partial_{t}{\hat{K}}^{\left(2\right)}-\frac{i}{2}\left[{\hat{K}}^{\left(1\right)},\hat{V}\left(t\right)\right]\\ & =i\left[{\hat{K}}^{\left(1\right)},{\hat{H}}_{0}\right]+\frac{i}{2}\left[{\hat{K}}^{\left(1\right)},\hat{V}\left(t\right)\right]-\frac{1}{\omega }\partial_{t}{\hat{K}}^{\left(2\right)}. \end{array}$$

By definition, any non-vanishing elements of $i\left[{\hat{K}}^{\left(1\right)},{\hat{H}}_{0}\right]$ and $-\frac{1}{\omega }\partial_{t}{\hat{K}}^{\left(2\right)}$ are time-dependent. The only time-independent term that could contribute to ${\hat{H}}_{\mathrm{eff}}^{\left(1\right)}$ comes from the term $\frac{i}{2}\left[{\hat{K}}^{\left(1\right)},\hat{V}\left(t\right)\right]$, which can be written explicitly as
(45)$$\begin{array}{cc} \frac{i}{2}\left[{\hat{K}}^{\left(1\right)},\hat{V}\left(t\right)\right] & =\frac{i}{2}\left[\sum_{m\neq 0}\frac{1}{im}{\hat{V}}_{m}e^{im\omega t},\sum_{m^{\prime }\neq 0}{\hat{V}}_{m^{\prime }}e^{im^{\prime }\omega t}\right]\\ & =\sum_{m,m^{\prime }\neq 0}\frac{e^{i(m+m^{\prime })\omega t}}{2m}\left[{\hat{V}}_{m},{\hat{V}}_{m^{\prime }}\right]. \end{array}$$
The time-independent terms are those with $m^{\prime }=-m$. Collecting these terms together, we find
(46)$${\hat{H}}_{\mathrm{eff}}^{\left(1\right)}=\sum_{m\neq 0}\frac{\left[{\hat{V}}_{m},{\hat{V}}_{-m}\right]}{2m}=\sum_{m\neq 0}\frac{{\hat{V}}_{m}{\hat{V}}_{-m}}{m}.$$
Therefore, up to the first order of 1/ω, we have the following approximation for the effective Hamiltonian
(47)$${\hat{H}}_{\mathrm{eff}}\approx {\hat{H}}_{\mathrm{eff}}^{\left(0\right)}+\frac{1}{\omega }{\hat{H}}_{\mathrm{eff}}^{\left(1\right)}={\hat{H}}_{0}+\frac{1}{\omega }\sum_{m\neq 0}\frac{{\hat{V}}_{m}{\hat{V}}_{-m}}{m}.$$
The form of ${\hat{K}}^{\left(2\right)}$ is determined by the remaining time-dependent terms. Performing the integration over time directly, we obtain
(48)$$\begin{array}{cc} {\hat{K}}^{\left(2\right)}= & \sum_{m\neq 0}\frac{1}{m}\left[{\hat{V}}_{m},{\hat{H}}_{0}\right]\int \omega e^{im\omega t}dt+\sum_{m,m^{\prime }\neq 0,-m}\frac{\left[{\hat{V}}_{m},{\hat{V}}_{m^{\prime }}\right]}{2m}\int \omega e^{i(m+m^{\prime })\omega t}dt\\ = & \sum_{m\neq 0}\frac{1}{im^{2}}\left[{\hat{V}}_{m},{\hat{H}}_{0}\right]e^{im\omega t}+\sum_{m,m^{\prime }\neq 0,-m}\frac{\left[{\hat{V}}_{m},{\hat{V}}_{m^{\prime }}\right]}{2im(m+m^{\prime })}e^{i(m+m^{\prime })\omega t}. \end{array}$$
Up to the second order of 1/ω, we can thus approximate the kick operator by
(49)$$\begin{array}{cc} \hat{K}\left(t\right) & \approx \frac{1}{\omega }{\hat{K}}^{\left(1\right)}+\frac{1}{\omega^{2}}{\hat{K}}^{\left(2\right)}\\ & =\frac{1}{\omega }\sum_{m\neq 0}\frac{{\hat{V}}_{m}}{im}e^{im\omega t}+\frac{1}{\omega^{2}}\sum_{m\neq 0}\frac{1}{im^{2}}\left[{\hat{V}}_{m},{\hat{H}}_{0}\right]e^{im\omega t}\\ & +\frac{1}{\omega^{2}}\sum_{m,m^{\prime }\neq 0,-m}\frac{\left[{\hat{V}}_{m},{\hat{V}}_{m^{\prime }}\right]}{2im(m+m^{\prime })}e^{i(m+m^{\prime })\omega t}. \end{array}$$
We could continue these self-consistent calculations to obtain higher-order terms in the high-frequency expansion of ${\hat{H}}_{\mathrm{eff}}$ and $\hat{K}\left(t\right)$. For example, for the second-order component ${\hat{H}}_{\mathrm{eff}}^{\left(2\right)}$, we have
(50)$$\begin{array}{cc} {\hat{H}}_{\mathrm{eff}}^{\left(2\right)}= & \frac{-1}{2}\left[{\hat{K}}^{\left(1\right)},\left[{\hat{K}}^{\left(1\right)},{\hat{H}}_{0}+\hat{V}\left(t\right)\right]\right]+i\left[{\hat{K}}^{\left(2\right)},{\hat{H}}_{0}+\hat{V}\left(t\right)\right]\\ + & \frac{-1}{\omega }\partial_{t}{\hat{K}}^{\left(3\right)}+\frac{-i}{2\omega }\left[{\hat{K}}^{\left(1\right)},\partial_{t}{\hat{K}}^{\left(2\right)}\right]+\frac{-i}{2\omega }\left[{\hat{K}}^{\left(2\right)},\partial_{t}{\hat{K}}^{\left(1\right)}\right]\\ + & \frac{1}{6\omega }[{\hat{K}}^{\left(1\right)},\left[{\hat{K}}^{\left(1\right)},\partial_{t}{\hat{K}}^{\left(1\right)}\right]. \end{array}$$
Dropping the time-dependent terms, we are left with
(51)$${\hat{H}}_{\mathrm{eff}}^{\left(2\right)}=\frac{1}{2}\sum_{m\neq 0}\frac{\left[\left[{\hat{V}}_{m},{\hat{H}}_{0}\right],{\hat{V}}_{-m}\right]}{m^{2}}+\frac{1}{3}\sum_{l,m\neq 0}\frac{\left[{\hat{V}}_{l},\left[{\hat{V}}_{m},{\hat{V}}_{-l-m}\right]\right]}{lm}.$$
Therefore, up to the second-order correction in 1/ω, the effective Hamiltonian ${\hat{H}}_{\mathrm{eff}}$ turns out to be
(52)$$\begin{array}{cc} {\hat{H}}_{\mathrm{eff}} & \approx {\hat{H}}_{0}+\frac{1}{\omega }\sum_{m\neq 0}\frac{{\hat{V}}_{m}{\hat{V}}_{-m}}{m}\\ & +\frac{1}{2\omega^{2}}\sum_{m\neq 0}\frac{\left[\left[{\hat{V}}_{m},{\hat{H}}_{0}\right],{\hat{V}}_{-m}\right]}{m^{2}}+\frac{1}{3\omega^{2}}\sum_{l,m\neq 0}\frac{\left[{\hat{V}}_{l},\left[{\hat{V}}_{m},{\hat{V}}_{-l-m}\right]\right]}{lm}. \end{array}$$

This approximation holds for both Hermitian and non-Hermitian Floquet systems under high-frequency driving fields. Note that for stroboscopic evolution, we have $t_{1}-t_{0}=NT$ with N∈Z and $\hat{K}\left(t_{1}\right)=\hat{K}\left(t_{0}\right)$ in Equation (Equation 33). In this case, the term $\hat{K}\left(t_{0}\right)$ in Equation (Equation 33) describes an initial phase, which can be set to zero as the expansion of $\hat{K}\left(t\right)$ at every order of 1/ω is only determined up to a constant (see Equations (Equation 41) and (Equation 48)). Therefore, for stroboscopic dynamics, we can simply use the ${\hat{H}}_{\mathrm{eff}}$ to capture most of the essential physics. In Section 3, we will showcase the application of the high-frequency methods presented here to the Floquet engineering of non-Hermitian quasicrystals.

To be concrete, we illustrate the usage of ${\hat{H}}_{\mathrm{eff}}$ here with a simple example. Consider a harmonically driven two-level Hamiltonian of the form
(53)$$\hat{H}\left(t\right)=\left(h_{x}+i\gamma /2\right)\left[\cos\left(\omega t\right)\sigma_{x}+\sin\left(\omega t\right)\sigma_{y}\right]+h_{z}\sigma_{z},$$
where $h_{x},h_{z},\gamma \in R$, and $\sigma_{\alpha }$ for α=x,y,z are Pauli matrices. It is not hard to verify that by choosing the kick operator as
(54)$$e^{i\hat{K}\left(t\right)}=\left(\begin{array}{cc} 1 & 0\\ 0 & e^{-i\omega t} \end{array}\right),$$
the Floquet effective Hamiltonian in the rotating frame can be exactly obtained from Equation (Equation 32), i.e.,
(55)$${\hat{H}}_{\mathrm{eff}}=\left(\omega /2\right)\sigma_{0}+\left(h_{x}+i\gamma /2\right)\sigma_{x}+\left(h_{z}-\omega /2\right)\sigma_{z},$$
where $\sigma_{0}$ denotes the 2×2 identity matrix. We see that the two levels of H(t) have the instantaneous eigenenergies
(56)$$E_{\pm }=\pm \sqrt{h_{x}^{2}+h_{z}^{2}-\gamma^{2}/4+ih_{x}\gamma },$$
which are time-independent and complex in general. They could meet with each other at a second-order EP with E=0 when $h_{x}=0$ and $h_{z}=\pm \gamma /2$. Meanwhile, ${\hat{H}}_{\mathrm{eff}}$ in Equation (Equation 55) has two quasienergy levels defined by
(57)$$\varepsilon_{\pm }=\frac{\omega }{2}\pm \sqrt{h_{x}^{2}+{\left(h_{z}-\frac{\omega }{2}\right)}^{2}-\gamma^{2}/4+ih_{x}\gamma }.$$
They could meet with each other at a second-order Floquet EP with ε=ω/2 when $h_{x}=0$ and $h_{z}=\omega /2\pm \gamma /2$. It is clear that the EP is shifted by the driving in both its energy and location in the parameter space. If we define εT as the dimensionless quasienergy *E*, the Floquet EP of our two-level system appears at the quasienergy E=π. This could lead to Floquet exceptional topology and anomalous π edge modes, which are unique to driven non-Hermitian systems. We will give an explicit example to demonstrate this point in Section 3.

### 2.4. Adiabatic Perturbation Theory

We now consider a non-Hermitian system that is subject to slow-in-time and cyclic modulations. It can be viewed as the opposite limit to a high-frequency driven system. Introducing a re-scaled and dimensionless time variable s=t/T, we can express the Schrödinger Equation (Equation 1) as
(58)$$i\frac{\partial }{\partial s}|\Psi \left(s\right)\rangle =T\hat{H}\left(s\right)|\Psi \left(s\right)\rangle .$$
Our purpose is to find an expansion for |Ψ(s)〉 in the series of 1/T. This constitutes an adiabatic perturbation theory (APT) of the system [272], providing that different energy levels of $\hat{H}\left(s\right)$ are gapped for all s∈[0,1].

The Hamiltonian $\hat{H}\left(s\right)$ is periodic in the scaled time *s* with $\hat{H}\left(s\right)=\hat{H}\left(s+1\right)$. We denote its instantaneous right and left biorthonormal eigenvectors as {|n(s)〉} and $\left\{|\tilde{n}\left(s\right)\rangle \right\}$ [273], such that
(59)$$\hat{H}\left(s\right)|n\left(s\right)\rangle =E_{n}\left(s\right)|n\left(s\right)\rangle ,\;\langle \tilde{n}\left(s\right)|\hat{H}\left(s\right)=\langle \tilde{n}\left(s\right)|E_{n}\left(s\right),$$
(60)$$\langle \tilde{m}\left(s\right)|n\left(s\right)\rangle =\delta_{mn},\;\sum_{n}|n\left(s\right)\rangle \langle \tilde{n}\left(s\right)|=1.$$
Here, $E_{n}\left(s\right)$ is the instantaneous eigenenergy associated with |n(s)〉, which could be complex if $\hat{H}\left(s\right)\neq {\hat{H}}^{†}\left(s\right)$. To proceed, we introduce an ansatz solution for the time-evolved state |Ψ(s)〉 as
(61)$$|\Psi \left(s\right)\rangle =\sum_{n}e^{-i\Omega_{n}\left(s\right)}c_{n}\left(s\right)|n\left(s\right)\rangle ,$$
where
(62)$$\Omega_{n}\left(s\right)\equiv T\int_{0}^{s}E_{n}\left(s^{\prime }\right)ds^{\prime }$$
describes a dynamical phase accumulated over the time interval [0,s]. At s=0, we have $\Omega_{n}\left(0\right)=0$ and
(63)$$c_{n}\left(0\right)=\langle \tilde{n}\left(0\right)|\Psi \left(0\right)\rangle$$
for the initial state |Ψ(0)〉. Plugging Equation (Equation 61) into Equation (Equation 58), we obtain
(64)$$\sum_{n}e^{-i\Omega_{n}\left(s\right)}{\dot{c}}_{n}\left(s\right)|n\left(s\right)\rangle =-\sum_{n}e^{-i\Omega_{n}\left(s\right)}c_{n}\left(s\right)|\dot{n}\left(s\right)\rangle ,$$
where ${\dot{c}}_{n}\left(s\right)=dc_{n}\left(s\right)/ds$ and $|\dot{n}\left(s\right)\rangle =d|n\left(s\right)\rangle /ds$. Acting $\langle \tilde{m}\left(s\right)|$ from the left on both sides of the above equation and using Equation (Equation 60), we find
(65)$${\dot{c}}_{m}\left(s\right)=-\sum_{n\neq m}e^{i\Omega_{mn}\left(s\right)}c_{n}\left(s\right)\langle \tilde{m}\left(s\right)|\dot{n}\left(s\right)\rangle ,$$
where $\Omega_{mn}\left(s\right)\equiv \Omega_{m}\left(s\right)-\Omega_{n}\left(s\right)$. We have made the parallel transport gauge choice so that $\langle \tilde{n}\left(s\right)|\dot{n}\left(s\right)\rangle =0$. Performing the integration over *s* and keeping the terms on the right-hand side up to the first order of 1/T (i.e., up to the first-order non-adiabatic correction), we arrive at
(66)$$c_{m}\left(s\right)=c_{m}\left(0\right)+\frac{1}{T}\sum_{n\neq m}{\left.\left[\frac{i\langle \tilde{m}\left(s^{\prime }\right)|\dot{n}\left(s^{\prime }\right)\rangle }{\Delta_{mn}\left(s^{\prime }\right)}e^{i\Omega_{mn}\left(s^{\prime }\right)}\right]\right|}_{s^{\prime }=0}^{s^{\prime }=s}c_{n}\left(0\right),$$
where $\Delta_{mn}\left(s\right)=E_{m}\left(s\right)-E_{n}\left(s\right)$. To reach Equation (Equation 66), we have assumed $\Omega_{mn}\left(s\right)$ to be real for all m≠n and s∈[0,1]. This can be achieved if every instantaneous energy level possesses the same imaginary part iγ(s). On the other hand, Equation (Equation 66) is valid if the $\hat{H}\left(s\right)$ is PT-invariant at every *s*, such that $E_{n}\left(s\right)\in R$ for all *n*. The approximation in Equation (Equation 66) does not hold if $\Delta_{mn}\left(s\right)\notin R$, which would cause the exponential amplification or decay of the amplitude $c_{n}\left(s\right)$. So the APT developed here is more restrictive in application than its Hermitian counterparts. Inserting Equation (Equation 66) into Equation (Equation 61) results in the solution to Equation (Equation 58) up to first-order non-adiabatic corrections, i.e.,
(67)$$|\Psi \left(s\right)\rangle =\sum_{m}e^{-i\Omega_{m}\left(s\right)}\left\{c_{m}\left(0\right)+\frac{1}{T}\sum_{n\neq m}{\left.\left[\frac{i\langle \tilde{m}\left(s^{\prime }\right)|\dot{n}\left(s^{\prime }\right)\rangle }{\Delta_{mn}\left(s^{\prime }\right)}e^{i\Omega_{mn}\left(s^{\prime }\right)}\right]\right|}_{s^{\prime }=0}^{s^{\prime }=s}c_{n}\left(0\right)\right\}|m\left(s\right)\rangle .$$
Since we have employed the formalism of biorthonormal eigenvectors, we may consider another ansatz solution for the left eigenvector
(68)$$\langle \tilde{\Psi }\left(s\right)|=\sum_{n}e^{i\Omega_{n}\left(s\right)}c_{n}^{*}\left(s\right)\langle \tilde{n}\left(s\right)|,$$
which obeys a conjugate Schrödinger equation
(69)$$-i\frac{\partial }{\partial s}\langle \tilde{\Psi }\left(s\right)|=\;\langle \tilde{\Psi }\left(s\right)|\hat{H}.$$
Following the same reasoning in deriving Equation (Equation 67), we find, up to the first order of 1/T, that
(70)$$\langle \tilde{\Psi }\left(s\right)|=\sum_{m}e^{+i\Omega_{m}\left(s\right)}\left\{c_{m}^{*}\left(0\right)+\frac{1}{T}\sum_{n\neq m}{\left[\frac{i\langle \dot{\tilde{n}}\left(s^{\prime }\right)|m\left(s^{\prime }\right)\rangle }{\Delta_{nm}\left(s^{\prime }\right)}e^{i\Omega_{nm}\left(s^{\prime }\right)}\right]}_{s^{\prime }=0}^{s^{\prime }=s}c_{n}^{*}\left(0\right)\right\}\langle \tilde{m}\left(s\right)|.$$
Assuming that initially only the state with m=ℓ is occupied such that $c_{m}\left(0\right)=\delta_{m\ell }$, we can further simplify Equations (Equation 67) and (Equation 70) to
(71)$$|\Psi_{\ell }\left(s\right)\rangle =e^{-i\Omega_{\ell }\left(s\right)}\left[|\ell \left(s\right)\rangle +\frac{1}{T}\sum_{m\neq \ell }\frac{i\langle \tilde{m}\left(s\right)|\dot{\ell }\left(s\right)\rangle }{\Delta_{m\ell }\left(s\right)}|m\left(s\right)\rangle \right]-\frac{1}{T}\sum_{m\neq \ell }e^{-i\Omega_{m}\left(s\right)}\frac{i\langle \tilde{m}\left(0\right)|\dot{\ell }\left(0\right)\rangle }{\Delta_{m\ell }\left(0\right)}|m\left(s\right)\rangle ,$$
(72)$$\langle {\tilde{\Psi }}_{\ell }\left(s\right)|=e^{+i\Omega_{\ell }\left(s\right)}\left[\langle \tilde{\ell }\left(s\right)|+\frac{1}{T}\sum_{m\neq \ell }\frac{i\langle \dot{\tilde{\ell }}\left(s\right)|m\left(s\right)\rangle }{\Delta_{\ell m}\left(s\right)}\langle \tilde{m}\left(s\right)|\right]-\frac{1}{T}\sum_{m\neq \ell }e^{+i\Omega_{m}\left(s\right)}\frac{i\langle \dot{\tilde{\ell }}\left(0\right)|m\left(0\right)\rangle }{\Delta_{\ell m}\left(0\right)}\langle \tilde{m}\left(s\right)|.$$
For any observable $\hat{O}$, up to the correction of order 1/T, we can now express its biorthogonal average over the states $\{|\Psi_{\ell }\left(s\right)\rangle ,|{\tilde{\Psi }}_{\ell }\left(s\right)\rangle \}$ as
(73)$$\begin{array}{cc} \langle {\tilde{\Psi }}_{\ell }\left(s\right)|\hat{O}|\Psi_{\ell }\left(s\right)\rangle & =\langle \tilde{\ell }\left(s\right)|\hat{O}|\ell \left(s\right)\rangle +\frac{1}{T}\sum_{m\neq \ell }\frac{i\langle \tilde{m}\left(s\right)|\dot{\ell }\left(s\right)\rangle }{\Delta_{m\ell }\left(s\right)}\langle \tilde{\ell }\left(s\right)|\hat{O}|m\left(s\right)\rangle \\ & +\frac{1}{T}\sum_{m\neq \ell }\frac{i\langle \dot{\tilde{\ell }}\left(s\right)|m\left(s\right)\rangle }{\Delta_{\ell m}\left(s\right)}\langle \tilde{m}\left(s\right)|\hat{O}|\ell \left(s\right)\rangle . \end{array}$$
We notice that this average does not contain any time-oscillating phase factors.

We now illustrate this APT with an application in the study of dynamical topological phenomena. Let us consider noninteracting particles in a one-dimensional (1D) periodic lattice, whose onsite potential is also varied slowly and periodically in time. This is the typical situation encountered in the topological Thouless pump [16,17,18]. The group velocity of the particle can be expressed as $\hat{v}=\partial_{k}\hat{H}$, where *k* is the quasimomentum. At the initial time s=0, we assume that the band *ℓ* is uniformly filled, and it is separated from the other bands at all *k* and *s*. The pumped number of particles over one adiabatic cycle due to this initially filled band is then given by
(74)$$N_{\ell }=\int_{-\pi }^{\pi }\frac{dk}{2\pi }\int_{0}^{1}ds\langle {\tilde{\Psi }}_{\ell }\left(s\right)|\partial_{k}\hat{H}|\Psi_{\ell }\left(s\right)\rangle .$$
With the aid of Equation (Equation 59), it is not hard to identify that
(75)$$\langle \tilde{\ell }\left(s\right)|\partial_{k}\hat{H}|\ell \left(s\right)\rangle =\partial_{k}E_{\ell }\left(k,s\right),$$
(76)$$\frac{\langle \tilde{\ell }\left(s\right)|\partial_{k}\hat{H}|m\left(s\right)\rangle }{\Delta_{m\ell }\left(s\right)}=\langle \tilde{\ell }\left(s\right)|\partial_{k}|m\left(s\right)\rangle ,\;m\neq \ell ,$$
where $E_{\ell }\left(k,s\right)$ denotes the energy dispersion of the *ℓ*th adiabatic Bloch band. Plugging Equations (Equation 75), (Equation 76), and (Equation 73) into Equation (Equation 74), we obtain
(77)$$\begin{array}{cc} N_{\ell }= & \int_{-\pi }^{\pi }\frac{dk}{2\pi }\int_{0}^{1}ds\partial_{k}E_{\ell }\left(k,s\right)\\ + & \sum_{m\neq \ell }\int_{-\pi }^{\pi }\frac{dk}{2\pi i}\int_{0}^{T}dt\langle \partial_{k}\tilde{\ell }\left(k,t\right)|m\left(k,t\right)\rangle \langle \tilde{m}\left(k,t\right)|\partial_{t}\ell \left(k,t\right)\rangle \\ - & \sum_{m\neq \ell }\int_{-\pi }^{\pi }\frac{dk}{2\pi i}\int_{0}^{T}dt\langle \partial_{t}\tilde{\ell }\left(k,t\right)|m\left(k,t\right)\rangle \langle \tilde{m}\left(k,t\right)|\partial_{k}\ell \left(k,t\right)\rangle . \end{array}$$

Noting that $E_{\ell }\left(k=-\pi ,s\right)=E_{\ell }\left(k=\pi ,s\right)$ and $\sum_{m}|m\left(k,t\right)\rangle \langle \tilde{m}\left(k,t\right)|=1$, we can simplify the above equation and arrive at the pumped number of particles over an adiabatic cycle, i.e.,
(78)$$N_{\ell }=\int_{-\pi }^{\pi }\frac{dk}{2\pi i}\int_{0}^{T}dt\left[\langle \partial_{k}\tilde{\ell }\left(k,t\right)|\partial_{t}\ell \left(k,t\right)\rangle -\langle \partial_{t}\tilde{\ell }\left(k,t\right)|\partial_{k}\ell \left(k,t\right)\rangle \right].$$

Equation (Equation 78) describes nothing but the Chern number of the adiabatic Bloch band *ℓ*, whose energy dispersion is defined on a two-dimensional (2D) torus (k,t)∈[−π,π)×[0,T). The conditions for Equation (Equation 78) to hold are as follows. First, the Hamiltonian of the system should be quasi-Hermitian with a real spectrum (e.g., PT-invariant) in our considered parameter regime. Second, the band $E_{\ell }$ should be well gapped from the other bands throughout the 2D torus (k,t)∈[−π,π)×[0,T), and ℏ/T should be much smaller than $|\Delta_{\ell m}|$ for all m≠ℓ in order to guarantee the adiabatic condition. Third, the evolution of the left and right vectors of the system should follow Equations (Equation 58) and (Equation 69). Note here that the expression of the Chern number is not sensitive to the choice of biorthonormal basis. We will obtain the same $N_{\ell }$ after changing $\tilde{\ell }\to \ell$ or $\ell \to \tilde{\ell }$ in Equation (Equation 78), as proved before for non-Hermitian Chern bands [140].

### 2.5. Symmetry and Topological Characterization

Over the past few years, rich symmetry classifications and topological invariants have been identified for non-Hermitian topological matter [140,141,142,143,144,145,146,147,148]. In this subsection, we mainly recap two symmetries together with their associated topological numbers. They are the most relevant ones for the characterization of non-Hermitian Floquet topological phases reviewed in this work.

We first discuss PT symmetry, which is associated with the operator PT. Here, P denotes the parity operator and T denotes the time-reversal operator. When the Hamiltonian of a non-Hermitian system $\hat{H}$ respects PT symmetry, we have $[\mathrm{PT},\hat{H}]=0$. In this case, the system could have a real spectrum in the PT-unbroken regime, where the eigenstates |ψ〉 and PT|ψ〉 of $\hat{H}$ are coincident up to a global phase. To see this, let us consider a non-degenerate eigenstate |ψ〉 of $\hat{H}$ that satisfies the eigenvalue equation
(79)$$\hat{H}|\psi \rangle =E|\psi \rangle .$$
The PT symmetry of $\hat{H}$ then implies that
(80)$$\hat{H}(\mathrm{PT}|\psi \rangle )=E^{*}(\mathrm{PT}|\psi \rangle ).$$
Therefore, PT|ψ〉 is also an eigenstate of $\hat{H}$ with the energy $E^{*}$. If |ψ〉 and $\hat{H}$ share the same PT symmetry, |ψ〉 should be the common eigenstate of $\hat{H}$ and PT. PT|ψ〉 can thus only differ from |ψ〉 up to a global phase, which means that $E=E^{*}\in R$. However, with the change in system parameters (e.g., the strengths of gain and loss), the PT symmetry of |ψ〉 could be spontaneously broken and the spectrum of $\hat{H}$ could switch from real to complex after undergoing a PT-symmetry-breaking transition. A topological invariant, defined as [190]
(81)$$w=\int_{0}^{2\pi }\frac{d\theta }{2\pi i}\partial_{\theta }\ln\det\left[\hat{H}\left(\theta \right)-E_{0}\right],$$
might be employed to characterize such a real-to-complex spectral transition. Here, $E_{0}$ is a base energy chosen appropriately on the complex plane (not belong to the spectrum of $\hat{H}$). The parametrized Hamiltonian $\hat{H}\left(\theta \right)=\hat{H}\left(\theta +2\pi \right)$, where θ can be viewed as the quasimomentum along an artificial dimension. We have also taken the periodic boundary condition (PBC) for $\hat{H}$ before implementing its θ-parametrization. The *w* in Equation (Equation 81), thus, depicts a spectral winding number with respect to the base energy $E_{0}$, i.e., it counts the number of times that the spectrum of $\hat{H}\left(\theta \right)$ winds around $E_{0}$ on the complex plane when the synthetic quasimomentum θ is varied over a cycle. When the spectrum of $\hat{H}\left(\theta \right)$ is real, we must have w=0, as a spectral loop cannot be formed on the complex plane in this case. When the spectrum of $\hat{H}\left(\theta \right)$ is complex, *w* may take an integer-quantized value if $\hat{H}\left(\theta \right)$ possesses spectral loops around $E_{0}$. A suitably chosen $E_{0}$ could then yield a nonzero *w* when the first spectral loop appears on the complex-*E* plane, thereby detecting the topological changes in the spectrum of $\hat{H}$ across the PT-breaking transition.

For a Floquet system, if the time-periodic Hamiltonian $\hat{H}\left(t\right)$ possesses PT symmetry at each instant *t*, the resulting Floquet operator also has PT symmetry. In this case, we can define the spectral winding number *w* as in Equation (Equation 81) for the stroboscopic Floquet effective Hamiltonian so as to capture the PT-breaking transition in the quasienergy spectrum. We will provide explicit examples for this usage in Section 3.4, where we consider PT transitions, localization transitions, and topological transitions in non-Hermitian Floquet quasicrystals.

We next consider the chiral (or sublattice) symmetry, whose associated operator will be denoted by S. When the Hamiltonian $\hat{H}$ of a system respects chiral symmetry S, we have $S\hat{H}S=-\hat{H}$, where S is both Hermitian and unitary [274]. The implication of this symmetry on the spectrum of $\hat{H}$ is as follows. Suppose that |ψ〉 is an eigenstate of a chiral-symmetric $\hat{H}$ with the energy *E*, i.e., $\hat{H}|\psi \rangle =E|\psi \rangle$. We have
(82)$$\hat{H}(S|\psi \rangle )=-E(S|\psi \rangle ).$$
Therefore, S|ψ〉 is also an eigenstate of $\hat{H}$ with the energy −E. The eigenstates of a chiral-symmetric $\hat{H}$ should then come in pairs of {|ψ〉,S|ψ〉} with the energies {E,−E} that are symmetric with respect to E=0. This further leads to a chiral-symmetry-protected degeneracy for any eigenstate with E=0. When the spectrum of $\hat{H}$ is gapped at E=0, we can group its energy levels into two clusters with ReE<0 and ReE>0. If these two clusters meet with each other at E=0 and then separate with the change in certain system parameters, the system may undergo a phase transition. In one dimension, the change in band topology of the system before and after such a transition could be characterized by a winding number $w_{0}$, defined as [274,275,276]
(83)$$w_{0}=\int_{-\pi }^{\pi }\frac{dk}{4\pi }\mathrm{Tr}\left[SQ\left(k\right)i\partial_{k}Q\left(k\right)\right].$$
Here, the PBC has been assumed and k∈[−π,π) denotes the quasimomentum. The sign-resolved projector Q(k) is obtained from the spectral decomposition of $\hat{H}=\sum_{k}\left|k\rangle H\left(k\right)\langle k\right|$ with $H\left(k\right)=\sum_{n}E_{n}\left(k\right)|n\left(k\right)\rangle \langle \tilde{n}\left(k\right)|$ at each *k* by attributing +1 (−1) to every energy band *n* with ReEn>0 (ReEn<0). Q(k) can thus be expressed as [263]
(84)Q(k)=∑nsgn{Re[En(k)]}|n(k)〉〈n˜(k)|.
Here, $H\left(k\right)|n\left(k\right)\rangle =E_{n}\left(k\right)|n\left(k\right)\rangle$ and $\langle \tilde{n}\left(k\right)|H\left(k\right)=\langle \tilde{n}\left(k\right)|E_{n}\left(k\right)$. The set $\{|n\left(k\right)\rangle ,|\tilde{n}\left(k\right)\rangle \}$ of basis satisfies the biorthonormal relations in Equation (Equation 60). Under the PBC, the winding number $w_{0}$ could characterize the bulk topological properties of a 1D chiral-symmetric Hamiltonian $\hat{H}$, either Hermitian or non-Hermitian. It could further distinguish between different bulk topological insulating phases by showing a quantized jump at the transition point, where the two band clusters of $\hat{H}$ meet with each other at E=0. However, under the open boundary condition (OBC), due to the possible existence of NHSE, the $w_{0}$ defined in Equation (Equation 83) may not be able to correctly predict the gap-closing points of the spectrum in the parameter space and determine the number of degenerate edge modes at E=0 in different parameter regions. This non-Hermiticity-induced breakdown of bulk–edge correspondence may be recovered by introducing a real-space counterpart of $w_{0}$, defined as [169]
(85)$$W_{0}=-\frac{1}{L_{B}}{\mathrm{Tr}}_{B}\left(SQ\left[Q,\hat{N}\right]\right).$$
Here S is the chiral symmetry operator of $\hat{H}$, $\hat{N}$ is the position operator in real space, and Q is the flat band projector defined as
(86)Q=∑|ψj〉∈bulksgn[Re(Ej)]|ψj〉〈ψ˜j|,
where the summation is now taken over all the bulk eigenstates $\left\{|\psi_{j}\rangle \right\}$ of $\hat{H}$ under the OBC. The whole lattice of length $L=L_{B}+2L_{E}$ is decomposed into three segments, with a bulk region of length $L_{B}$ in the middle, and two edge regions of the same length $L_{E}$ at the left and right boundaries of the open chain. The trace ${\mathrm{Tr}}_{B}\left(\cdot \right)$ is only taken over the bulk region, which excludes all possible interruptions caused by the NHSE in the edge regions. The resulting $W_{0}$ was found to be able to faithfully capture the topological phase transitions and bulk-edge correspondence in 1D, chiral-symmetric non-Hermitian systems even in the presence of NHSE [169]. It was also suggested to be equivalent to the topological winding number defined through the generalized Brillouin zone of non-Hermitian systems. By definition, the winding number $W_{0}$ in Equation (Equation 85) is also robust to perturbations induced by symmetry-preserved disorders and impurities, making it applicable to more general situations. In the clean, Hermitian, and thermodynamic limits, the $W_{0}$ in Equation (Equation 85) can be further reduced to the $w_{0}$ in Equation (Equation 83) [274,275,276].

For a non-Hermitian Floquet system, we can state its chiral symmetry as follows [256]. From Equation (Equation 28), we can express the Floquet operator as $\hat{U}=e^{-i{\hat{H}}_{\mathrm{eff}}}$, where we have set the driving period T=1 for brevity. Viewing the ${\hat{H}}_{\mathrm{eff}}$ as a static Hamiltonian, we say that it respects the chiral symmetry if there exists a unitary and Hermitian operator S such that $S{\hat{H}}_{\mathrm{eff}}S=-{\hat{H}}_{\mathrm{eff}}$. At the level of $\hat{U}$, the chiral symmetry then implies that
(87)$$S\hat{U}S={\hat{U}}^{-1}.$$
As in the case of static systems, the chiral symmetry of $\hat{U}$ has a direct implication for the symmetry of its spectrum. If $|\Psi_{E}\rangle$ is an eigenstate of $\hat{U}$ with the quasienergy *E*, i.e., $\hat{U}|\Psi_{E}\rangle =e^{-iE}|\Psi_{E}\rangle$, we immediately have
(88)$$\hat{U}(S|\Psi_{E}\rangle )=e^{-i(-E)}(S|\Psi_{E}\rangle ),$$
which means that $S|\Psi_{E}\rangle$ is also an eigenstate of $\hat{U}$ with the quasienergy −E. The Floquet spectrum of $\hat{U}$ is then symmetric with respect to both the quasienergies E=0 and E=π. The latter is because −E and *E* are identified at the quasienergy π. When the spectrum is gapped at E=0 and π, the quasienergy levels of $\hat{U}$ could be grouped into two clusters. One of them has the quasienergy ReE∈(−π,0) and the other one has ReE∈(0,π). They could meet with each other at either the quasienergy zero or π, leading to two possible phase transitions. This implies that a complete topological characterization of a chiral-symmetric Floquet system should require at least two winding numbers, which is rather different from the case of static systems where a single winding number is sufficient. To identify these winding numbers, let us consider the example of a periodically kicked 1D system, whose Hamiltonian and Floquet operator take the forms of Equations (Equation 15) and (Equation 16). We also assume that the two parts of the Hamiltonians ${\hat{H}}_{0}$ and ${\hat{H}}_{1}$ in Equation (Equation 15) have the same chiral symmetry S. At the level of $\hat{U}=e^{-i{\hat{H}}_{0}}e^{-i{\hat{H}}_{1}}$ (assuming T=1), the form of a chiral symmetry is not transparent. However, we can apply similarity transformations to $\hat{U}$ and express it in two symmetric time frames [69] as
(89)$${\hat{U}}_{1}=e^{-\frac{i}{2}{\hat{H}}_{1}}e^{-i{\hat{H}}_{0}}e^{-\frac{i}{2}{\hat{H}}_{1}},$$
(90)$${\hat{U}}_{2}=e^{-\frac{i}{2}{\hat{H}}_{0}}e^{-i{\hat{H}}_{1}}e^{-\frac{i}{2}{\hat{H}}_{0}}.$$
It is then clear that $S{\hat{U}}_{\alpha }S={\hat{U}}_{\alpha }^{-1}$ for α=1,2, that is, the Floquet operators ${\hat{U}}_{1}$ and ${\hat{U}}_{2}$ in the two symmetric time frames respect the same chiral symmetry S. We can, thus, introduce a winding number for each of them under the PBC as [256]
(91)$$w_{\alpha }=\int_{-\pi }^{\pi }\frac{dk}{4\pi }\mathrm{Tr}\left[SQ_{\alpha }\left(k\right)i\partial_{k}Q_{\alpha }\left(k\right)\right],$$
where k∈[−π,π), and
(92)Qα(k)=∑nsgn{Re[En(k)]}|nα(k)〉〈n˜α(k)|.
The $E_{n}\left(k\right)$ in Equation (Equation 92) now denotes the quasienergy of the Floquet eigenstate $|n_{\alpha }\left(k\right)\rangle$ of ${\hat{U}}_{\alpha }$ at the quasimomentum *k* under the PBC. Using $w_{1}$ and $w_{2}$, we can construct another pair of winding numbers $w_{0}$ and $w_{\pi }$, given by [256]
(93)$$w_{0}=\frac{w_{1}+w_{2}}{2},\;w_{\pi }=\frac{w_{1}-w_{2}}{2}.$$

In Section 3.2 and Section 3.3, we will demonstrate with explicit examples that $w_{0}$ ($w_{\pi }$) can correctly capture the bulk topological transitions of non-Hermitian Floquet bands through the gap closing/reopening at the quasienergy E=0 (E=π) in various chiral-symmetric, non-Hermitian Floquet insulating and superconducting models. Furthermore, in the absence of NHSE, the $w_{0}$ and $w_{\pi }$ can also capture the numbers of Floquet edge modes at zero and π quasienergies under the OBC, and thus, are capable of describing the bulk–edge correspondence of the related models. In the presence of NHSE, we can retrieve the characterization of topological transitions and bulk–edge correspondence in chiral-symmetric, non-Hermitian Floquet systems under the OBC through the open-bulk winding numbers, in analogy with Equation (Equation 85). For the ${\hat{U}}_{1}$ and ${\hat{U}}_{2}$ in Equations (Equation 89) and (Equation 90), we can define a winding number for each of them under the OBC as [263]
(94)$$W_{\alpha }=-\frac{1}{L_{B}}{\mathrm{Tr}}_{B}\left(SQ_{\alpha }\left[Q_{\alpha },\hat{N}\right]\right),\;\alpha =1,2.$$
Here, the meanings of $L_{B}$, ${\mathrm{Tr}}_{B}$, S, and $\hat{N}$ are the same as those in Equation (Equation 85). The Floquet band projector in the time frame α is given by
(95)Qα=∑|ψjα〉∈bulksgn[Re(Ej)]|ψjα〉〈ψ˜jα|,
where $|\psi_{j}^{\alpha }\rangle$ is the *j*th bulk eigenstate of ${\hat{U}}_{\alpha }$ (α=1,2) with the quasienergy $E_{j}$ under the OBC. The linear combinations of $W_{1}$ and $W_{2}$ lead to another pair of winding numbers [263]
(96)$$W_{0}=\frac{W_{1}+W_{2}}{2},\;W_{\pi }=\frac{W_{1}-W_{2}}{2}.$$

In Section 3.2, we will illustrate that with the help of the winding numbers $\left(w_{0},w_{\pi }\right)$ and $\left(W_{0},W_{\pi }\right)$, a dual topological characterization of the phase transitions, edge states, and bulk–edge correspondence can be established for 1D, chiral-symmetric non-Hermitian Floquet systems under different boundary conditions, regardless of whether the NHSE is present or not [263]. Interestingly, the winding numbers $\left(w_{0},w_{\pi }\right)$ may both become half-integer quantized due to the presence of Floquet EP in the bulk, thus revealing the presence of Floquet exceptional topology. Meanwhile, the open-bulk winding numbers $\left(W_{0},W_{\pi }\right)$ are always integer quantized.

### 2.6. Dynamical Indicators

In this subsection, we review two complementary dynamical probes in position and momentum spaces. Both of them can be used to characterize the topological properties of 1D non-Hermitian Floquet systems with chiral symmetry. These indicators allow us to extract the topological winding numbers of the system from its long-time stroboscopic dynamics [256,257,258]. The measurement of these indicators could, thus, provide evidence for the existence of non-Hermitian Floquet topological matter.

#### 2.6.1. Dynamic Winding Number (DWN)

The DWN [277,278], obtained from the long-time stroboscopic average of spin textures, could provide us with information about the bulk topological properties of non-Hermitian Floquet systems [258]. Let us consider a 1D, chiral-symmetric non-Hermitian Floquet system with two quasienergy bands. Under the PBC, we can express its Floquet operator in the symmetric time frame α (=1,2) as ${\hat{U}}_{\alpha }=\sum_{k\in \mathrm{BZ}}|k\rangle e^{-iH_{\alpha }\left(k\right)}\langle k|$. Here, $H_{\alpha }\left(k\right)$ is the effective Hamiltonian in time frame α and k∈[−π,π) is the quasimomentum. ${\hat{U}}_{\alpha }$ and $H_{\alpha }\left(k\right)$ share the same chiral symmetry S, i.e., $SH_{\alpha }\left(k\right)S=-H_{\alpha }\left(k\right)$. For a non-Hermitian system, the ${\hat{U}}_{\alpha }$ is generally not unitary and $H_{\alpha }\left(k\right)$ is also not Hermitian. The right and left biorthonormal eigenvectors $\left\{|n_{\alpha }\left(k\right)\rangle \right\}$ and $\left\{|{\tilde{n}}_{\alpha }\left(k\right)\rangle \right\}$ of $H_{\alpha }\left(k\right)$ satisfy the eigenvalue equations
(97)$$H_{\alpha }\left(k\right)|n_{\alpha }\left(k\right)\rangle =E_{n}\left(k\right)|n_{\alpha }\left(k\right)\rangle ,$$
(98)$$\langle {\tilde{n}}_{\alpha }\left(k\right)|H_{\alpha }\left(k\right)=\langle {\tilde{n}}_{\alpha }\left(k\right)|E_{n}\left(k\right).$$
Here, n=± are the indices of the two Floquet bands with the quasienergies $E_{\pm }\left(k\right)\equiv \pm E\left(k\right)$. The biorthonormal relationship requires
(99)$$\langle {\tilde{n}}_{\alpha }\left(k\right)|n_{\alpha }^{\prime }\left(k\right)\rangle =\delta_{nn^{\prime }},\;\sum_{n=\pm }|n_{\alpha }\left(k\right)\rangle \langle {\tilde{n}}_{\alpha }\left(k\right)|=1,\;\alpha =1,2.$$
We consider the case in which the system is prepared in a general initial state $|\psi_{\alpha }\left(k,0\right)\rangle =\sum_{n=\pm }c_{n}\left(k\right)|n_{\alpha }\left(k\right)\rangle$. The corresponding initial state in the left Hilbert space reads $|{\tilde{\psi }}_{\alpha }\left(k,0\right)\rangle =\sum_{n\pm }c_{n}\left(k\right)|{\tilde{n}}_{\alpha }\left(k\right)\rangle$, such that initially $\sum_{n=\pm }{\left|c_{n}\left(k\right)\right|}^{2}=1$ at each *k*. After the stroboscopic evolution over a number of *ℓ* driving periods, the right initial state becomes
(100)$$|\psi_{\alpha }\left(k,\ell \right)\rangle =\sum_{n=\pm }c_{n}\left(k\right)e^{-i\ell E_{n}\left(k\right)}|n_{\alpha }\left(k\right)\rangle .$$
For the left initial state, we assume it to be evolved by a different effective Hamiltonian ${\tilde{H}}_{\alpha }\left(k\right)=\sum_{\pm }E_{n}\left(k\right)|{\tilde{n}}_{\alpha }\left(k\right)\rangle \langle n_{\alpha }\left(k\right)|$ (In our theoretical analysis, we find that in order to get a quantized DWN which can be connected to the static winding number, the left eigenvector should be evolved by the effective Hamiltonian ${\tilde{H}}_{\alpha }$ instead of $H_{\alpha }^{†}$ [258]), so that after the evolution over *ℓ* driving periods it reaches the state
(101)$$|{\tilde{\psi }}_{\alpha }\left(k,\ell \right)\rangle =\sum_{n=\pm }c_{n}\left(k\right)e^{-i\ell E_{n}\left(k\right)}|{\tilde{n}}_{\alpha }\left(k\right)\rangle .$$
Note that the dynamical equation of $|{\tilde{\psi }}_{\alpha }\left(k,0\right)\rangle$ we used here is different from that employed in our study of the APT in Section 2.4.

The stroboscopic average of an observable $\hat{O}$ over $|{\tilde{\psi }}_{\alpha }\left(k,t\right)\rangle$ at the time t=ℓT is then given by
(102)$${\langle \hat{O}\left(k,\ell \right)\rangle }_{\alpha }=\frac{\langle {\tilde{\psi }}_{\alpha }\left(k,\ell \right)|\hat{O}|\psi_{\alpha }\left(k,\ell \right)\rangle }{\langle {\tilde{\psi }}_{\alpha }\left(k,\ell \right)|\psi_{\alpha }\left(k,\ell \right)\rangle }.$$
Without the loss of generality, we consider the chiral-symmetric $H_{\alpha }\left(k\right)$ to be in the form of
(103)$$H_{\alpha }\left(k\right)=h_{\alpha x}\left(k\right)\sigma_{x}+h_{\alpha y}\left(k\right)\sigma_{y}.$$
Note that any two out of the three Pauli matrices ($\sigma_{x},\sigma_{y},\sigma_{z}$) can be chosen to enter the $H_{\alpha }\left(k\right)$, and the other Pauli matrix (e.g., $\sigma_{z}$ for the $H_{\alpha }\left(k\right)$ in Equation (Equation 103)) plays the role of the chiral symmetry operator S. By diagonalizing $H_{\alpha }\left(k\right)$, we obtain the biorthonormal eigenvectors as
(104)$$|n_{\alpha }\left(k\right)\rangle =\frac{1}{\sqrt{2}E_{n}\left(k\right)}\left[\begin{array}{c} h_{\alpha x}\left(k\right)-ih_{\alpha y}\left(k\right)\\ E_{n}\left(k\right) \end{array}\right],\;|{\tilde{n}}_{\alpha }\left(k\right)\rangle =\frac{1}{\sqrt{2}E_{n}^{*}\left(k\right)}\left[\begin{array}{c} h_{\alpha x}^{*}\left(k\right)-ih_{\alpha y}^{*}\left(k\right)\\ E_{n}^{*}\left(k\right) \end{array}\right],$$
where $E_{n}\left(k\right)=n\sqrt{h_{\alpha x}^{2}\left(k\right)+h_{\alpha y}^{2}\left(k\right)}$ for n=±. We can now compute the stroboscopic-averaged spin textures in the long-time limit. For the $H_{\alpha }\left(k\right)$ in Equation (Equation 103), this means that we need to find the multi-cycle averages of $\sigma_{x}$ and $\sigma_{y}$. According to Equation (Equation 102), they are given by
(105)$$r_{j}^{\alpha }\left(k\right)\equiv \lim_{N\to \infty }\frac{1}{N}\sum_{\ell =1}^{N}{\langle \sigma_{j}\left(k,\ell \right)\rangle }_{\alpha },$$
where j=x,y and α=1,2. *N* counts the total number of driving periods. From the averaged spin textures $\left[r_{x}^{\alpha }\left(k\right),r_{y}^{\alpha }\left(k\right)\right]$, we can define the dynamic winding angle as
(106)$$\theta_{yx}^{\alpha }\left(k\right)\equiv \arctan\left[\frac{r_{y}^{\alpha }\left(k\right)}{r_{x}^{\alpha }\left(k\right)}\right].$$
The net winding number of $\theta_{yx}^{\alpha }\left(k\right)$ over a cycle in *k*-space defines the DWN in the time frame α, i.e.,
(107)$$\nu_{\alpha }=\int_{-\pi }^{\pi }\frac{dk}{2\pi }\partial_{k}\theta_{yx}^{\alpha }\left(k\right),\;\alpha =1,2.$$

In Ref. [258], it was proved with straightforward calculations that in the limit N→∞, the $\nu_{\alpha }$ converges to the $w_{\alpha }$ in Equation (Equation 91) if the initial condition satisfies $c_{\pm }\left(k\right)\neq 0$ at each *k*. Therefore, by preparing the initial state at different *k* under this condition and measuring the averaged spin textures over a long stroboscopic time, we can obtain the winding numbers $\left(w_{0},w_{\pi }\right)$ of a chiral-symmetric 1D Floquet system (either Hermitian or non-Hermitian) through the following combinations of $\left(\nu_{1},\nu_{2}\right)$ in two symmetric time frames [258], i.e.,
(108)$$w_{0}=\frac{\nu_{1}+\nu_{2}}{2},\;w_{\pi }=\frac{\nu_{1}-\nu_{2}}{2}.$$

In Section 3, we will illustrate the application of this dynamical–topological correspondence to non-Hermitian Floquet topological insulators in one dimension. We will see that both integer and half-integer quantized topological winding numbers can be extracted from the DWN.

#### 2.6.2. Mean Chiral Displacement (MCD)

The MCD allows us to detect the winding numbers of a Floquet system from the long-time-averaged chiral displacement of an initially localized wavepacket. This was first proposed as a means to probe the topological invariants of chiral-symmetric topological insulators in one dimension [279,280,281]. Later, the MCD was used to obtain the winding numbers of Floquet systems [78,81] and also generalized to 2D higher-order topological insulators [80]. Its applicability was demonstrated both theoretically and experimentally [279,281].

For a chiral-symmetric non-Hermitian system, the chiral displacement in the symmetric time frame α (=1,2) can be defined as
(109)$$C_{\alpha }\left(t\right)=\mathrm{Tr}\left[\rho_{0}{\left({\hat{\tilde{U}}}_{\alpha }^{†}\right)}^{t}\left(\hat{N}⊗S\right){\hat{U}}_{\alpha }^{t}\right].$$
Here, t=ℓT denotes the stroboscopic time, with *T* being the driving period. $\hat{N}$ is the unit cell position operator and S is the chiral symmetry operator. The initial state $\rho_{0}$ can be chosen as a state localized in the middle of the lattice. For example, $\rho_{0}$ may take the form of $(|0\rangle \langle 0|⊗\sigma_{0})/2$ for a 1D bipartite lattice, where |0〉 is the eigenbasis of the central unit cell and $\sigma_{0}$ is the identity operator acting on the internal space of the two sublattices. Both the Floquet operator ${\hat{U}}_{\alpha }$ and its dual ${\hat{\tilde{U}}}_{\alpha }$ respect the chiral symmetry S. In the lattice representation, they can be expressed as
(110)$${\hat{U}}_{\alpha }=\sum_{j=1}^{L}e^{-iE_{j}}|\psi_{j}^{\alpha }\rangle \langle {\tilde{\psi }}_{j}^{\alpha }|,\;{\hat{\tilde{U}}}_{\alpha }=\sum_{j=1}^{L}e^{-iE_{j}}|{\tilde{\psi }}_{j}^{\alpha }\rangle \langle \psi_{j}^{\alpha }|.$$
Here, *L* denotes the total number of degrees of freedom of the lattice. $|\psi_{j}^{\alpha }\rangle$ and $\langle {\tilde{\psi }}_{j}^{\alpha }|$ denote the right and left eigenvectors of ${\hat{U}}_{\alpha }$ with the quasienergy $E_{j}$. They form a biorthonormal basis such that
(111)$$\langle {\tilde{\psi }}_{j}^{\alpha }|\psi_{m}^{\alpha }\rangle =\delta_{jm},\;\sum_{j=1}^{L}|\psi_{j}^{\alpha }\rangle \langle {\tilde{\psi }}_{j}^{\alpha }|=1.$$
Note that ${\hat{\tilde{U}}}_{\alpha }$ is not the Hermitian conjugate of ${\hat{U}}_{\alpha }$ in general.

We now consider the stroboscopic long-time average of $C_{\alpha }\left(t\right)$ in the time frames α=1,2 for a 1D non-Hermitian Floquet system with chiral symmetry. Under the PBC, taking the initial state to be $\rho_{0}=(|0\rangle \langle 0|⊗\sigma_{0})/2$ and performing the Fourier transformation from position to momentum representations, we find the $C_{\alpha }\left(t\right)$ in Equation (Equation 109) to be [257]
(112)$$C_{\alpha }\left(t\right)=\frac{1}{2}\int_{-\pi }^{\pi }\frac{dk}{2\pi }\mathrm{Tr}\left[{\tilde{U}}_{\alpha }^{†t}\left(k\right)Si\partial_{k}U_{\alpha }^{t}\left(k\right)\right].$$
Here k∈[−π,π) is the quasimomentum. $U_{\alpha }\left(k\right)=\langle k|{\hat{U}}_{\alpha }|k\rangle$ and ${\tilde{U}}_{\alpha }\left(k\right)=\langle k|{\hat{\tilde{U}}}_{\alpha }|k\rangle$ act on the internal degrees of freedom (spins and/or sublattices) of the system at a fixed *k*. Taking the long-time stroboscopic average and incorporating the normalization factor $\mathrm{Tr}[\rho_{0}{\left({\hat{\tilde{U}}}_{\alpha }^{†}\right)}^{t}{\hat{U}}_{\alpha }^{t}]$, we find the expression of MCD as [257]
(113)$${\bar{C}}_{\alpha }=\lim_{\ell \to \infty }\frac{1}{\ell T}\sum_{t=T}^{\ell T}\int_{-\pi }^{\pi }\frac{dk}{2\pi }\frac{\mathrm{Tr}\left[{\tilde{U}}_{\alpha }^{†t}\left(k\right)Si\partial_{k}U_{\alpha }^{t}\left(k\right)\right]}{\mathrm{Tr}\left[{\tilde{U}}_{\alpha }^{†t}\left(k\right)U_{\alpha }^{t}\left(k\right)\right]}.$$
For a given $H_{\alpha }\left(k\right)$ of the form in Equation (Equation 103), we have $S=\sigma_{z}$, $U_{\alpha }\left(k\right)=e^{-iH_{\alpha }\left(k\right)}$, and ${\tilde{U}}_{\alpha }\left(k\right)=e^{-iE\left(k\right)H_{\alpha }^{†}\left(k\right)/E^{*}\left(k\right)}$, where $E\left(k\right)=\frac{1}{2}\mathrm{Tr}\left[H_{\alpha }^{2}\left(k\right)\right]$. One can then work out Equation (Equation 113) explicitly and obtain [257]
(114)$${\bar{C}}_{\alpha }=\frac{w_{\alpha }}{2},\;\alpha =1,2.$$
Here, the $w_{\alpha }$ is nothing but the winding number of ${\hat{U}}_{\alpha }$ as defined in Equation (Equation 91). Therefore, by measuring the MCDs $\left({\bar{C}}_{1},{\bar{C}}_{2}\right)$ in two symmetric time frames, we can determine the topological winding numbers $\left(w_{0},w_{\pi }\right)$ of a 1D chiral-symmetric Floquet system through the relations [257]
(115)$$w_{0}=C_{0}\equiv {\bar{C}}_{1}+{\bar{C}}_{2},\;w_{\pi }=C_{\pi }\equiv {\bar{C}}_{1}-{\bar{C}}_{2}.$$

The MCD provides a real space complementary to the DWN. In Section 3, we will demonstrate the usage of MCD to dynamically probing the winding numbers of first- and second-order non-Hermitian Floquet topological insulators in one [257] and two [261] spatial dimensions.

### 2.7. Localization Transition and Mobility Edge

In the last part of this subsection, we recap some tools that can be used to characterize the real-to-complex spectral transitions, localization transitions, and mobility edges in disordered non-Hermitian Floquet systems [264,266]. Considering the relevance to this review, we focus on a 1D quadratic lattice model of the form
(116)$$\hat{H}\left(t\right)=\sum_{\langle n,n^{\prime }\rangle }\left[J_{n}\left(t\right)e^{\gamma_{n}}{\hat{c}}_{n}^{†}{\hat{c}}_{n^{\prime }}+J_{n}^{*}\left(t\right)e^{-\gamma_{n}}{\hat{c}}_{n^{\prime }}^{†}{\hat{c}}_{n}\right]+\sum_{n}V_{n}\left(t\right){\hat{c}}_{n}^{†}{\hat{c}}_{n}.$$
Here, $\langle n,n^{\prime }\rangle$ includes the lattice site indices *n* and $n^{\prime }$ with $n^{\prime }>n$. ${\hat{c}}_{n}^{†}$ (${\hat{c}}_{n}$) creates (annihilates) a particle on the site *n* and the parameter γ∈R. The Hamiltonian $\hat{H}\left(t\right)$ is non-Hermitian if $\gamma_{n}\neq 0$ for some *n* (nonreciprocal hopping) or $V_{n}\left(t\right)\neq V_{n}^{*}\left(t\right)$ (onsite gain and loss). $\hat{H}\left(t\right)$ is further time-periodic if $J_{n}\left(t\right)=J_{n}\left(t+T\right)$ and $V_{n}\left(t\right)=V_{n}\left(t+T\right)$ for all *n*, with *T* being the driving period. The disorder terms may be included within $V_{n}\left(t\right)$ (diagonal disorder) or $J_{n}\left(t\right)$ (off-diagonal disorder). As an example, for a 1D non-Hermitian quasicrystal (NHQC) with correlated onsite disorder, the $V_{n}\left(t\right)$ may take the form of $V_{n}\left(t\right)=V\left(t\right)\cos\left(2\pi \alpha n+i\beta \right)$. Here, α is irrational and β∈R, with iβ describing an imaginary phase shift in the superlattice potential $V_{n}$.

The Floquet operator of the system described by the $\hat{H}\left(t\right)$ in Equation (Equation 116) takes the general form of $\hat{U}=\hat{T}e^{-i\int_{0}^{T}\hat{H}\left(t\right)dt}=e^{-i{\hat{H}}_{\mathrm{eff}}T}$. If $\hat{H}\left(t\right)$ respects the PT symmetry, ${\hat{H}}_{\mathrm{eff}}$ could have the same symmetry and the quasienergy spectrum of $\hat{U}$ could be real. With the increase in the non-Hermitian parameter of $\hat{H}\left(t\right)$, the Floquet spectrum of $\hat{U}$ may undergo a PT-breaking transition, after which certain quasienergies of $\hat{U}$ (obtained by solving $\hat{U}|\Psi \rangle =e^{-iE}|\Psi \rangle$) may acquire non-vanishing imaginary parts. To take into account such a spectral transition, we introduce the following quantities
(117)max|ImE|≡maxj∈{1,…,L}(|ImEj|),
(118)ρ≡N(ImE≠0)/L.
Here, $E_{j}$ is the *j*th quasienergy eigenvalue of $\hat{U}$. *L* counts the Hilbert space dimension of $\hat{U}$, i.e., the total number of degrees of freedom of the lattice. N(ImE≠0) denotes the number of quasienergy eigenvalues whose imaginary parts are nonzero. ρ thus describes the density of states with complex quasienergies in the system. In the PT-invariant phase, we would have max|ImE|=ρ=0, implying that all the quasienergies of $\hat{U}$ are real. In the PT-broken phase, we would have max|ImE|>0 and ρ∈(0,1], meaning that there is a finite number of eigenstates of $\hat{U}$ whose quasienergies are complex. In particular, we would have ρ≃1 if almost all the Floquet eigenstates of $\hat{U}$ possess complex quasienergies. Therefore, by locating the positions where both the max|ImE| and ρ start to deviate from zero in the parameter space, we can identify the PT transition points for the Floquet spectrum of $\hat{U}$. We will see that due to the interplay between periodic drivings and non-Hermitian effects, both PT-symmetry breaking and restoring transitions could be induced by tuning a single parameter in the Floquet NHQC.

The localization nature of the Floquet eigenstates of $\hat{U}$ can be characterized by the statistics of its quasienergy levels and the inverse participation ratios (IPRs). Let us denote the normalized right eigenvectors of $\hat{U}$ and their corresponding quasienergies as $\left\{|\psi_{j}\rangle |j=1,\ldots ,L\right\}$ and $\left\{E_{j}|j=1,\ldots ,L\right\}$. Along the real axis, the spacing between the *j*th and the (j−1)th quasienergies of $\hat{U}$ is given by ϵj=ReEj−ReEj−1, from which we obtain the ratio between two adjacent spacings of quasienergy levels as
(119)$$g_{j}=\frac{\min(\epsilon_{j},\epsilon_{j+1})}{\max(\epsilon_{j},\epsilon_{j+1})},\;j=2,\ldots ,L-1.$$
Here, the $\max(\epsilon_{j},\epsilon_{j+1})$ and $\min(\epsilon_{j},\epsilon_{j+1})$ are the maximum and minimum between the $\epsilon_{j}$ and $\epsilon_{j+1}$, respectively. The statistical property of adjacent gap ratios can then be obtained by averaging over all $g_{j}$ in the thermodynamic limit, i.e.,
(120)$$\bar{g}=\lim_{L\to \infty }\frac{1}{L}\sum_{j}g_{j}.$$

We would have $\bar{g}\to 0$ if all the bulk Floquet eigenstates of $\hat{U}$ are extended. Comparatively, we expect $\bar{g}$ to approach a constant ${\bar{g}}_{\max}>0$ if all the bulk eigenstates of $\hat{U}$ are localized. If we find $\bar{g}\in \left(0,{\bar{g}}_{\max}\right)$, extended and localized eigenstates of $\hat{U}$ should coexist and be separated by some mobility edges in the quasienergy spectrum. The $\bar{g}$ can thus be utilized to distinguish between phases with different localization nature in 1D non-Hermitian Floquet systems from the perspective of level statistics [282,283,284,285,286]. In the lattice representation, we can expand the right eigenvector $|\psi_{j}\rangle$ as $|\psi_{j}\rangle =\sum_{n=1}^{L}\psi_{n}^{j}|n\rangle$, where $\psi_{n}^{j}=\langle n|\psi_{j}\rangle$ and $\sum_{n}{\left|\psi_{n}^{j}\right|}^{2}=1$. The inverse and normalized participation ratios of $|\psi_{j}\rangle$ in the real space can then be defined as
(121)$${\mathrm{IPR}}_{j}=\sum_{n=1}^{L}{\left|\psi_{n}^{j}\right|}^{4},\;{\mathrm{NPR}}_{j}=\frac{1}{L}{\mathrm{IPR}}_{j}^{-1}.$$

For a localized Floquet eigenstate $|\psi_{j}\rangle$, we have ${\mathrm{IPR}}_{j}\to \lambda_{j}$ and ${\mathrm{NPR}}_{j}\to 0$ in the thermodynamic limit, where the Lyapunov exponent $\lambda_{j}$ of $|\psi_{j}\rangle$ could be a function of its quasienergy $E_{j}$. If $|\psi_{j}\rangle$ represents an extended state, we have ${\mathrm{IPR}}_{j}\to 0$ and ${\mathrm{NPR}}_{j}\to 1$. The global localization features of the Floquet system described by $\hat{U}$ can then be extracted from the combined information of $\left\{{\mathrm{IPR}}_{j}|j=1,\ldots ,L\right\}$ and $\left\{{\mathrm{NPR}}_{j}|j=1,\ldots ,L\right\}$. For ease of usage, we introduce the following localization indicators
(122)$${\mathrm{IPR}}_{\max}=\max_{j\in \{1,\ldots ,L\}}\left({\mathrm{IPR}}_{j}\right),$$
(123)$${\mathrm{IPR}}_{\min}=\min_{j\in \{1,\ldots ,L\}}\left({\mathrm{IPR}}_{j}\right),$$
(124)$$\zeta =\log_{10}\left({\mathrm{IPR}}_{\mathrm{ave}}\cdot {\mathrm{NPR}}_{\mathrm{ave}}\right),$$
where the ${\mathrm{IPR}}_{\mathrm{ave}}=\frac{1}{L}\sum_{j=1}^{L}{\mathrm{IPR}}_{j}$ and ${\mathrm{NPR}}_{\mathrm{ave}}=\frac{1}{L}\sum_{j=1}^{L}{\mathrm{NPR}}_{j}$ are the averages of ${\mathrm{IPR}}_{j}$ and ${\mathrm{NPR}}_{j}$ over all Floquet eigenstates. It is not hard to see that in the metallic phase, where all Floquet eigenstates are extended, we have ${\mathrm{IPR}}_{\max}\to 0$, ${\mathrm{IPR}}_{\min}\to 0$ and $\zeta ∼-\log_{10}\left(L\right)\to -\infty$ in the limit L→∞. Instead, in the insulating phase with all Floquet eigenstates being localized, we have ${\mathrm{IPR}}_{\max}>0$, ${\mathrm{IPR}}_{\min}>0$ and ζ→−∞ [287]. In the critical phase, where extended and localized Floquet eigenstates are coexistent, we have ${\mathrm{IPR}}_{\max}>0$, ${\mathrm{IPR}}_{\min}\to 0$ together with a finite ζ. Assembling the information obtained from ${\mathrm{IPR}}_{\max}$, ${\mathrm{IPR}}_{\min}$, and ζ thus allows us to distinguish the extended, localized, and critical mobility edge phases of a non-Hermitian Floquet system with disorder. These tools will be applied to our study of the Floquet NHQC in Section 3.4.

We can also probe the transport nature of distinct non-Hermitian Floquet phases from the wave packet dynamics. Let us consider a generic and normalized initial state |Ψ(0)〉 in the lattice representation. After the stroboscopic evolution over a number of *ℓ* driving periods by $\hat{U}$, the final state turns out to be $|\Psi^{\prime }\left(t=\ell T\right)\rangle ={\hat{U}}^{\ell }|\Psi \left(0\right)\rangle$. Since the Floquet operator $\hat{U}$ is not unitary for a non-Hermitian $\hat{H}\left(t\right)$ in general, the norm of |Ψ(0)〉 cannot be preserved during the evolution. We can express the normalized state at t=ℓT as $\left|\Psi \left(t\right)\rangle =\right|\Psi^{\prime }\left(t\right)\rangle /\sqrt{\langle \Psi^{\prime }\left(t\right)|\Psi^{\prime }\left(t\right)\rangle }$. The real space expansion $|\Psi \left(t\right)\rangle =\sum_{n=1}^{L}\psi_{n}\left(t\right)|n\rangle$ then provides us with the probability amplitude $\psi_{n}\left(t\right)=\langle n|\Psi \left(t\right)\rangle$ of the normalized final state |Ψ(t)〉 on the lattice site *n* at time *t*. Using the collection of amplitudes $\left\{\psi_{n}\left(t\right)|n=1,\ldots ,L\right\}$, we can define the following dynamical quantities
(125)$$\langle x\left(t\right)\rangle =\sum_{n=1}^{L}n|\psi_{n}{\left(t\right)|}^{2},\;\langle x^{2}\left(t\right)\rangle =\sum_{n=1}^{L}n^{2}{\left|\psi_{n}\left(t\right)\right|}^{2},$$
(126)$$\Delta x\left(t\right)=\sqrt{\langle x^{2}\left(t\right)\rangle -{\langle x\left(t\right)\rangle }^{2}},\;v\left(t\right)=\frac{1}{t}\sqrt{\langle x^{2}\left(t\right)\rangle },$$
where the stroboscopic time t=ℓT and ℓ∈Z. It is clear that 〈x(t)〉, Δx(t), and v(t) describe the center, standard deviation, and spreading speed of the wavepacket |Ψ(t)〉 in the lattice representation, respectively. For simplicity, we usually choose the initial state |Ψ(0)〉 to be exponentially localized at a single site that is deep inside the bulk of the lattice. If the Floquet system described by $\hat{U}$ resides in a localized phase, we expect the 〈x(t)〉 and Δx(t) to stay around their initial values, which means that the wavepacket almost does not move and spread. In this case, the speed v(t) should also tend to zero for a long-time evolution (ℓ≫1). If the system stays in an extended phase, we expect the increasing of Δx(t) with time due to the spreading of the initial wavepacket. Meanwhile, the 〈x(t)〉 may or may not change with time, depending on whether the hopping amplitudes are symmetric. For a nearest-neighbor, nonreciprocal hopping, as in Equation (Equation 116), we may have 〈x(t)〉∝t and $\Delta x\left(t\right)\propto \sqrt{t}$ for a metallic phase of the system. The v(t) should take a maximal possible value $v_{\max}\left(t\right)$ in this case. If the system is prepared in a critical mobility edge phase, we expect both the 〈x(t)〉 and Δx(t) to show intervening behaviors, while the averaged spreading speed v(t) should satisfy $0<v\left(t\right)<v_{\max}\left(t\right)$. Therefore, we can exploit the dynamical quantities 〈x(t)〉, Δx(t), and v(t) to discriminate phases with different localization properties in disordered non-Hermitian Floquet systems. We will illustrate such an application for Floquet NHQC in Section 3.4.

## 3. Non-Hermitian Floquet Phases of Matter

This section collects and treats typical examples of non-Hermitian Floquet phases discovered in our previous work [256,257,258,259,260,261,262,263,264,265,266]. We will see that the interplay between periodic driving fields and gain/loss or nonreciprocal effects can induce rich phases and transitions in non-Hermitian Floquet systems.

### 3.1. Non-Hermitian Floquet Exceptional Topology

Let us start with a simple model, whose Floquet effective Hamiltonian and quasienergy dispersion are given by Equations (Equation 55) and (Equation 57). To be explicit, we choose
(127)$$h_{x}=\sin k,\;h_{z}=\mu +\cos k,$$
where k∈(−π,π] denotes the quasimomentum in the first Brillouin zone, and the system parameter μ∈R. The $H_{\mathrm{eff}}$ in Equation (Equation 55), thus, describes the Bloch Hamiltonian of a 1D two-band lattice model under the PBC in the rotating frame, where μ corresponds to the amplitude of onsite potential and the nearest-neighbor hopping amplitude has been chosen to be the unit of energy. From Equations (Equation 57) and (Equation 127), we find that the two quasienergy bands of $H_{\mathrm{eff}}$ can meet with each other at the quasienergy ω/2 when
(128)$$\mu =\left\{\begin{array}{cc} \frac{\omega }{2}\pm \frac{\gamma }{2}-1, & \mathrm{for}\;k=0,\\ \frac{\omega }{2}\pm \frac{\gamma }{2}+1, & \mathrm{for}\;k=\pi . \end{array}\right.$$

If μ satisfies one of the above two equalities, there will be a second-order Floquet EP at k=0 or k=π in the conventional Brillouin zone. Note that both the quasienergy of this Floquet EP and its location in the parameter space depend on the frequency ω, which highlights the impact of the harmonic driving field on phase transitions in the system. Moreover, the appearance of an EP in *k*-space may lead to the formation of spectral loops on the complex energy plane and also the possible breakdown of bulk–edge correspondence in conventional topological phases. This issue might be overcome by incorporating the formalism of generalized Brillouin zone (GBZ) and non-Bloch band theory [164,166]. Following the standard procedure, we first make the substitutions $e^{ik}\to \beta$ and $e^{-ik}\to \beta^{-1}$, where the complex number β∈GBZ. The effective Hamiltonian in Equation (Equation 55) now takes the form of
(129)$$H_{\mathrm{eff}}\left(\beta \right)=\frac{\omega }{2}\sigma_{0}+\left(\frac{\beta -\beta^{-1}}{2i}+i\frac{\gamma }{2}\right)\sigma_{x}+\left(\mu +\frac{\beta +\beta^{-1}}{2}-\frac{\omega }{2}\right)\sigma_{z},$$
with the quasienergy bands
(130)$$\varepsilon_{\pm }\left(\beta \right)=\frac{\omega }{2}\pm \sqrt{{\left(\frac{\beta -\beta^{-1}}{2i}+\frac{i\gamma }{2}\right)}^{2}+{\left(\mu -\frac{\omega }{2}+\frac{\beta +\beta^{-1}}{2}\right)}^{2}}.$$
We next focus on the non-constant part of $\varepsilon_{\pm }\left(\beta \right)$, whose square takes the form of
(131)$$\epsilon_{\pm }^{2}\left(\beta \right)=\frac{a\beta^{2}+b\beta +c}{\beta },$$
where
(132)$$a=\mu -\frac{\omega }{2}+\frac{\gamma }{2},\;b=1+{\left(\mu -\frac{\omega }{2}\right)}^{2}-\frac{\gamma^{2}}{4},\;c=\mu -\frac{\omega }{2}-\frac{\gamma }{2}.$$

The $\varepsilon_{\pm }^{2}\left(\beta \right)$ is a Laurent polynomial of β. According to Ref. [166] (see also Ref. [288]), β∈GBZ for the $H_{\mathrm{eff}}\left(\beta \right)$ if $\epsilon_{\pm }^{2}\left(\beta \right)=\epsilon_{\pm }^{2}\left(\beta e^{i\theta }\right)$, with θ being a phase factor. For Equation (Equation 131), this means that $\beta^{2}=ce^{-i\theta }/a$. The GBZ is, thus, a circle of the radius
(133)$$r=\left|\beta \right|={\left|\frac{\mu -\omega /2-\gamma /2}{\mu -\omega /2+\gamma /2}\right|}^{1/2}.$$

It is clear that the GBZ radius *r* is controlled by both the driving field (through ω) and the non-Hermitian effect (through γ). In the Hermitian limit (γ→0), we have r→1 and the GBZ is reduced to the conventional BZ, as expected. The radius *r* of GBZ becomes zero or infinity if μ−ω/2=γ/2 or −γ/2, respectively. Note that these limiting cases do not correspond to second-order EPs where the two Floquet bands touch in the conventional BZ (see Equation (Equation 128)) or in the GBZ (see Equation (Equation 134) below). They instead correspond to the locations of intra-band high-order EPs of each Floquet band, as observable in the GBZ spectrum of Figure 2. Finally, we can use the GBZ to determine the gap-closing (phase transition) points of the system under the OBC. Setting the $\epsilon_{\pm }^{2}\left(\beta \right)=0$ in Equation (Equation 131), we find
(134)$$\mu =\frac{\omega }{2}\pm \left\{\begin{array}{cc} \sqrt{\gamma^{2}/4+1}, & |\gamma |\leq 2,\\ \sqrt{\gamma^{2}/4\pm 1}, & |\gamma |>2. \end{array}\right.$$

These bulk gap-closing conditions are clearly different from those found in conventional BZ under the PBC in Equation (Equation 128). Using the transfer matrix method [168], we can further obtain the parameter regions in which degenerate Floquet edge states appear at the quasienergy ω/2 (anomalous Floquet π edge modes), i.e.,
(135)$$\mu -\frac{\omega }{2}\in \left\{\begin{array}{cc} \left(-\sqrt{\gamma^{2}/4+1},\sqrt{\gamma^{2}/4+1}\right), & |\gamma |\leq 2,\\ \left(-\sqrt{\gamma^{2}/4+1},-\sqrt{\gamma^{2}/4-1}\right)\cup \left(\sqrt{\gamma^{2}/4-1},\sqrt{\gamma^{2}/4+1}\right), & |\gamma |>2. \end{array}\right.$$
Finally, we can characterize the different topological phases of the system by a non-Bloch winding number [169]
(136)$$W=\int_{\mathrm{GBZ}}\frac{d\beta }{4\pi i}\mathrm{Tr}\left[\sigma_{y}Q\left(\beta \right)\partial_{\beta }Q\left(\beta \right)\right],$$
where $\sigma_{y}$ is the chiral symmetry operator of the shifted effective Hamiltonian $H_{\mathrm{eff}}\left(\beta \right)-\frac{\omega }{2}\sigma_{0}$. The Q matrix is defined as $Q\left(\beta \right)=|\psi_{+}\left(\beta \right)\rangle \langle {\tilde{\psi }}_{+}\left(\beta \right)|-|\psi_{-}\left(\beta \right)\rangle \langle {\tilde{\psi }}_{-}\left(\beta \right)|$. $|\psi_{\pm }\left(\beta \right)\rangle$ are the right eigenvectors of $H_{\mathrm{eff}}\left(\beta \right)$ with the quasienergies $\varepsilon_{\pm }\left(\beta \right)$. $\langle {\tilde{\psi }}_{\pm }\left(\beta \right)|$ are the corresponding left eigenvectors. For our model, their explicit expressions are given by
(137)$$|\psi_{\pm }\left(\beta \right)\rangle =\frac{1}{\sqrt{2\left(\varepsilon_{\pm }-\frac{\omega }{2}\right)\left(\varepsilon_{\pm }-h_{z}\right)}}\left(\begin{array}{c} h_{x}+i\gamma /2\\ \varepsilon_{\pm }-h_{z} \end{array}\right),$$
(138)$$\langle {\tilde{\psi }}_{\pm }\left(\beta \right)|=\frac{1}{\sqrt{2\left(\varepsilon_{\pm }-\frac{\omega }{2}\right)\left(\varepsilon_{\pm }-h_{z}\right)}}\left(\begin{array}{cc} h_{x}+i\gamma /2 & \varepsilon_{\pm }-h_{z} \end{array}\right).$$

In Figure 2, we show the Floquet spectra of $H_{\mathrm{eff}}$ under different boundary conditions and the non-Bloch winding numbers of $H_{\mathrm{eff}}$ versus μ for some typical cases. We see that the Floquet spectra and the gap-closing points at the quasienergies E=±π could indeed be very different under the PBC and the OBC. Some high-order EPs appear in the bulk Floquet spectra under the OBC at μ=ω/2±γ/2 and the gap closing points, which signify the emergence of Floquet exceptional topology. In addition, the non-Bloch winding number *W* can correctly discriminate different topological phases and characterize topological phase transitions accompanied by quasienergy-band touchings under the OBC at E=π. We find W=1 (W=0) in the topologically nontrivial (trivial) phases with (without) anomalous Floquet π edge modes. The bulk–edge correspondence is then recovered. The gap-closing points and the parameter regions with Floquet π modes are found to be perfectly coincident with the predictions of Equations (Equation 134) and (Equation 135). These observations demonstrate the applicability of non-Bloch band theories to non-Hermitian Floquet systems. It is also straightforward to check that the system possesses Floquet NHSEs in a broad parameter regime under the OBC. The simple model introduced in this subsection, thus, allows us to obtain a bird’s-eye view on the nontrivial topological phenomena that could be brought about by the interplay between Floquet driving fields and non-Hermitian effects. Further examples with more striking features will be reviewed in the following subsections.

### 3.2. Non-Hermitian Floquet Topological Insulators

In this subsection, we review three types of non-Hermitian Floquet topological insulators in one and two spatial dimensions. For all the cases, we uncover that the collaboration between driving and gain/loss or nonreciprocal effects could induce topological insulating states unique to non-Hermitian Floquet systems. They are characterized by large topological invariants, many topological edge or corner modes, and separated by rich topological phase transitions. The bulk–edge (or bulk–corner) correspondence will also be established for each class of systems considered in this subsection.

#### 3.2.1. First-Order Topological Phase

We start with the characterization of 1D non-Hermitian Floquet topological insulators. One typical model that incorporates their rich topological properties is the following periodically quenched dimerized tight-binding lattice, whose time-dependent Hamiltonian reads [256]
(139)$$\hat{H}\left(t\right)=\left\{\begin{array}{cc} {\hat{H}}_{1}, & t\in [\ell T,\ell T+T/2),\\ {\hat{H}}_{2}, & t\in [\ell T+T/2,\ell T+T), \end{array}\right.$$
where
(140)$${\hat{H}}_{1}=\sum_{n}\left(ir_{y}{\hat{c}}_{n+1}^{†}{\hat{c}}_{n}+H.c.+2i\gamma {\hat{c}}_{n}^{†}{\hat{c}}_{n}\right)⊗\sigma_{y},$$
(141)$${\hat{H}}_{2}=\sum_{n}\left(r_{x}{\hat{c}}_{n}^{†}{\hat{c}}_{n+1}+\mu {\hat{c}}_{n}^{†}{\hat{c}}_{n}+H.c.\right)⊗\sigma_{x}.$$
Here, *T* is the driving period and we will set ℏ=T=1 in our calculations. ${\hat{c}}_{n}^{†}$$\left({\hat{c}}_{n}\right)$ creates (annihilates) a particle in the unit cell *n* of the lattice. $\sigma_{\alpha }$, α=x,y,z, are Pauli matrices acting on the two sublattice degrees of freedom A and B within each unit cell. The system parameters $r_{x},r_{y},\mu ,\gamma$ are all real. The non-Hermitian effect is introduced by the nonreciprocal intracell coupling term $2i\gamma \sigma_{y}$ applied over the first half of each driving period (see Figure 3a for an illustration of the model). Experimentally, such a term might be realized by a coupled-resonator optical waveguide with asymmetric internal scattering [289].

Considering the one-period evolution of the system from $t=\ell +0^{-}$ to $t=\ell +1+0^{-}$, the Floquet operator of $\hat{H}\left(t\right)$ takes the form of $\hat{U}=e^{-\frac{i}{2}{\hat{H}}_{2}}e^{-\frac{i}{2}{\hat{H}}_{1}}$. Its quasienergy spectrum and Floquet eigenstates are obtained by solving the eigenvalue equation $\hat{U}|\psi \rangle =e^{-iE}|\psi \rangle$. Under the OBC, we show the Floquet spectrum of $\hat{U}$ on the complex quasienergy plane ReE−ImE for some typical cases in Figure 3b–e (with $\left(\mu ,r_{y},\gamma \right)=\left(0,\pi /2,\mathrm{arccosh}\left(\sqrt{2}\right)\right)$, $r_{x}=\pi /2,\pi ,3\pi /2,2\pi$, and the number of unit cells N=150). The numbers of degenerate Floquet edge mode pairs at the quasienergies E=0 and E=π are denoted by $n_{0}$ and $n_{\pi }$ in the corresponding figure captions. We observe one or multiple pairs of edge modes at both the center (E=0) and boundary (E=π) of the first quasienergy Brillouin zone. Interestingly, when the numbers of these edge modes are the same, i.e., $n_{0}=n_{\pi }$, we find them to appear in different types of quasienergy gaps (see Figure 3c,e). For example, we find a pair of Floquet zero modes in the line gap at E=0, while another pair of Floquet π modes are found in the point gap at E=π. This is rather different from the situation in Hermitian Floquet systems, where the quasienergy is real and there is no distinction between point and line gaps. It is also different from the cases encountered in 1D, chiral-symmetric non-Hermitian static systems with two bands, where topological edge modes can only appear in either a point gap or a line gap at E=0. The Floquet spectra with hybrid (point plus line) quasienergy gaps are, thus, unique to non-Hermitian Floquet systems.

To understand the origin of the non-Hermitian Floquet edge modes at E=0 and E=π, we study the bulk topological properties of the system. Under the PBC, the Floquet operator of the system reads
(142)$$U\left(k\right)=e^{-i\left(\mu +r_{x}\cos k\right)\sigma_{x}}e^{-i\left(r_{y}\sin k+i\gamma \right)\sigma_{y}},$$
where k∈[−π,π) is the quasimomentum. The quasienergy spectrum of U(k) is given by
(143)$$\pm E\left(k\right)=\pm \arccos[\cos\left(\mu +r_{x}\cos k\right)\cos\left(r_{y}\sin k+i\gamma \right)],$$
and the gap-closing conditions can be obtained analytically by setting cos[E(k)]=±1 [256]. Transforming U(k) to symmetric time frames, we obtain
(144)$$U_{1}\left(k\right)=e^{-\frac{i}{2}\left(\mu +r_{x}\cos k\right)\sigma_{x}}e^{-i\left(r_{y}\sin k+i\gamma \right)\sigma_{y}}e^{-\frac{i}{2}\left(\mu +r_{x}\cos k\right)\sigma_{x}},$$
(145)$$U_{2}\left(k\right)=e^{-\frac{i}{2}\left(r_{y}\sin k+i\gamma \right)\sigma_{y}}e^{-i\left(\mu +r_{x}\cos k\right)\sigma_{x}}e^{-\frac{i}{2}\left(r_{y}\sin k+i\gamma \right)\sigma_{y}}.$$

It is clear that both $U_{1}\left(k\right)$ and $U_{2}\left(k\right)$ respect the chiral (sublattice) symmetry $S=\sigma_{z}$. We can, thus, characterize the non-Hermitian Floquet topological phases of U(k) by the winding numbers $\left(w_{0},w_{\pi }\right)$, according to Equation (Equation 93). In Figure 3f,g, we present the $w_{0}$ and $w_{\pi }$ of U(k) versus $\left(r_{x},r_{y}\right)$, respectively, yielding the topological phase diagram of the system. We find various non-Hermitian Floquet insulating phases. They are characterized by large integer winding numbers and separated by a series of topological phase transitions with quasienergy level crossings at E=0 (white solid lines in Figure 3f) and E=π (white dashed lines in Figure 3g). The winding numbers $\left(w_{0},w_{\pi }\right)$ could become arbitrarily large with the increase of system parameters (e.g., the intercell hopping amplitude $r_{x}$), which implies that Floquet phases with large topological invariants could indeed survive even with finite non-Hermitian effects (μ=0 and $\gamma =\mathrm{arccosh}(\sqrt{2})$ in Figure 3f–g). Moreover, these winding numbers correctly count the numbers of degenerate Floquet edge mode pairs $n_{0}$ and $n_{\pi }$ at E=0 and E=π under the OBC. That is, we have the bulk–edge correspondence for our 1D chiral-symmetric non-Hermitian Floquet topological insulator as [256]
(146)$$n_{0}=|w_{0}|,\;n_{\pi }=\left|w_{\pi }\right|.$$

Representative examples of the Floquet spectra under the OBC are shown in Figure 3h–j (with μ=0, $r_{y}=\pi /2$, $\gamma =\mathrm{arccosh}(\sqrt{2})$ and the number of unit cells N=150), which provide verifications for the bulk–edge correspondence found in Equation (Equation 146). Note in passing that the system does not show NHSE under the OBC for μ=0, even though the Floquet spectra possess point quasienergy gaps under the PBC. Meanwhile, many pairs of degenerate edge modes at the quasienergies E=0 and E=π are found in the topological phases with large winding numbers. As these topological edge modes could survive in the presence of non-Hermitian effects, they may provide more resources for the realization of robust quantum state transfer and quantum computing schemes in open systems.

The winding numbers of non-Hermitian Floquet topological insulators can be experimentally probed by measuring the stroboscopic time-averaged spin textures [256] or the MCDs [257], as introduced in Section 2.6.1 and Section 2.6.2. Following Ref. [258], we consider a periodically quenched non-Hermitian lattice model, whose Floquet operator in the momentum space reads $U\left(k\right)=e^{-iJ_{2}\sin k\sigma_{y}}e^{-iJ_{1}\cos k\sigma_{x}}$. Here, $J_{1}=u_{1}+iv_{1}$, $J_{2}=u_{2}+iv_{2}$, and $u_{1},u_{2},v_{1},v_{2}\in R$. The winding numbers $\left(w_{0},w_{\pi }\right)$ of U(k) can be obtained using the approach of symmetric time frames (see Equations (Equation 89)–(Equation 93)). Following the procedures outlined in Section 2.6.1 and Section 2.6.2, we obtain the MCDs in Figure 4a (with $v=v_{1}=v_{2}$ and $\left(u_{1},u_{2}\right)=\left(4.5\pi ,0.5\pi \right)$) and the dynamic winding angles of time-averaged spin textures in Figure 4b–e (with $\left(u_{1},u_{2}\right)=\left(4.5\pi ,0.5\pi \right)$ and $v=v_{1}=v_{2}=0.2\pi ,0.35\pi ,0.6\pi ,0.9\pi$). In all the cases, the obtained MCDs and DWNs are coincident with the winding numbers $\left(w_{0},w_{\pi }\right)$ of the Floquet operator U(k) even when there are finite non-Hermitian effects (v≠0). The DWNs and the MCDs thus provide us with two complementary approaches to dynamically characterize 1D non-Hermitian Floquet topological phases with chiral (sublattice) symmetry in momentum and in real spaces, respectively. They can be applied to detect non-Hermitian Floquet matter in different types of experimental setups [215,224].

The models we considered above in this subsection do not possess the NHSE. It remains unclear whether the rich non-Hermitian Floquet topology could coexist with NHSEs, and how to characterize the topological bulk–edge correspondence in the presence of Floquet NHSE. To address this issue, we consider a periodically quenched non-Hermitian SSH model [263], whose *k*-space Hamiltonian under the PBC takes the form of
(147)$$H\left(k,t\right)=\left\{\begin{array}{cc} \left(\mu +J_{1}\cos k+i\lambda \sin k\right)\sigma_{x}, & t\in [\ell T,\ell T+T/2),\\ \left(J_{2}\sin k+i\lambda \cos k\right)\sigma_{y}, & t\in [\ell T+T/2,\ell T+T). \end{array}\right.$$
Here, μ and $J_{1}/2+J_{2}/2$ are the intracell and intercell hopping amplitudes. The non-Hermiticity is introduced by the asymmetric parts of hopping amplitudes ±iλ/2 between the two sublattices. A sketch of the model is shown in Figure 5. The Floquet operator associated with H(k,t) (for ℏ=1 and T=2) reads
(148)$$U\left(k\right)=e^{-i\left(J_{2}\sin k+i\lambda \cos k\right)\sigma_{y}}e^{-i\left(\mu +J_{1}\cos k+i\lambda \sin k\right)\sigma_{x}}.$$

In symmetric time frames, it respects the chiral symmetry $S=\sigma_{z}$ [263]. We may, thus, use the winding numbers $\left(w_{0},w_{\pi }\right)$ to characterize the Floquet topological phases of U(k). The calculation of such winding numbers following Equations (Equation 89)–(Equation 93) leads to the topological phase diagrams in Figure 5a,b. Interestingly, we find that despite Floquet topological insulators with large integer winding numbers, there are also various phases with half-integer-quantized topological invariants. These unique phases are absent in the Hermitian limit of the model. Their appearance is due to the Floquet exceptional topology induced by non-Hermitian effects in the system. The parameter regions which accommodate half-integer topological phases may further show Floquet NHSEs under the OBC. In Figure 5c–f, we compare the dynamic winding angles of U(k) with the winding patterns of its Floquet Hamiltonian vectors in different time frames (with $\left(J_{1},J_{2},\mu ,\lambda \right)=\left(2.4\pi ,0.5\pi ,0.4\pi ,0.25\right)$) [263]. Their coincidence implies that we can use the dynamic winding numbers introduced in Section 2.6.1 to probe the half-integer-quantized topological invariants of 1D non-Hermitian Floquet topological insulators with chiral symmetry.

Under the OBC, the bulk Floquet spectra and the gap-closing points of quasienergies are found to be different from those under the PBC, as shown in Figure 5g (with $\left(J_{2},\mu ,\lambda \right)=\left(0.5\pi ,0.4\pi ,0.25\right)$) [263]. This means that the bulk winding numbers $w_{0}$ (diamonds) and $w_{\pi }$ (pentagrams) of U(k) cannot be used to characterize the edge states and topological phase transitions under the OBC. To overcome this issue, we switch to the OBC winding numbers $\left(\nu_{0},\nu_{\pi }\right)$ (defined as the $\left(W_{0},W_{\pi }\right)$ in Equation (Equation 96)). These winding numbers are found to be integer quantized. They change their values only when the Floquet spectrum of the system closes its gap at E=0 or E=π under the OBC. Furthermore, in each gapped phase, the winding numbers $\left(\nu_{0},\nu_{\pi }\right)$ (circles and crosses in Figure 5g) are related to the numbers of zero and π Floquet edge modes $\left(n_{0},n_{\pi }\right)$ through the bulk–edge correspondence $\left(n_{0},n_{\pi }\right)=2(|\nu_{0}\left|,\right|\nu_{\pi }\left|\right)$. This relation holds even with NHSEs in our system [263]. Therefore, our work provides a dual topological characterization of non-Hermitian Floquet topological insulators under different boundary conditions. In Figure 5h–k, we present examples of the skin-localized bulk states (in Figure 5h,j) and two types of Floquet edge states (in Figure 5i,k) in our system, which are also consistent with the theoretical predictions [263].

To summarize, we found different types of 1D non-Hermitian Floquet topological insulators in a series of studies [256,257,258,260,263]. All the models considered in these studies exhibit rich non-Hermitian Floquet phases with large topological invariants, many topological edge states, and unique topological transitions induced by the interplay between non-Hermitian effects and time-periodic driving fields. The bulk–edge correspondences and dynamic topological characterizations of these intriguing new phases are also established, leading to an explicit and all-round physical picture of 1D non-Hermitian Floquet topological matter. In the following subsections, we uncover the essential role of Floquet engineering in other types of non-Hermitian topological matter.

#### 3.2.2. Second-Order Topological Phase

We next consider the example of a non-Hermitian Floquet second-order topological insulator (SOTI) in two dimensions. An *n*th-order topological insulator in *d* (d≥n) spatial dimensions possesses localized eigenmodes along its (d−n)-dimensional boundaries, which are topologically nontrivial and protected by the symmetries of the *d*-dimensional bulk. For example, an SOTI in two dimensions usually has localized topological states around its zero-dimensional geometric corners. Assisted by time-periodic drivings, degenerate corner modes can further appear at different quasienergies in Floquet second-order topological phases [80,290].

In this subsection, we focus on one typical model of non-Hermitian Floquet SOTI, which is obtained following the procedure of coupled-wire construction [80]. A schematic diagram of the static lattice model is shown in Figure 6a. The system is formed by a stack of SSH chains along the *y*-direction, with dimerized couplings $\left(J_{1},J_{2}\right)$ between adjacent chains. We consider a Floquet variant of the model by applying time-periodic quenches to the coupling and onsite parameters $\left(J_{1},J_{2},\mu \right)$ along the *y*-direction. Under the PBC along both *x*- and *y*-directions, the time-dependent Bloch Hamiltonian of the Floquet model takes the form of
(149)$$H\left(k_{x},k_{y}\right)=H_{x}\left(k_{x}\right)⊗\tau_{0}+\sigma_{0}⊗H_{y}\left(k_{y},t\right),$$
where $\left(k_{x},k_{y}\right)\in \left[-\pi ,\pi \right)\times \left[-\pi ,\pi \right)$ are the quasimomenta. $\sigma_{0}$ and $\tau_{0}$ are two-by-two identity matrices. The Hamiltonians of the 1D subsystems $H_{x}\left(k_{x}\right)$ and $H_{y}\left(k_{y},t\right)$ are explicitly given by
(150)$$H_{x}\left(k_{x}\right)=\left[\left(J-\delta \right)+\left(J+\delta \right)\cos k_{x}\right]\sigma_{x}+\left(J+\delta \right)\sin k_{x}\sigma_{y},$$
(151)$$H_{y}\left(k_{y},t\right)=\left\{\begin{array}{cc} 2J_{1}\cos k_{y}\tau_{x} & t\in [\ell T,\ell T+T/2)\\ 2\left(\mu +J_{2}\sin k_{y}\right)\tau_{z} & t\in [\ell T+T/2,\ell T+T) \end{array}\right..$$
Here, *T* is the driving period and ℓ∈Z. $\sigma_{x,y,z}$ and $\tau_{x,y,z}$ are Pauli matrices acting on the sublattice degrees of freedom in the *x*-and *y*-directions, respectively. Setting ℏ=T=1, J=δ=Δ/2, and choosing μ=u+iv (u,v∈R), the Floquet operator of the system in k-space is found to be [261]
(152)$$U\left(k_{x},k_{y}\right)=e^{-i\Delta (\cos k_{x}\sigma_{x}+\sin k_{x}\sigma_{y})}⊗e^{-ih_{z}\left(k_{y}\right)\tau_{z}}e^{-ih_{x}\left(k_{y}\right)\tau_{x}},$$
where
(153)$$h_{x}\left(k_{y}\right)=J_{1}\cos k_{y},\;h_{z}\left(k_{y}\right)=u+iv+J_{2}\sin k_{y}.$$
The non-Hermitian effect is brought about by the onsite gain and loss encoded in the term $iv\tau_{z}$.

In symmetric time frames, the system has the chiral (sublattice) symmetry $S=\sigma_{z}⊗\tau_{y}$ [261]. One can, thus, characterize its Floquet second-order topological phases by the integer winding numbers [261]
(154)$$\nu_{0}=\frac{\nu_{1}+\nu_{2}}{2},\;\nu_{\pi }=\frac{\nu_{1}-\nu_{2}}{2}.$$
Here, $\nu_{\alpha }={\mathrm{ww}}_{\alpha }$ for α=1,2. *w* is the winding number of the static Hamiltonian $\Delta (\cos k_{x}\sigma_{x}+\sin k_{x}\sigma_{y})$, which is always equal to one for the topological flat-band limit of the SSH model. $\left(w_{1},w_{2}\right)$ are the winding numbers of the subsystem Floquet operator $e^{-ih_{z}\left(k_{y}\right)\tau_{z}}e^{-ih_{x}\left(k_{y}\right)\tau_{x}}$, which can be defined via the procedure outlined in Section 2.5. They both take integer-quantized values, even with the non-Hermitian effects considered in our system [261].

In Figure 6b,c, we show the topological phase diagrams of $U(k_{x},k_{y})$ in the parameter space $J_{1}-v$ (with $\left(\Delta ,J_{2},u\right)=\left(\pi /20,3\pi ,0\right)$). We find that both the winding numbers $\left(\nu_{0},\nu_{\pi }\right)$ can take large values, which indicates the presence of non-Hermitian Floquet SOTI phases with large topological invariants in our setting. Furthermore, with the increase in the gain and loss strength *v*, the system can undergo topological phase transitions and even enter non-Hermitian Floquet SOTI phases with larger winding numbers. These non-Hermiticity-boosted topological signatures are also not expected in non-driven systems. Their appearance is, thus, due to the interaction between Floquet driving fields and non-Hermitian gain and loss.

Under the OBC along both *x*- and *y*-directions, localized Floquet eigenmodes with the quasienergies zero and π appear at the four corners of the square lattice. Some spatial profiles of these corner modes are shown in Figure 6f–j (with $\left(\Delta ,J_{1},J_{2},u,v\right)=\left(\pi /20,2.5\pi ,3\pi ,0,2\right)$). There are twelve (eight) Floquet corner modes at the quasienergy zero (π). More generally, their numbers $\left(n_{0},n_{\pi }\right)$ are related to the winding numbers $\left(\nu_{0},\nu_{\pi }\right)$ through the algebraic relations [261]
(155)$$\left(n_{0},n_{\pi }\right)=4(|\nu_{0}\left|,\right|\nu_{\pi }\left|\right).$$

We, thus, established the bulk–corner correspondence for a class of non-Hermitian Floquet SOTI with chiral symmetry. With the growth in the non-Hermitian parameter *v*, topological phase transitions accompanied by the increase in bulk winding numbers $\left(\nu_{0},\nu_{\pi }\right)$ and corner mode numbers $\left(n_{0},n_{\pi }\right)$ are observed from the gap functions (F0,Fπ)≡(1π|ε|,1π(|Reε|−π)2+(Imε)2) under the OBC (here ε denotes the quasienergy), as showcased in Figure 6d,e. The topological phase transition points $v_{1}$ and $v_{2}$ under the OBC are precisely consistent with the gap-closing points of the 2D bulk under the PBC, which can be found analytically [261]. Finally, even with non-Hermitian effects, the bulk winding numbers $\left(\nu_{0},\nu_{\pi }\right)$ can be dynamically probed by 2D generalizations of the MCD as introduced in Section 2.6.2 (see Ref. [261] for more details).

Overall, we discovered rich non-Hermitian Floquet SOTI phases in a periodically quenched 2D lattice with balanced gain and loss. Different from the static case, each SOTI phase is now depicted by two topological invariants $\left(\nu_{0},\nu_{\pi }\right)$. Many non-Hermitian Floquet SOTIs with large topological invariants and gain- or loss-induced topological phase transitions were identified. Under the OBC, the winding numbers $\left(\nu_{0},\nu_{\pi }\right)$ determine the numbers of symmetry-protected Floquet corner modes at zero and π quasienergies. The interplay between driving and dissipation, thus, results in a series of non-Hermitian Floquet SOTI phases with multiple zero and π corner modes, which may find applications in topological state preparations, information scrambling, and quantum computation. The work of Ref. [261] offers one of the earliest findings of rich non-Hermitian Floquet topological phases beyond one spatial dimension. In the meantime, it introduces a generic scheme to construct non-Hermitian Floquet higher-order topological phases across different physical dimensions, which is expected to be applicable to insulating, superconducting, and semi-metallic systems. Note in passing that some latter studies also considered the NHSEs and anomalous π modes in Floquet higher-order topological phases with somewhat different focuses [237,240,248].

#### 3.2.3. qth-Root Topological Phase

We now consider an exotic class of non-Hermitian Floquet topological phase, which could have symmetry-protected boundary states at the quasienergies pπ/q, with $p,q\in Z^{\pm }$ and p/q≠0,1. A systematic procedure for obtaining these new phases is to take the nontrivial *q*th-root of a propagator $\hat{U}$ that describes Floquet topological matter. The general procedure is developed in Ref. [265], which extends the previous schemes of taking $2^{n}$th and $3^{n}$th roots for static Hamiltonians [291,292,293,294,295,296,297,298,299,300] to Floquet systems. The basic idea is first outlined in Ref. [301], and then expanded by utilizing $Z_{q}$ generalizations of Pauli matrices as ancillary degrees of freedom before taking the *q*th-root of a Floquet operator in an enlarged Hilbert space. This is in parallel with Dirac’s original idea of adding internal degrees of freedom for electrons before taking the square root to obtain their relativistic wave equation [302].

To be explicit, we consider the construction of a cubic-root non-Hermitian Floquet topological insulator. An illustration of the scheme is given in Figure 7. Let us consider a three-step periodically quenched parent system, whose time-periodic Hamiltonian takes the form of [265]
(156)$$\hat{H}\left(t\right)=\left\{\begin{array}{cc} {\hat{H}}_{3}, & t\in [\ell ,\ell +1/3),\\ {\hat{H}}_{2}, & t\in [\ell +1/3,\ell +2/3),\\ {\hat{H}}_{1}, & t\in [\ell +2/3,\ell +1), \end{array}\right.$$
where ℓ∈Z and we have assumed ℏ=T=1. The Floquet operator of the system is then given by $\hat{U}=e^{-\frac{i}{3}{\hat{H}}_{1}}e^{-\frac{i}{3}{\hat{H}}_{2}}e^{-\frac{i}{3}{\hat{H}}_{3}}$. Following the methodology of Ref. [265], the nontrivial cubic root ${\hat{U}}_{1/3}$ of $\hat{U}$ is found to be
(157)$${\hat{U}}_{1/3}=\left(\begin{array}{ccc} 0 & e^{-i\frac{{\hat{H}}_{1}}{3}} & 0\\ 0 & 0 & e^{-i\frac{{\hat{H}}_{2}}{3}}\\ e^{-i\frac{{\hat{H}}_{3}}{3}} & 0 & 0 \end{array}\right).$$
The cube of ${\hat{U}}_{1/3}$ then gives
(158)$${\hat{U}}_{1/3}^{3}=\left(\begin{array}{ccc} e^{-\frac{i}{3}{\hat{H}}_{1}}e^{-\frac{i}{3}{\hat{H}}_{2}}e^{-\frac{i}{3}{\hat{H}}_{3}} & 0 & 0\\ 0 & e^{-\frac{i}{3}{\hat{H}}_{2}}e^{-\frac{i}{3}{\hat{H}}_{3}}e^{-\frac{i}{3}{\hat{H}}_{1}} & 0\\ 0 & 0 & e^{-\frac{i}{3}{\hat{H}}_{3}}e^{-\frac{i}{3}{\hat{H}}_{1}}e^{-\frac{i}{3}{\hat{H}}_{2}} \end{array}\right).$$
It is clear that ${\hat{U}}_{1/3}^{3}$ contains three identical copies of $\hat{U}$, in the sense that its three diagonal blocks are all related by similarity transformations and, thus, share the same spectrum and topological properties with $\hat{U}$ [265].

As an example, we choose the Floquet model introduced in Ref. [263] as the parent system, whose Hamiltonian can now be expressed as
(159)$$\hat{H}\left(t\right)=\left\{\begin{array}{cc} \frac{3}{4}{\hat{H}}_{1} & t\in [\ell ,\ell +1/3)\\ \frac{3}{2}{\hat{H}}_{2} & t\in [\ell +1/3,\ell +2/3)\\ \frac{3}{4}{\hat{H}}_{1} & t\in [\ell +2/3,\ell +1) \end{array}\right.,$$
where
(160)$${\hat{H}}_{1}=J_{2}\sum_{n}\left(i{\hat{c}}_{n+1}^{†}{\hat{c}}_{n}+H.c.\right)⊗\sigma_{y}+i\lambda \sum_{n}\left({\hat{c}}_{n}^{†}{\hat{c}}_{n+1}+H.c.\right)⊗\sigma_{y},$$
(161)$${\hat{H}}_{2}=\sum_{n}\left(\mu {\hat{c}}_{n}^{†}{\hat{c}}_{n}+J_{1}{\hat{c}}_{n}^{†}{\hat{c}}_{n+1}+H.c.\right)⊗\sigma_{x}+i\lambda \sum_{n}\left(i{\hat{c}}_{n+1}^{†}{\hat{c}}_{n}+H.c.\right)⊗\sigma_{x}.$$

The cubic-root Floquet operator ${\hat{U}}_{1/3}$ of the system then takes the form of Equation (Equation 157). The non-Hermitian effect is introduced by the asymmetric hopping amplitude iλ. The Floquet quasienergy spectrum and gap functions of ${\hat{U}}_{1/3}$ versus $J_{1}$ under the PBC (gray dots) and OBC (blue dots, red and blue lines) are shown in Figure 7a,b (with $\left(J_{2},\mu ,\lambda \right)=\left(0.5\pi ,0.4\pi ,0.25\right)$ and the length of lattice L=400). Here, the gap function
(162)Fε≡(ReE−ε)2+(ImE)2,
and the blue (red) lines in Figure 7b denote $F_{\pi /3}$ ($F_{2\pi /3}$). We observe a series of phase transitions accompanied by quasienergy-gap closings at E=pπ/q for p=0,1,2,3 and q=3. After these transitions, more and more degenerate Floquet edge modes emerge at the fractional quasienergies E=π/3 and 2π/3, which are impossible in conventional Floquet topological insulators with two quasienergy bands in one dimension. These fractional-quasienergy edge modes are, thus, unique to cubic-root Floquet topological insulators, either Hermitian or non-Hermitian. A zoom in on their quasienergies for one exemplary case is shown in Figure 7c (with $\left(J_{1},J_{2},\mu ,\lambda \right)=\left(\pi ,0.5\pi ,0.4\pi ,0.25\right)$ and the length of lattice L=400), and their spatial profiles are shown in Figure 7d (for the E=π/3 eigenmodes) and Figure 7e (for the E=2π/3 eigenmodes). Moreover, the numbers of these fractional-quasienergy edge modes $\left(n_{\pi /3},n_{2\pi /3}\right)$ are determined by the open-bulk winding numbers $\left(\nu_{0},\nu_{\pi }\right)$ [263] of the parent system $\hat{U}=e^{-i\frac{{\hat{H}}_{1}}{4}}e^{-i\frac{{\hat{H}}_{2}}{2}}e^{-i\frac{{\hat{H}}_{1}}{4}}$. More precisely, we have the following bulk–edge correspondence for our cubic-root non-Hermitian Floquet topological insulator ${\hat{U}}_{1/3}$, i.e., [265]
(163)$$n_{0}=n_{2\pi /3}=2|\nu_{0}|,\;n_{\pi /3}=n_{\pi }=2\left|\nu_{\pi }\right|,$$
where $n_{0}$ and $n_{\pi }$ are the numbers of zero and π degenerate Floquet edge modes. Finally, we notice that the system also possesses NHSE under the OBC, as reflected by the profiles of its Floquet bulk states in Figure 7f (with $\left(J_{1},J_{2},\mu ,\lambda \right)=\left(\pi ,0.5\pi ,0.4\pi ,0.25\right)$ and the length of lattice L=400). In Figure 7a,b, we also notice the discrepancy between the gap-closing points of the system under the PBC and OBC. Nevertheless, the open-bulk winding numbers $\left(\nu_{0},\nu_{\pi }\right)$ correctly capture the bulk–edge correspondence of the system under the OBC, even with NHSEs. Therefore, the work conducted in Ref. [265] also established a dual topological characterization for *q*th-root non-Hermitian Floquet topological insulators. The formalism developed in Ref. [265] is equally applicable to the construction of *q*th-root Floquet topological insulators, superconductors, and semi-metals in other (non-)Hermitian systems and across different physical dimensions. The fractional quasienergy edge modes at E=pπ/q might also be employed to generate boundary time crystals with different temporal periodicity and topological properties [99]. They may also find applications in Floquet quantum computing.

### 3.3. Non-Hermitian Floquet Topological Superconductors

Similar to the static case, non-Hermitian Floquet topological phases could also appear in superconducting systems. We review one such example in this subsection, which admits many Floquet Majorana zero and π edge modes, even with non-Hermitian effects.

The model we are going to consider describes a Floquet Kitaev chain, whose superconducting pairing terms are subject to time-periodic kicks. The Hamiltonian of the model takes the form of
(164)$$\hat{H}\left(t\right)=\frac{1}{2}\sum_{n}\left[J\left({\hat{c}}_{n}^{†}{\hat{c}}_{n+1}+H.c.\right)+\Delta \delta_{T}\left(t\right)\left({\hat{c}}_{n}{\hat{c}}_{n+1}+H.c.\right)+\mu \left(2{\hat{c}}_{n}^{†}{\hat{c}}_{n}-1\right)\right],$$
where *J* is the nearest-neighbor hopping amplitude, Δ is the p-wave superconducting pairing amplitude, and μ is the chemical potential. $\delta_{T}\left(t\right)\equiv \sum_{\ell \in Z}\delta \left(t/T-\ell \right)$ implements delta kickings periodically with the period *T*. The system is made non-Hermitian by setting $J=J_{r}+iJ_{i}$ or $\mu =\mu_{r}+i\mu_{i}$ with $J_{i}\neq 0$ or $\mu_{i}\neq 0$, respectively. Possible experimental realizations of the delta kickings and non-Hermitian terms in this model are discussed in Ref. [259].

Under the PBC and in symmetric time frames, the Floquet operator of $\hat{H}\left(t\right)$ respects the chiral (sublattice) symmetry $S=\sigma_{x}$. Its bulk topological phases can then be characterized by the winding numbers $\left(w_{0},w_{\pi }\right)$ (see Equations (Equation 89)–(Equation 93)), for which we choose
(165)$${\hat{H}}_{0}=\frac{1}{2}\sum_{n}\left[J\left({\hat{c}}_{n}^{†}{\hat{c}}_{n+1}+H.c.\right)+\mu \left(2{\hat{c}}_{n}^{†}{\hat{c}}_{n}-1\right)\right],$$
(166)$${\hat{H}}_{1}=\frac{1}{2}\sum_{n}\Delta \left({\hat{c}}_{n}{\hat{c}}_{n+1}+H.c.\right),$$
and assume ℏ=T=1. Typical topological phase diagrams of the system are shown in Figure 8a (for J∈R, μ∈C, $\left(\mu_{r},\Delta \right)=\left(0.3\pi ,0.5\pi \right)$) and Figure 8b (for μ∈R, J∈C, (μ,Δ)=(0.4π,0.9π)). Different uniformly colored regions correspond to different topological phases and the winding numbers $\left(w_{0},w_{\pi }\right)$ of each phase are labeled therein. The boundaries between different phases can be analytically determined from the gap-closing conditions of the system (see Ref. [259]). Two notable features are brought about by the interplay between periodic drivings and non-Hermitian effects. First, the system could undergo rich topological phase transitions and enter Floquet superconducting phases with arbitrarily large topological winding numbers in principle. These phases could support arbitrarily many zero and π Majorana edge modes in the thermodynamic limit, which might be employed to engineer boundary time crystals and realize Floquet quantum computing protocols. Second, Floquet superconducting phases with larger winding numbers and, therefore, stronger topological signatures may appear with the increase in the non-Hermitian parameter. Therefore, new topological states could be induced and stabilized solely by non-Hermitian effects in Floquet superconducting systems. This is not expected in the non-driven counterparts of our model, where nontrivial topological properties are usually destroyed with the growth of non-Hermitian effects.

Under the OBC, the quasienergy spectrum and edge states of the system can be obtained by diagonalizing the Floquet operator $\hat{U}=e^{-i{\hat{H}}_{0}}e^{-i{\hat{H}}_{1}}$ in Majorana or BdG bases [259]. As examples, the real parts of Floquet spectra and gap functions $\left(F_{0},F_{\pi }\right)$ versus $\mu_{i}$ (with $\left(J,\Delta ,\mu_{r}\right)=\left(4\pi ,0.5\pi ,0.3\pi \right)$) and $J_{r}$ (with $\left(J_{i},\Delta ,\mu \right)=\left(1,0.4\pi ,0.9\pi \right)$) are shown in Figure 8c,e, and Figure 8d,f, respectively. $y_{1}∼y_{8}$ refer to gap-closing points of the Floquet spectrum under the PBC (see Table 1). The gap functions are defined as
(167)F0=1π(Reε)2+(Imε)2,Fπ=1π(|Reε|−π)2+(Imε)2,
where ε includes all the quasienergies of $\hat{U}$ under the OBC. The eigenmodes with $F_{0}=0$ and $F_{\pi }=0$ thus have the quasienergies zero and π, respectively. In Table 1, we list the numbers of Floquet Majorana edge modes $\left(n_{0},n_{\pi }\right)$ with the quasienergies (0,π) and the bulk topological winding numbers $\left(w_{0},w_{\pi }\right)$. A simple bulk–edge correspondence can be further found between these numbers, i.e.,
(168)$$n_{0}=2|w_{0}|,\;n_{\pi }=2\left|w_{\pi }\right|.$$
These relations hold in other parameter regions of our model as well. We conclude that our non-Hermitian Floquet Kitaev chain could indeed possess many Majorana zero and π edge modes due to its large winding numbers. The localization nature of these Majorana modes was also demonstrated in Ref. [259].

Therefore, our study in Ref. [259] established the topological characterization and bulk–edge correspondence of 1D non-Hermitian Floquet topological superconductors that belong to an extended BDI symmetry class. The interaction between time-periodic driving fields and non-Hermitian effects was found to produce rich Floquet superconducting phases with large topological winding numbers and many Majorana edge modes at two distinct quasienergies zero and π. These non-Hermitian Floquet Majorana modes might allow Floquet quantum computing schemes to be more robust to environmental-induced nonreciprocity, dissipation, and quasiparticle poisoning effects. The existence of many pairs of Floquet Majorana modes may also create stronger transport signals at the ends of the chain, making it easier to experimentally detect their topological properties in open-system settings [228]. In future studies, topics like non-Hermitian Floquet topological superconductors under different driving protocols, in other symmetry classes, with more complicated lattice effects (e.g., sublattice structures, long-range hoppings or disorder), in higher-spatial dimensions and with NHSEs, deserve to be explored. Possible changes in topological classifications due to many-body effects in non-Hermitian Floquet superconductors are also awaiting to be revealed.

### 3.4. Non-Hermitian Floquet Quasicrystals

In this subsection, we go beyond the clean limit of driven non-Hermitian lattices and showcase that the interplay among correlated disorder, temporal driving, and non-Hermitian effects could yield Floquet quasicrystals with rich PT-symmetry-breaking transitions, localization transitions, and topological phase transitions. We will consider systems under both high-frequency and near-resonant driving fields.

We start with the example of a 1D non-Hermitian quasicrystal under high-frequency harmonic driving forces. The Hamiltonian of the model takes the form of [264]
(169)$$\hat{H}\left(t\right)=\sum_{n}\left\{J\left(e^{-\gamma }{\hat{c}}_{n}^{†}{\hat{c}}_{n+1}+e^{\gamma }{\hat{c}}_{n+1}^{†}{\hat{c}}_{n}\right)+\left[V\cos\left(2\pi \alpha n\right)-nK\cos\left(\omega t\right)\right]{\hat{c}}_{n}^{†}{\hat{c}}_{n}\right\}.$$
An illustration of the model is given in Figure 9a. Here, γ∈R and $Je^{-\gamma }$ ($Je^{\gamma }$) describes the right-to-left (left-to-right) nearest-neighbor hopping amplitude. The hopping is nonreciprocal and thus $\hat{H}\left(t\right)\neq {\hat{H}}^{†}\left(t\right)$ if γ≠0. ${\hat{c}}_{n}^{†}$ (${\hat{c}}_{n}$) creates (annihilates) a particle on the lattice site *n*. V∈R is the amplitude of an onsite potential, and α is chosen to be irrational in order for the potential to be spatially quasiperiodic. K∈R is the driving amplitude and ω is the driving frequency. This model can be viewed as a Floquet and quasicrystal variant of the Hatano–Nelson model [185]. In the absence of the driving force and under the PBC, all the eigenstates of the system are extended (localized) with complex (real) eigenvalues if $\left|V|<|2J\right|e^{\left|\gamma \right|}$ ($\left|V|>|2J\right|e^{\left|\gamma \right|}$) [191]. The system could, thus, undergo a PT transition, a localization–delocalization transition, and also a topological transition accompanied by the quantized change in its spectral winding number (Equation (Equation 81)) at $\left|V\right|=\left|2J\right|e^{\left|\gamma \right|}$ [191].

Transforming the Hamiltonian $\hat{H}\left(t\right)$ in Equation (Equation 169) to a rotating frame and applying the method discussed in Section 2.3, we can obtain the Floquet effective Hamiltonian of our system in the high-frequency limit [264], i.e.,
(170)$${\hat{H}}_{F}=\sum_{n}\left[J_{0}\left(\frac{K}{\omega }\right)J\left(e^{-\gamma }{\hat{c}}_{n}^{†}{\hat{c}}_{n+1}+e^{\gamma }{\hat{c}}_{n+1}^{†}{\hat{c}}_{n}\right)+V\cos\left(2\pi \alpha n\right){\hat{c}}_{n}^{†}{\hat{c}}_{n}\right].$$
Here, $J_{0}\left(K/\omega \right)$ is the Bessel function of first kind, which is a non-monotonous function of K/ω. Under the PBC, the Floquet NHQC described by ${\hat{H}}_{F}$ could, thus, undergo multiple and re-entrant PT, localization and topological transitions with the change in the ratio K/ω between the driving amplitude and driving frequency whenever [264]
(171)$$|V|=|2JJ_{0}\left(K/\omega \right)|e^{\left|\gamma \right|}.$$
This is indeed the case, as demonstrated by the maximum of the imaginary parts of quasienergies, the minimum of IPRs, and the winding numbers in Figure 9b–d (for $\left(J,V,\alpha \right)=\left(1,1,\frac{\sqrt{5}-1}{2}\right)$ and the length of lattice L=610). In the metallic phase, we have max|ImE|>0, min(IPR)→0, and the winding number w=−1, which means that all the Floquet eigenstates of ${\hat{H}}_{F}$ are extended with complex quasienergies. In the insulating phase, we have max|ImE|=0, min(IPR)>0, and the winding number w=0, implying that all the Floquet eigenstates of ${\hat{H}}_{F}$ are localized with real quasienergies. The boundaries between these phases (red dashed lines in Figure 9b–d) are precisely depicted by Equation (Equation 171). The Lyapunov exponent $\lambda =\ln|\left(Ve^{-|\gamma |}\right)/\left[2JJ_{0}\left(K/\omega \right)\right]|$ is quasienergy-independent for all the Floquet eigenstates, which is positive (negative and thus ill-defined) in the localized (extended) phases [264]. Therefore, it is clear that the phases and transitions in the quasicrystal Hatano–Nelson model can be significantly modified by Floquet driving fields even in the high-frequency limit. The periodic driving also provides us with a flexible knob to control and engineer different types of phase transitions in NHQCs, with further examples discussed in Ref. [264].

We move on to the example of an NHQC under near-resonant driving fields. In this case, the interplay between driving and non-Hermitian effects not only induces multiple and re-entrant localization transitions but also generates critical mobility edge phases that are absent in non-driven limits. A schematic diagram of our model is shown in Figure 10a. Its Hamiltonian takes the form of $\hat{H}\left(t\right)=\hat{K}$ for $t\in [\ell T,\ell T+T_{1})$ and $\hat{H}\left(t\right)=\hat{V}$ for $t\in [\ell T+T_{1},\ell T+T_{1}+T_{2})$, where ℓ∈Z and the driving period $T=T_{1}+T_{2}$. The system is piecewisely quenched between $\hat{K}=J\sum \left(e^{\gamma }{\hat{c}}_{n}^{†}{\hat{c}}_{n+1}+e^{-\gamma }{\hat{c}}_{n+1}^{†}{\hat{c}}_{n}\right)$ and $\hat{V}=V\sum_{n}\cos\left(2\pi \alpha n\right){\hat{c}}_{n}^{†}{\hat{c}}_{n}$ over each driving period. For an irrational α, we thus arrive at a periodically quenched, spatially quasiperiodic variant of the Hatano–Nelson model, whose Floquet operator is given by [266]
(172)$$\hat{U}=e^{-iV\sum_{n}\cos\left(2\pi \alpha n\right){\hat{c}}_{n}^{†}{\hat{c}}_{n}}e^{-iJ\sum (e^{\gamma }{\hat{c}}_{n}^{†}{\hat{c}}_{n+1}+e^{-\gamma }{\hat{c}}_{n+1}^{†}{\hat{c}}_{n})},$$
where $V=VT_{2}/\hbar$ and $J=JT_{1}/\hbar$. Solving the eigenvalue equation $\hat{U}|\psi \rangle =e^{-iE}|\psi \rangle$ and using the tools introduced in Section 2.7, we could obtain the maximal imaginary parts of quasienergies max|ImE|, the density of states with complex eigenvalues ρ, the minimum of IPRs and the spreading velocity of an initially localized wavepacket *v*. A collection of these quantities versus the strength of quasiperiodic potential *V* is shown in Figure 10b for a typical case (with $\left(J,\gamma ,\alpha \right)=\left(\pi /6,0.8,\frac{\sqrt{5}-1}{2}\right)$ and the length of lattice L=4181). We observe that with the increase of *V* from V=0, the system first undergoes a complex-to-real PT transition in its quasienergy spectrum, which is also accompanied by a localization transition of all its Floquet eigenstates from spatially extended to localized. Interestingly, with the further enhancement of the quasiperiodic potential *V* (thus with stronger correlated disorder), some localized Floquet states can again become extended with real eigenvalues. The system then enters a critical phase, in which extended and localized eigenstates coexist and are separated by mobility edges on the complex quasienergy plane (see also Figure 3 in Ref. [266]). The further increase in *V* leads to re-entrant transitions between localized and critical mobility edge phases in the system. This is also reflected by the two-parameter phase diagrams in Figure 10c–f (see Section 2.3 for the definitions of ρ, $\bar{g}$, ${\mathrm{IPR}}_{\min}$, and ζ). Note that both the critical phases and the re-entrant localization transitions are absent in the non-driven Hatano–Nelson quasicrystal. They are brought about by the nearest-resonant Floquet driving field. It induces long-range spatial couplings and quasienergy windings in the system, thus yielding the observed phenomena (see Ref. [266] for more detailed discussions).

In experiments, Floquet NHQCs might be realized by ultracold atoms in driven and quasiperiodic optical superlattices with particle losses [181,225]. Static NHQCs have also been realized by non-unitary photonic quantum walks [193,194]. Signatures of PT-breaking transitions, localization transitions, mobility edges, and topological properties related to NHSEs can be extracted from dynamical observables of initially localized wavepackets [193,194]. Since quantum walk models are intrinsically dynamical, they provide natural platforms to realize and detect the Floquet NHQCs reviewed in this subsection. Meanwhile, as the discrete-time quantum walk carries a synthetic spin-half degree of freedom, it may also be utilized to simulate other types of NHQCs, such as those with lattice dimerizations or non-Abelian potentials [303,304,305]. Topological Anderson insulators induced by uncorrelated disorder could also be explored in similar settings [189].

Overall, we find that both high-frequency and near-resonant driving fields could be used to trigger, control, and enrich the phases and transitions in NHQCs, and even create unique Floquet NHQC phases that are absent in the static limit [264,266]. Non-Hermitian disordered systems, thus, provide further playgrounds for the exploration of new physics that are enabled by Floquet engineering and time-periodic driving fields.

## 4. Conclusions and Outlook

In this review, we recapitulated some progress we made in the study of intriguing topological phases in non-Hermitian Floquet systems. After a general introduction about the background and motivation, we first formulated the theoretical frameworks underpinning our study. These include the Floquet theory applicable in general situations and some approximation schemes suitable for exploring fast and slowly driven systems. Efficient means of characterizing the symmetry, topology, dynamical, and localization nature of non-Hermitian Floquet systems were then introduced. Equipped with these tools, we discussed prototypical examples of non-Hermitian Floquet topological phases in insulating, superconducting, and quasicrystalline systems, where the PT transitions, topological transitions, bulk-edge/corner correspondences and localization transitions within these non-Hermitian Floquet phases of matter were systematically characterized. Therefore, the collection of our works [256,257,258,259,260,261,262,263,264,265,266] lays a solid foundation for the study of topological phenomena in non-Hermitian Floquet systems and further uncovers the richness of topological phases that could emerge due to the interplay between periodic drivings and non-Hermitian effects.

There are several common themes behind the models explored in our reviewed works. First, all the non-Hermitian Floquet topological insulator and superconductor models we reviewed possess chiral (sublattice) symmetries, which offer topological protections for their degenerate Floquet edge or corner modes at different real quasienergies even in the presence of gain/loss, nonreciprocity, or NHSEs. The symmetry also allows us to treat the topological characterization of these different models within a coherent framework. Second, for all the models we considered here, the interplay between non-Hermitian effects and time-periodic drivings results in phases and transitions that appear to be much richer than those existing in the Hermitian or static limits of our models, including the Floquet NHSEs and half-integer-quantized winding numbers that are unfeasible in these limits. Third, in all the non-Hermitian Floquet topological insulator and superconductor models considered here, we obtain phases with large topological invariants, many real-quasienergy topological edge modes, as well as multiple and re-entrant PT/localization transitions in Floquet non-Hermitian quasicrystals. These phenomena arise thanks to the long-range couplings induced by Floquet driving fields. We also expect the second and third points to be generic for non-Hermitian Floquet systems under near-resonant and strong drivings, and the resulting phases and transitions could go beyond the non-driven limits of the related non-Hermitian systems.

We note that some other schemes concerning the topological classification and bulk–edge correspondence of non-Hermitian Floquet matter were discussed in Refs. [226,230,232,238,247]. The EPs and gapless topological phases in non-Hermitian Floquet systems were also considered in Refs. [217,233,236,239,241,246,249]. From the perspective of quantum control, the periodic driving fields may provide efficient means to stabilize non-Hermitian systems and control the PT transitions therein [212]. Some interesting aspects regarding the quantum dynamics and anomalous diffusion were revealed in several chaotic non-Hermitian Floquet systems [229,235,244,250]. Aspects of PT symmetry in non-Hermitian Floquet phases were also investigated in Refs. [211,218,219,223,234]. On the experimental side, the quantum walks in cold atoms, photonic setups, and circuit QED provide useful frameworks to realize and detect non-Hermitian Floquet topological matter [193,194,210,213,214,215,216,224,225,243]. Coupled resonators offer another class of setups to explore non-Hermitian Floquet topological matter, in which transient and anomalous Floquet NHSEs [182,183], Floquet PT symmetry [218], Floquet EPs and even non-Bloch Floquet band structures [306] have been experimentally studied.

Even with all the progress mentioned above, our knowledge about non-Hermitian Floquet topological matter is still quite limited. This is especially the case for systems beyond one spatial dimension, with uncorrelated disorders, and with many-body interactions. Dynamical quantum phase transitions may exhibit unique critical signatures triggered by the non-Hermitian Floquet exceptional topology [307,308,309,310,311]. The entanglement and transport properties of non-Hermitian Floquet systems [312,313] may also deviate significantly from their Hermitian or static counterparts. The exact connection between the topological phases of non-Hermitian Floquet Hamiltonians and the Floquet master equation (or Floquet Liouvillian) of driven open systems is unclear. More experimental efforts deserve to be made in order to realize and observe non-Hermitian Floquet phases with large topological invariants, many topological edge states, and multiple topological phase transitions. All these facts indicate that the area of non-Hermitian Floquet matter is still in its infancy, and further substantial developments are eagerly needed in this area.

## Figures and Tables

**Figure 1 entropy-25-01401-f001:**
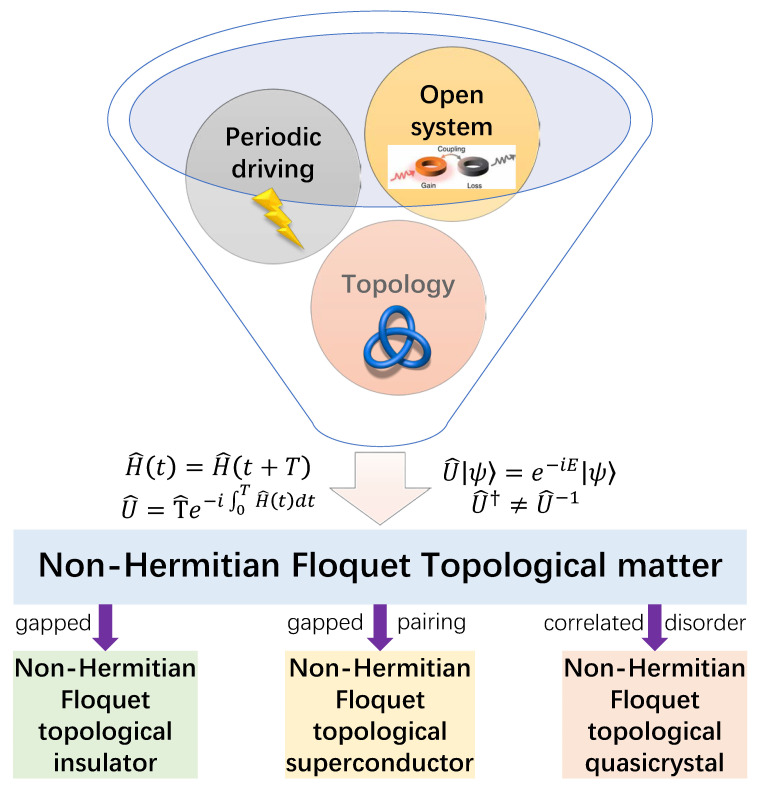
Non-Hermitian Floquet topological matter: a schematic diagram to illustrate the concepts and definitions of several typical phases.

**Figure 2 entropy-25-01401-f002:**
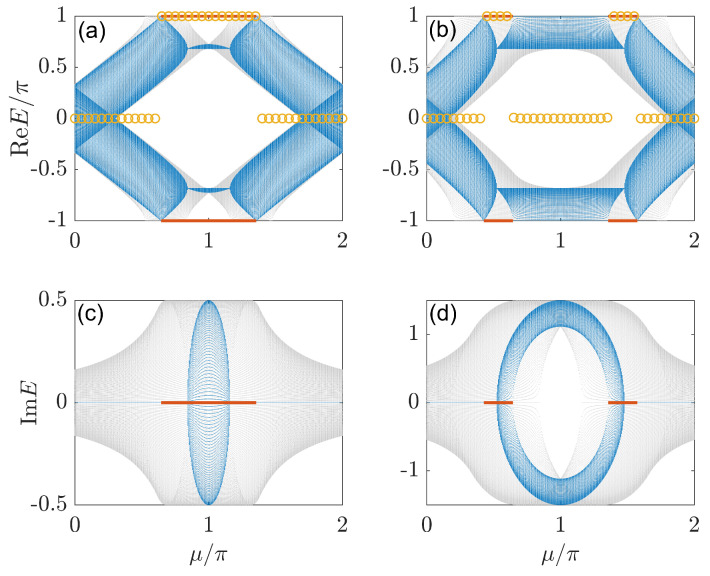
Floquet spectra and winding numbers of the harmonically driven 1D non-Hermitian lattice model. The gray dots at each μ denote the real (in (**a**,**b**)) and imaginary (in (**c**,**d**)) parts of quasienergies obtained under the PBC in the conventional BZ. The blue dots at each μ denote the real (in (**a**,**b**)) and imaginary (in (**c**,**d**)) parts of quasienergies obtained under the OBC in the GBZ. The yellow circles denote the winding number *W*, which is equal to 1 (0) in the topological (trivial) phases with (without) anomalous Floquet edge modes at the quasienergies ±π, which are highlighted by the red solid lines. Other system parameters are chosen as (ω,γ)=(2π,1) ((ω,γ)=(2π,3)) for the panels (**a**,**c**) (**b**,**d**).

**Figure 3 entropy-25-01401-f003:**
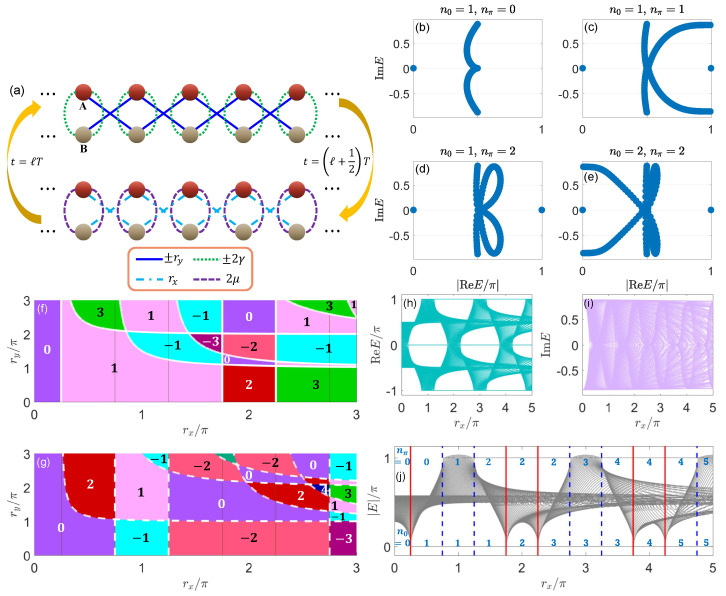
Non-Hermitian Floquet topological insulator [256]: (**a**) Schematic diagram of the lattice model. (**b**–**e**) Floquet spectrum at different $r_{x}$ with $\left(\mu ,r_{y},\gamma \right)=\left(0,\pi /2,\mathrm{arccosh}\left(\sqrt{2}\right)\right)$ under the OBC. The dots at ImE=0 and |Re*E*|=0 (|Re*E*|=π) denote edge modes with E=0 (E=π). Other dots denote bulk states. (**b**) $r_{x}=\pi /2$, with one pair of edge modes at E=0. (**c**) $r_{x}=\pi$, with one (one) pair of edge modes at E=0 (E=π). (**d**) $r_{x}=3\pi /2$, with one pair (two pairs) of edge modes at E=0 (E=π). (**e**) $r_{x}=2\pi$, with two (two) pairs of edge modes at E=0 (E=π). (**f**,**g**) Winding numbers $w_{0}$ (**f**) and $w_{\pi }$ (**g**) in the domain $\left(r_{x},r_{y}\right)\in \left(0,3\pi \right)\times \left(0,3\pi \right)$, with μ=0 and $\gamma =\mathrm{arccosh}(\sqrt{2})$. The regions with the same color have the same values of $w_{0}$ and $w_{\pi }$, as denoted by the integers in (**f**,**g**). The white solid/dashed lines are phase boundaries related to gap closings at E=0/π and topological phase transitions. (**h**–**j**) Floquet spectrum versus $r_{x}$ under OBC, with $\left(\mu ,r_{y},\gamma \right)=\left(0,\pi /2,\mathrm{arccosh}\left(\sqrt{2}\right)\right)$; (**h**,**i**) show the real and imaginary parts of *E*; (**j**) shows the absolute values of *E*. The red solid (blue dashed) lines denote boundaries where the gap close at E=0 (π). The integers in light blue are the numbers of edge mode pairs at E=0 and π.

**Figure 4 entropy-25-01401-f004:**
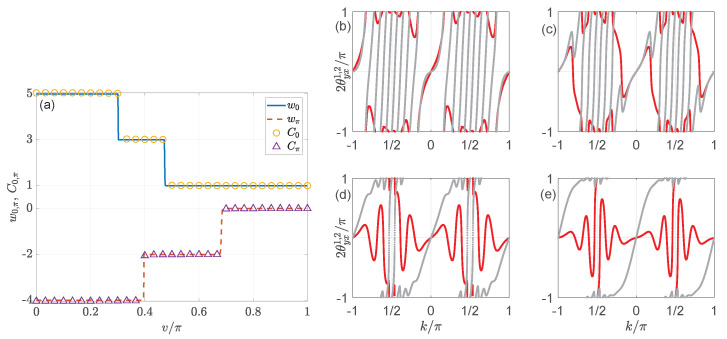
Dynamical characterization of non-Hermitian Floquet topological phases [258]: (**a**) Winding numbers ($w_{0}$, $w_{\pi }$) and MCDs ($C_{0}$, $C_{\pi }$) versus the imaginary part of hopping amplitudes $J_{1}=u_{1}+iv$ and $J_{2}=u_{2}+iv$, with $\left(u_{1},u_{2}\right)=\left(4.5\pi ,0.5\pi \right)$. The results for ($C_{0}$,$C_{\pi }$) are averaged over 50 driving periods. (**b**–**e**) Winding angles $\theta_{yx}^{1}$ (red dots) and $\theta_{yx}^{2}$ (gray dots) of the time-averaged spin textures versus the quasimomentum *k*, with $\left(u_{1},u_{2}\right)=\left(4.5\pi ,0.5\pi \right)$. The imaginary parts of hopping amplitudes are $v_{1}=v_{2}=v$ with v=0.2π for (**b**), v=0.35π for (**c**), v=0.6π for (**d**), and v=0.9π for (**e**). The DWNs $\left(\nu_{1},\nu_{2}\right)$, derived from the winding angles around the first BZ are (1,9) for (**b**), (−1,7) for (**c**), (−1,3) for (**d**) and (1,1) for (**e**), yielding $\left(\frac{\nu_{1}+\nu_{2}}{2},\frac{\nu_{1}-\nu_{2}}{2}\right)=\left(5,-4\right),\left(3,-4\right),\left(1,-2\right),\left(1,0\right)$ for (**b**–**e**), consistent with the $\left(w_{0},w_{\pi }\right)$ in (**a**) at the corresponding system parameters.

**Figure 5 entropy-25-01401-f005:**
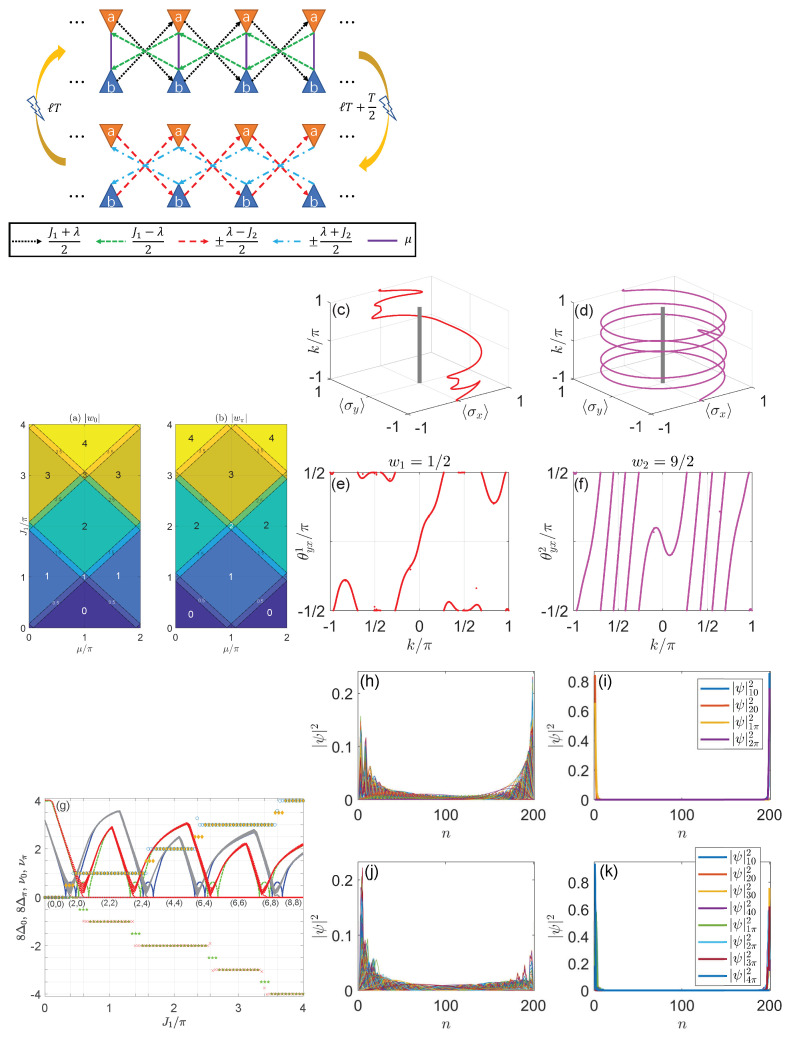
Non-Hermitian FTI with NHSE [263]: Top panel shows a sketch of the driven lattice model, with a and b denoting two different sublattices. (**a**,**b**) Topological phase diagrams vs μ and $J_{1}$ under PBCs. Other parameters are $\left(J_{2},\lambda \right)=\left(0.5\pi ,0.25\right)$. Each region with a uniform color denotes a topological phase characterized by $w_{0}$ and $w_{\pi }$, with their values denoted in (**a**,**b**). (**c**,**f**) Spin textures and dynamic winding angles in time frames α=1 (**c**,**e**) and α=2 (**d**,**f**). System parameters are $\left(J_{1},J_{2},\mu ,\lambda \right)=\left(2.4\pi ,0.5\pi ,0.4\pi ,0.25\right)$. An average over 500 driving periods is made to obtain winding angles $\theta_{yx}^{1,2}$ in (**e**,**f**). In (**c**,**d**), red (magenta) points denote ($\langle \sigma_{x}\rangle ,\langle \sigma_{y}\rangle$) in the first (second) time frame. Gray solid lines denote the origin of $\langle \sigma_{x}\rangle -\langle \sigma_{y}\rangle$ plane. In (**e**,**f**), red (magenta) points denote dynamic winding angles $\theta_{yx}^{1}$ ($\theta_{yx}^{2}$) in time frame 1 (2). (**g**) Gap functions $\Delta_{0}$ (blue and gray solid lines for PBC and OBC), $\Delta_{\pi }$ (green and red dotted lines for PBC and OBC), open bulk winding numbers $\nu_{0}$ (circles), $\nu_{\pi }$ (crosses), and PBC winding numbers $w_{0}$ (diamonds), $w_{\pi }$ (pentagrams). Other parameters are $\left(\mu ,J_{2},\lambda \right)=\left(0.4\pi ,0.5\pi ,0.25\right)$. Only 20 smallest values of $\left(\Delta_{0},\Delta_{\pi }\right)$ under OBC are shown for illustrations. The numbers of zero and π edge modes are given below the *x* axis. (**h**–**k**) Profiles of bulk modes (**h**,**j**) and edge modes (**i**,**k**). System parameters are $\left(J_{1},J_{2},\mu ,\lambda \right)=\left(\pi ,0,5\pi ,0.4\pi ,0.25\right)$ for (**h**,**i**) and (2π,0.5π,0.4π,0.25) for (**j**,**k**). *n* is the unit-cell index with 200 cells in total. In (**h**,**j**), NHSE is found as the accumulation of bulk modes around edges. In (**i**,**k**), one pair (two pairs) of edge modes at E=0 and π are coexistent with bulk skin modes.

**Figure 6 entropy-25-01401-f006:**
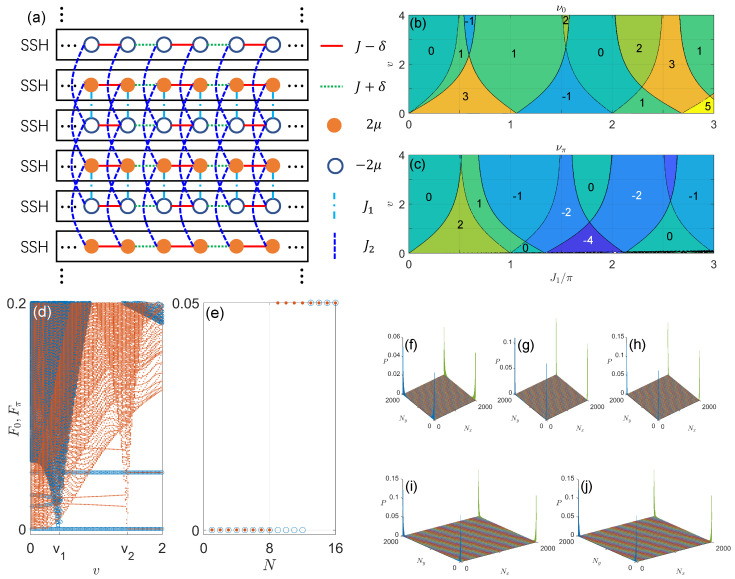
Non-Hermitian Floquet SOTI [261]: (**a**) Schematic diagram of the lattice model. (**b**,**c**) Topological phase diagram of the model in Equation (Equation 149) versus the hopping amplitude $J_{1}$ and gain/loss amplitude *v*. Other system parameters are Δ=π/20, $J_{2}=3\pi$, and u=0. The topological invariants $\nu_{0}$ and $\nu_{\pi }$ for each non-Hermitian Floquet SOTI phase with a uniform color are given in (**b**,**c**). Spectral gap functions $F_{0}$ (blue circles) and $F_{\pi }$ (red dots) versus *v* and the state index *N* are shown in (**d**,**e**). The system parameters are Δ=π/20, $J_{1}=2.5\pi$, $J_{2}=3\pi$, and u=0 for (**d**), with further v=2 for (**e**). $v_{1}$ and $v_{2}$ denote topological phase transition points. In (**e**), the system possesses twelve (eight) Floquet topological corner modes at the quasienergies zero (π). The spatial probability distributions of these corner modes are given in (**f**–**h**) and (**i**,**j**), respectively.

**Figure 7 entropy-25-01401-f007:**
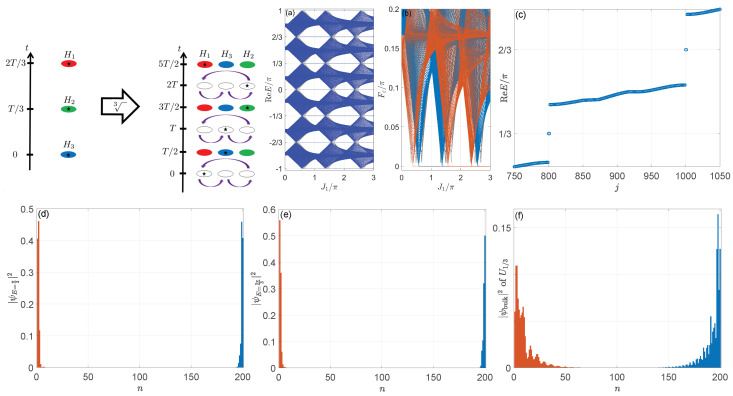
Cubic-root non-Hermitian Floquet topological insulator [265]: Top left: Schematic diagram of taking the cubic-root for a Floquet operator. (**a**) Real parts of quasienergy *E* versus $J_{1}$ for the cubic-root model $U_{1/3}$ under the OBC (blue dots) and PBC (gray dots in the background). (**b**) Gap function $F_{\varepsilon }$ of $U_{1/3}$, with blue solid and red dotted lines denoting the gap functions $F_{\pi /3}$ (=F${}_{\pi }$) and $F_{2\pi /3}$ (=$F_{0}$) under the OBC. Corresponding gap functions under PBC are given by the gray solid and dotted lines in (**b**). Other system parameters are $\left(J_{2},\mu ,\lambda \right)=\left(0.5\pi ,0.4\pi ,0.25\right)$ and the length of lattice is L=400 for (**a**,**b**). (**c**) Floquet spectrum of $U_{1/3}$ versus the state index *j* for $\left(J_{1},J_{2},\mu ,\lambda \right)=\left(\pi ,0.5\pi ,0.4\pi ,0.25\right)$, zoomed in around E=(π/3,2π/3). (**d**,**e**) show the spatial profiles of degenerate edge modes with E=π/3 and 2π/3 in (**c**). The two edge modes at each *E* are plotted in different colors. (**f**) shows a pair of typical bulk states (in different colors) of $U_{1/3}$ from (**c**), which appear around the boundaries and represent non-Hermitian Floquet skin modes.

**Figure 8 entropy-25-01401-f008:**
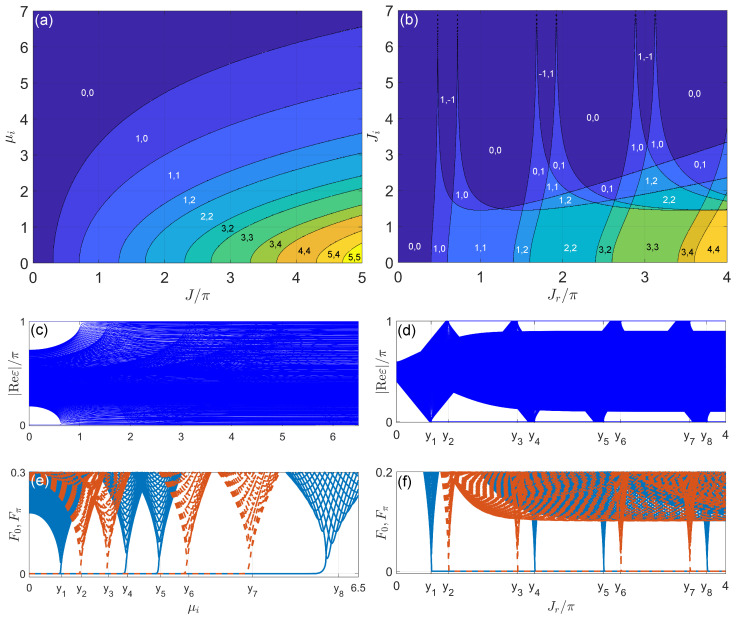
Periodically kicked non-Hermitian Kitaev chain [259]: (**a**,**b**) Topological phase diagrams under the PBC. (**a**) $\left(w_{0},w_{\pi }\right)$ versus the hopping amplitude *J* and imaginary part of onsite potential $\mu_{i}$. The real part of onsite potential and the superconducting pairing strength are $\mu_{r}=0.3\pi$ and Δ=0.5π. Every patch with a uniform color corresponds to a non-Hermitian Floquet topological superconducting phase with the winding numbers denoted therein. (**b**) $\left(w_{0},w_{\pi }\right)$ versus the real and imaginary parts of hopping amplitude $J_{r}$ and $J_{i}$. The onsite potential and superconducting pairing strength are μ=0.4π and Δ=0.9π. (**c**,**d**) Real parts of the quasienergy spectrum versus $\mu_{i}$ in (**c**) and $J_{r}$ in (**d**) under the OBC, with the lattice size L=1000. Other system parameters are $\left(\mu_{r},J,\Delta \right)=\left(0.3\pi ,4\pi ,0.5\pi \right)$ for (**c**) and $\left(\mu ,J_{i},\Delta \right)=\left(0.4\pi ,1,0.9\pi \right)$ for (**d**). (**e**,**f**) show gap functions $F_{0}$ (blue solid lines) and $F_{\pi }$ (red dashed lines) versus $\mu_{i}$ in (**e**) and $J_{r}$ in (**f**) under the OBC (see Equation (Equation 167)), with the same lattice size and system parameters as for (**c**,**d**), respectively.

**Figure 9 entropy-25-01401-f009:**
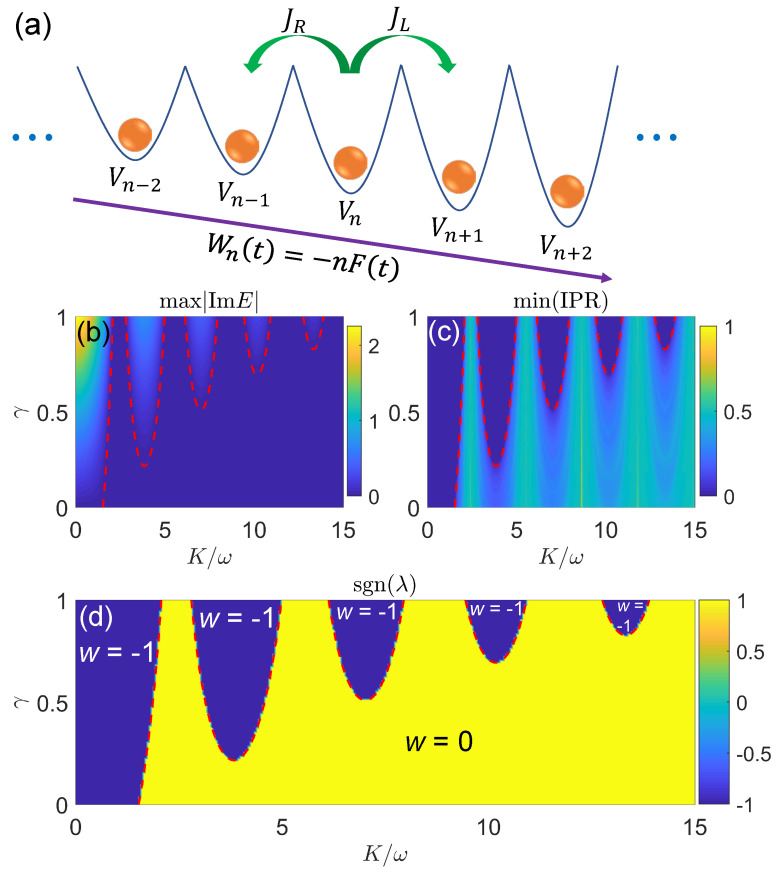
Floquet NHQC under high-frequency driving forces [264]: (**a**) Schematic diagram of the driven lattice model. (**b**) The maximal imaginary parts of quasienergies, (**c**) the minimal values of IPRs, and (**d**) the signs of Lyapunov exponents of the Floquet NHQC in Equation (Equation 169). Other system parameters are $\alpha =(\sqrt{5}-1)/2$, J=V=1, and the length of lattice is L=610. The red dashed line in each panel denotes the boundary between extended phase (with complex quasienergies) and localized phase (with real quasienergies), which satisfies the condition in Equation (Equation 171). The topological winding number *w* of each phase is denoted explicitly in (**d**).

**Figure 10 entropy-25-01401-f010:**
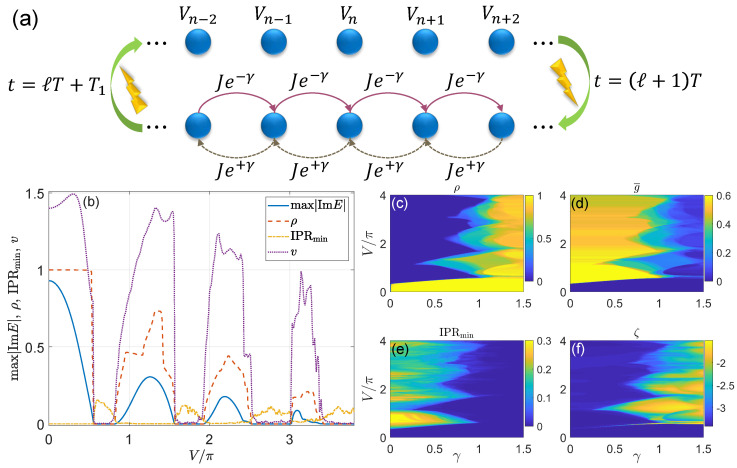
Floquet NHQC under time-periodic kickings [266]: (**a**) Schematic diagram of the driven lattice model. (**b**) Maximal imaginary parts of quasienergy (blue solid line), density of states with complex quasienergies (red dashed line), minimum of IPRs (yellow dash-dotted line), and averaged spreading velocity of a wave packet (purple dotted line) versus the onsite potential *V* under PBC. Other parameters are (J,γ)=(π/6,0.8). The lattice size is L=4181. The initial state in the calculation of *v* is $|\psi \left(0\right)\rangle =\sum_{n}\delta_{n0}|n\rangle$ and the average in Equation (Equation 126) is taken over 1000 driving periods. (**c**–**f**) show the density of states ρ, the averaged AGRs $\bar{g}$, the minimum of IPRs ${\mathrm{IPR}}_{\min}$, and the measure of critical phase ζ versus γ and *V* under PBC. The uniform part of hopping amplitude J=π/6 and the length of the lattice L=2584 for (**c**–**f**).

**Table 1 entropy-25-01401-t001:** Periodically kicked non-Hermitian Kitaev chain: Bulk topological winding numbers $\left(w_{0},w_{\pi }\right)$ and numbers of Floquet Majorana zero and π edge modes $\left(n_{0},n_{\pi }\right)$ in different topological phases. The numerical values of phase transition points $y_{1}∼y_{8}$ for $\mu_{i}$ can be analytically found and are approximately given by 0.64, 1.03, 1.57, 1.94, 2.60, 3.15, 4.41, and 6.11. The numerical values of phase transition points $y_{1}∼y_{8}$ for $J_{r}/\pi$ can also be found analytically and are given approximately by 0.42, 0.63, 1.47, 1.68, 2.51, 2.72, 3.56, and 3.77 [259].

$\mu_{i}$ (in Figure 8e)	($w_{0},w_{\pi }$)	($n_{0},n_{\pi }$)	$J_{r}$ (in Figure 8f)	($w_{0},w_{\pi }$)	($n_{0},n_{\pi }$)
$\left(0,y_{1}\right)$	(4,4)	(8,8)	$\left(0,y_{1}\right)$	(0,0)	(0,0)
$\left(y_{1},y_{2}\right)$	(3,4)	(6,8)	$\left(y_{1},y_{2}\right)$	(1,0)	(2,0)
$\left(y_{2},y_{3}\right)$	(3,3)	(6,6)	$\left(y_{2},y_{3}\right)$	(1,1)	(2,2)
$\left(y_{3},y_{4}\right)$	(3,2)	(6,4)	$\left(y_{3},y_{4}\right)$	(1,2)	(2,4)
$\left(y_{4},y_{5}\right)$	(2,2)	(4,4)	$\left(y_{4},y_{5}\right)$	(2,2)	(4,4)
$\left(y_{5},y_{6}\right)$	(1,2)	(2,4)	$\left(y_{5},y_{6}\right)$	(3,2)	(6,4)
$\left(y_{6},y_{7}\right)$	(1,1)	(2,2)	$\left(y_{6},y_{7}\right)$	(3,3)	(6,6)
$\left(y_{7},y_{8}\right)$	(1,0)	(2,0)	$\left(y_{7},y_{8}\right)$	(3,4)	(6,8)
$\left(y_{8},6.5\right)$	(0,0)	(0,0)	$\left(y_{8},4\pi \right)$	(4,4)	(8,8)

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
