# Peer review of "Non-Hermitian Floquet Topological Matter—A Review"

_entropy, 2023, doi:10.3390/e25101401_

Round 1
Reviewer 1 Report
In the manuscript, the authors systematically summarize the latest research on non-Hermitian Floquet topological insulators, which show many interesting topological phenomena and have attracted extensive attentions. The related non-Hermitian Floquet theorems are well organized and presented. In the third part, some typical effects in this field are demonstrated, but almost are based the tight-binding models. This review is timely and interesting, so I will recommend this work to be published in Entropy.In addition, I have some minor suggestions to help complete the content of this manuscript. Actually, the coupled ring resonators also are an important platform to realize Floquet topology. When the non-Hermiticity is introduced to this system, non-Hermitian skin effect can also be obtained, which are proposed in many existing works and even experimentally demonstrated in classical wave systems. This kind of non-Hermitian Floquet topological should be incorporated to this review.
Author Response
We thank our Reviewer 1 for the nice summary of our results and the recommendation of our work for publication. In response to the suggestion of our Reviewer 1, we have now added some discussions about the work on non-Hermitian Floquet systems in coupled ring resonators in the last section of our revised manuscript.
Reviewer 2 Report
In this work, the authors provide a comprehensive review of their work on non-Hermitian Floquet topological systems. They start by giving an introduction into Floquet theory with a particular focus on symmetry properties and topological characterization. They then move on to discuss their work on non-Hermitian Floquet topological models followed by a short conclusion and outlook. Indeed, the aim of this paper is to discuss the results of the authors in Refs. [254]-[264] in one work. As such, this work serves as a lecture note for researchers who wish to learn more about the work of the authors.
The paper is well written and is easy to follow. The figures are nice, and relevant. I believe the authors provide the relevant background to be able to understand the discussion of the particular models in section 3. The authors also briefly discuss works by others in the last section. While providing a self-consistent and comprehensive discussion, I find it a bit hard to see this papers as a review paper. To me, a review paper should give an overview of all the work in the field, whereas this paper gives an overview of the work of the authors in the field. Besides this point, I think it is a strong paper that deserves publishing. I do have a few comments and questions I would like the authors to address:
1. In line 159, the authors write that they consider H(t) ≠ H(t)^dagger. This made me wonder about a case, where H(t) ≠ H(t)^dagger but H(t) = H(t’)^dagger at t ≠ t’, i.e., the system is Hermitian to itself at a different time. Is this a case the authors have studied and could it be interesting?
2. Between Eqs.(100) and (101), the authors introduce a different effective Hamiltonian \tilde{H} to evolve the left initial state. Why don’t they simply use H^\dagger?
3. I was wondering whether the authors could explain a bit more what \bar{g} in Eq.(120) tell us. They say that it is zero if all bulk eigenstates are extended, while it approaches a positive constant if all bulk states are localized. How can that be seen from Eqs.(119) and (120)? Does it have to do with the fact that bands associated with localized states are flat?
4. In lines 749-750, the authors write that the appearance of an EP in k-space usually implies the breakdown of the bulk-boundary correspondence. I do not agree with this statement. EPs appear all the time in the spectrum, also for systems that preserve the bulk-boundary correspondence. It is a loop in the spectrum that is typically associated with the breakdown.
5. In lines 763-764, the authors mention that the BGZ becomes ill-defined for some parameter choices. Is this related to some other phenomenon, e.g., is this where the EP appears?
6. In line 834, it is mentioned that in static systems edge modes appear at E=0. As this depends on the model and symmetry, I was wondering whether they authors are referring in to the model they are studying, i.e., the ones mentioned in Eqs.(140) and (141) when making this statement?
7. The authors mention in passing in lines 859-861 that even though they see point gaps in their PBC spectrum they do not observe a NHSE. This is a rather interesting statement as typically these go hand in hand. Could the authors comment more on this? Is this a general feature in Floquet systems that these phenomena are decoupled? Or is there a different underlying reason here?
8. In lines 1056-1059, the authors mention that even though there is a discrepancy between the PBC and OBC spectra, the open-bulk winding numbers predict the right number of boundary states. This is a bit confusing to me. Do they mean the conventional bulk-boundary correspondence is preserved? If so, how does that work in this case?
9. While providing a clear explanation and description of each of the individual models discussed in section 3, I am missing a bit of an overarching story connecting the different studies in section 4. How do the found results for the different models relate, what is generic for Floquet non-Hermitian models, and how do the finding relate to the known results for static models?
I also have a few smaller comments/questions:
10. In line 97, it says that “a non-Hermitian system can usually be modelled by a Hamiltonian H that is not self-adjoint”. What do the authors mean by usually? What other cases exist in their opinion? I suppose this comments is related to what the authors mean exactly by a non-Hermitian system if they do not mean H ≠ H^\dagger.
11. In lines 693-696, two different cases are described: the metallic phase and the insulating phase. It says that in both cases \xi -> - infinity. Is that correct?
12. I have a general comment relating to the figures: While the figures are very nice, I believe they would improve a lot if more labels are provided. It would also improve the legibility of the paper, as one cannot get any information about what one is looking at in terms of quantities etc from just looking at the figures. In particular:
- Fig.3: Please add the values for r_x as labels to the plots in (b)-(e), add labels to (f) and (g), why does r_x go from 0 to 3 in (f) and (g) but from 0 to 5 in (h)-(j)?
- Fig.4: What is the difference between the red and grey lines in (b)-(e)? Also, please add labels
- Fig.5: Please add labels to (c)-(f), the label on the y-axis of (g) is very strange, and it is not clear at all to me what is plotted with the different colors; are the authors plotting the right states in (h)-(k)?
- Fig.6: It is not very clear what is shown in (d) and (e), what are the different colors?
- Fig.7: What do the red and blue curves in (d)-(f) correspond to?
- Fig. 8: Please add labels to (a) and (b) because it is not clear what is plotted there, also the x-axis labels of (a) and (b) are not the same; what is the difference between (c) and (d) and also what can we see here because it is hard to deduce any information from these plots
While I encountered a few small mistakes, the overall quality of the English is very good. This paper was a very pleasant read.
Author Response
Please find our response in the attached Word file.
